# Revisiting Robustness in Graph Machine Learning

**Lukas Gosch, Daniel Sturm, Simon Geisler, Stephan Günnemann**
Department of Computer Science & Munich Data Science Institute
Technical University of Munich
{l.gosch, da.sturm, s.geisler, s.guennemann}@tum.de

## Abstract

Many works show that node-level predictions of Graph Neural Networks (GNNs) are unrobust to small, often termed adversarial, changes to the graph structure. However, because manual inspection of a graph is difficult, it is unclear if the studied perturbations always preserve a core assumption of adversarial examples: that of unchanged semantic content. To address this problem, we introduce a more principled notion of an adversarial graph, which is aware of semantic content change. Using Contextual Stochastic Block Models (CSBMs) and real-world graphs, our results uncover: $i$) for a majority of nodes the prevalent perturbation models include a large fraction of perturbed graphs violating the unchanged semantics assumption; $ii$) surprisingly, all assessed GNNs show *over-robustness* - that is robustness *beyond* the point of semantic change. We find this to be a complementary phenomenon to adversarial examples and show that including the label-structure of the training graph into the inference process of GNNs significantly reduces over-robustness, while having a positive effect on test accuracy and adversarial robustness. Theoretically, leveraging our new semantics-aware notion of robustness, we prove that there is no robustness-accuracy tradeoff for inductively classifying a newly added node. [1]

## 1 Introduction

Graph Neural Networks (GNNs) are seen as state of the art for various graph learning tasks (Hu et al., 2020; 2021). However, there is strong evidence that GNNs are unrobust to changes to the underlying graph (Zügner et al., 2018; Geisler et al., 2021). This has led to the general belief that GNNs can be easily fooled by adversarial examples and many works trying to increase the robustness of GNNs through various defenses (Günnemann, 2022). Originating from the study of deep image classifiers (Szegedy et al., 2014), an adversarial example has been defined as a small perturbation, usually measured using an $\ell_p$-norm, which does not change the semantic content (i.e. category) of an image, but results in a different prediction. These perturbations are often termed unnoticeable relating to a human observer for whom a normal and an adversarially perturbed image are nearly indistinguishable (Goodfellow et al., 2015; Papernot et al., 2016). However, compared to visual tasks, it is difficult to visually inspect (large-scale) graphs. This has led to a fundamental question:

*What constitutes a small, semantics-preserving perturbation to a graph?*

The de facto standard in the literature is to measure small changes to the graph's structure using the $\ell_0$-pseudonorm (Zheng et al., 2021; Günnemann, 2022). Then, the associated threat models restrict the total number of inserted and deleted edges globally in the graph and/or locally per node. However, if the observation of semantic content preservation for these kind of perturbation models transfers to the graph domain can be questioned: Due to the majority of low-degree nodes in real-world graphs, small $\ell_0$-norm restrictions still allow to completely remove a significant number of nodes from their original neighbourhood. Only few works introduce measures beyond $\ell_0$-norm restrictions. In particular, it was proposed to additionally use different global graph properties as a proxy for unnoticeability, such as the degree distribution (Zügner et al., 2018), degree assortativity (Li et al., 2021), or other homophily metrics (Chen et al., 2022).

---

[1] Project page: https://www.cs.cit.tum.de/daml/revisiting-robustness/

While these are important first steps, the exact relation between preserving certain graph properties and the graph's semantic content (e.g., node-categories) is unclear (see Appendix B). For instance, one can completely rewire the graph by iteratively interchanging the endpoints of two randomly selected edges and preserve the global degree distribution. As a result, current literature lacks a principled understanding of semantics-preservation in their employed notions of smallness as well as robustness studies using threat models only including provable semantics-preserving perturbations to a graph.

We bridge this gap by being the first to directly address the problem of exactly measuring (node-level) semantic content preservation

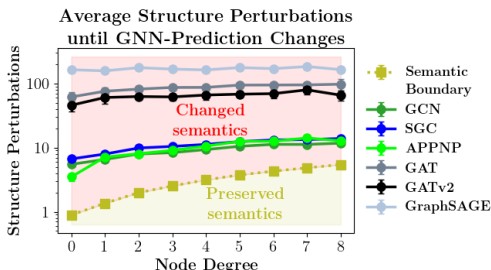

Figure 1: Average degree-dependent node-classification robustness. Semantic boundary indicates when the semantics (i.e., the most likely class) of a node of a given degree changes on average. Data from CSBM graphs[2]. All GNNs show robustness *beyond* the point of semantic change.

in a graph under structure perturbations. Surprisingly, using Contextual Stochastic Block Models (CSBMs), this leads us to discover a novel phenomenon: GNNs show strong robustness *beyond* the point of semantic change (see Figure 1). This does *not* contradict the existence of adversarial examples for the same GNNs. Related to the small degree of nodes, we find that common perturbation sets include both: graphs which are truly adversarial as well as graphs with changed semantic content. Our contributions are:

1. We define a semantics-aware notion of adversarial robustness (Section 3) for node-level predictions. Using this, we introduce a novel concept into the graph domain: *over-robustness* - that is (un-wanted) robustness against admissible perturbations with changed semantic content (i.e., changed ground-truth labels).

2. Using CSBMs, we find: $i$) common perturbations sets, next to truly adversarial examples, include a large fraction of graphs with changed semantic content (Section 5.1); $ii$) all examined GNNs show significant *over-robustness* to these graphs (Section 5.2) and we observe similar patterns on real-world datasets (Section 5.2.1). Using $\ell_0$-norm bounded adversaries on CSBM graphs, we find a considerable amount of a conventional adversarial robustness to be in fact over-robustness.

3. Including the known label-structure through Label Propagation (LP) (Huang et al., 2021) into the inference process of GNNs significantly reduces over-robustness with no negative effects on test accuracy or adversarial robustness (Section 5.2) and similar behaviour on real-world graphs.

4. Using semantic awareness, we prove the existence of a model achieving both, optimal robustness and accuracy in classifying an inductively sampled node (Section 4.1), i.e., no robustness-accuracy tradeoff for a non-i.i.d. data setting.

## 2 PRELIMINARIES

Let $n$ be the number of nodes and $d$ the feature dimension. We denote the node feature matrix $\mathbf{X} \in \mathbb{R}^{n \times d}$, the (symmetric) adjacency matrix $\mathbf{A} \in \{0, 1\}^{n \times n}$, and the node labels $y \in \{0, 1\}^n$ of which $y_L \in \{0, 1\}^l, l \leq n$ are known. We assume a graph has been sampled from a graph data generating distribution $\mathcal{D}_n$ denoted as $(\mathbf{X}, \mathbf{A}, y) \sim \mathcal{D}_n$. We study inductive node classification (Zheng et al., 2021). Due to the non-i.i.d data generation, a node-classifier $f$ may depend its decision on the whole known graph $(\mathbf{X}, \mathbf{A}, y_L)$. As a result, we write $f(\mathbf{X}, \mathbf{A}, y_L)_v$ to denote the classification of a node $v$. All GNNs covered in this work are a function $f(\mathbf{X}, \mathbf{A})$ only depending on the node features $\mathbf{X}$ and adjacency matrix $\mathbf{A}$. A list of the used symbols and abbreviations can be found in Appendix A.

**Label Propagation.** We use label spreading (Zhou et al., 2004), which builts a classifier $f(\mathbf{A}, y_L)$ by taking the row-wise $\arg\max$ of the iterate $F^t = \alpha \mathbf{D}^{-1/2} \mathbf{A} \mathbf{D}^{-1/2} F^{t-1} + (1 - \alpha)Y$, with $\mathbf{D}$ being the diagonal degree matrix; $Y \in \mathbb{R}^{n \times c}$ with $Y_{ij} = 1$ if $i \leq l$ and $y_L^i = j$, otherwise $Y_{ij} = 0$; and $\alpha \in [0, 1]$. Similar to Huang et al. (2021), we combine LP and GNNs by replacing the zero-rows for $i > l$ in $Y$ with GNN soft-predictions, effectively forming a function $f(\mathbf{X}, \mathbf{A}, y_L)$.

---

[2]CSBMs parametrized as outlined in Section 5 using $K = 1.5$ and $\ell_2$-*weak* attack.

**Adversarial Robustness.** We study (inductive) evasion attacks, that is an adversary $\mathcal{A}$ can produce a perturbed graph $(\tilde{\mathbf{X}}, \tilde{\mathbf{A}}) \in \mathcal{B}(\mathbf{X}, \mathbf{A})$ given the clean data $(\mathbf{X}, \mathbf{A})$ with the goal to fool a given node-classifier $f(\tilde{\mathbf{X}}, \tilde{\mathbf{A}}, y_L)_v \neq f(\mathbf{X}, \mathbf{A}, y_L)_v$ on nodes $v$. The perturbation set $\mathcal{B}(\mathbf{X}, \mathbf{A})$ collects all admissible perturbed graphs, defined by the threat model. We focus on direct structure attacks (Zügner et al., 2018). This means $\mathcal{A}$ can remove or add at most $\Delta \in \mathbb{N}$ edges for a target node $v$.

**Data Model.** We leverage *Contextual Stochastic Block Models* (CSBMs) (Deshpande et al., 2018) to generate synthetic graphs with analytically tractable distributions. It defines edge probabilities $p$ between same-class nodes and $q$ between different-class nodes and node-features are drawn from a Gaussian mixture model. Sampling from a CSBM can be written as an iterative process over nodes $i \in [n]$: 1) Sample label $y_i \sim \text{Ber}(1/2)$ (Bernoulli distribution). 2) Sample feature vector $\mathbf{X}_{i,:}|y_i \sim \mathcal{N}((2y_i - 1)\mu, \sigma\mathbf{I})$ with $\mu \in \mathbb{R}^d, \sigma \in \mathbb{R}$. 3) For all $j \in [n], j < i$ sample $\mathbf{A}_{j,i} \sim \text{Ber}(p)$ if $y_i = y_j$ and $\mathbf{A}_{j,i} \sim \text{Ber}(q)$ otherwise, and set $\mathbf{A}_{i,j} = \mathbf{A}_{j,i}$. We denote this $(\mathbf{X}, \mathbf{A}, y) \sim \text{CSBM}_{n,p,q}^{\mu,\sigma^2}$. To inductively add $m$ nodes, one repeats the above process for $i = n+1, \ldots, m$. Fountoulakis et al. (2022) show that depending on the distance of the means $d(-\mu, \mu)$, one can separate an easy regime, where a linear classifier ignoring $\mathbf{A}$ can perfectly separate the data and a hard regime, defined by $d(-\mu, \mu) = K\sigma$, with $0 < K \leq \mathcal{O}(\sqrt{\log n})$, where this is not possible. CSBMs are commonly used to study transductive tasks. Understanding the sampling process as an iteration extends their application to inductive node classification. For an alternative we call *Contextual Barabási–Albert Model with Community Structure* (CBA) see Appendix C. Appendix D discusses the model choices.

## 3 REVISITING ADVERSARIAL PERTURBATIONS

Given a clean graph $(\mathbf{X}, \mathbf{A}, y_L)$ and target node $v$. The perturbation set $\mathcal{B}(\mathbf{X}, \mathbf{A})$ comprises all possible (perturbed) graphs $(\tilde{\mathbf{X}}, \tilde{\mathbf{A}})$, which can be chosen by an adversary $\mathcal{A}$, with the goal to change the prediction of a node classifier $f$, i.e., $f(\tilde{\mathbf{X}}, \tilde{\mathbf{A}}, y_L)_v \neq f(\mathbf{X}, \mathbf{A}, y_L)_v$. The prevalent works implicitly assume that every $(\tilde{\mathbf{X}}, \tilde{\mathbf{A}}) \in \mathcal{B}(\mathbf{X}, \mathbf{A})$ preserves the node-level semantic content of the clean graph, i.e., the original ground-truth label of $v$. If we would have an oracle $\Omega$, which tells us the semantic content the known graph encodes about $v$, this assumption can be made explicit by writing $\Omega(\tilde{\mathbf{X}}, \tilde{\mathbf{A}}, y_L)_v = \Omega(\mathbf{X}, \mathbf{A}, y_L)_v$. Usually, we do not have access to such an oracle. However, we can try to model its behaviour by introducing a reference or base node classifier $g$. Then, the idea is to use $g$ to indicate semantic content change and thereby, define the semantic boundary (see Figure 1). Exemplary, $g$ could be derived from knowledge about the data generating process. We do so in Section 4 and 5, where we use the (Bayes) optimal classifier for CSBMs as $g$. Note that labels themselves are often generated following a base classifier. Exemplary, this can be humans labelling selected nodes in a graph to generate a dataset for (semi-) supervised learning. Using a reference classifier $g$ as a proxy for semantic content enables us to make a refined definition of an adversarial graph, which makes the unchanged-semantics assumption explicit:

**Definition 1.** *Let $f$ be a node classifier and $g$ a reference node classifier. Then the perturbed graph $(\tilde{\mathbf{X}}, \tilde{\mathbf{A}}) \in \mathcal{B}(\mathbf{X}, \mathbf{A})$ chosen by an adversary $\mathcal{A}$ is said to be adversarial for $f$ at node $v$ w.r.t. the reference classifier $g$ if the following conditions are satisfied:*

$$\begin{aligned} &i. \quad f(\mathbf{X}, \mathbf{A}, y_L)_v = g(\mathbf{X}, \mathbf{A}, y_L)_v && \textit{(correct clean prediction)} \\ &ii. \quad g(\tilde{\mathbf{X}}, \tilde{\mathbf{A}}, y_L)_v = g(\mathbf{X}, \mathbf{A}, y_L)_v && \textit{(perturbation preserves semantics)} \\ &iii. \quad f(\tilde{\mathbf{X}}, \tilde{\mathbf{A}}, y_L)_v \neq g(\mathbf{X}, \mathbf{A}, y_L)_v && \textit{(node classifier changes prediction)} \end{aligned}$$

Definition 1 says that a perturbed graph $(\tilde{\mathbf{X}}, \tilde{\mathbf{A}}) \in \mathcal{B}(\mathbf{X}, \mathbf{A})$ only then is adversarial, if $(\tilde{\mathbf{X}}, \tilde{\mathbf{A}})$ does not only change the prediction of the node classifier $f$ $(iii)$, but also lets the original label unchanged $(ii)$. The first constraint $(i)$ stems from the fact that if $f$ and $g$ disagree on the clean graph at node $v$ this should represent a case of misclassification captured by standard error metrics such as accuracy.

Suggala et al. (2019) use the concept of a reference classifier to, in similar spirit, define semantics-aware adversarial perturbations for i.i.d. data, with a focus on the image domain. However, what has not been considered is that the reference classifier allows us to characterize the exact opposite behaviour of an adversarial example: if a classifier $f$ does not change its prediction for a perturbed graph $(\tilde{\mathbf{X}}, \tilde{\mathbf{A}})$ even though the semantic content has changed. As this would mean that $f$ is robust beyond the point of semantic change, we call this behaviour *over-robustness*:

**Definition 2.** *Let $f$ be a node classifier and $g$ a reference node classifier. Then the perturbed graph $(\tilde{\mathbf{X}}, \tilde{\mathbf{A}}) \in \mathcal{B}(\mathbf{X}, \mathbf{A})$ chosen by an adversary $\mathcal{A}$ is said to be an over-robust example for $f$ at node $v$ w.r.t. the reference classifier $g$ if the following conditions are satisfied:*

$$i. \quad f(\mathbf{X}, \mathbf{A}, y_L)_v = g(\mathbf{X}, \mathbf{A}, y_L)_v \qquad \textit{(correct clean prediction)}$$

$$ii. \quad g(\tilde{\mathbf{X}}, \tilde{\mathbf{A}}, y_L)_v \neq g(\mathbf{X}, \mathbf{A}, y_L)_v \qquad \textit{(perturbation changes semantics)}$$

$$iii. \quad f(\tilde{\mathbf{X}}, \tilde{\mathbf{A}}, y_L)_v = g(\mathbf{X}, \mathbf{A}, y_L)_v \qquad \textit{(node classifier stays unchanged)}$$

*If there exists such an over-robust example, we call $f$ over-robust at node $v$ w.r.t. $g$.*

Definition 2 is of particular interest in the graph domain, where perturbation sets $\mathcal{B}(\mathbf{X}, \mathbf{A})$ often include graphs $(\tilde{\mathbf{X}}, \tilde{\mathbf{A}})$, which allow significant changes to the neighbourhood structure of $v$, but do not allow easy manual content inspection. Indeed, Section 5 shows that all assessed GNNs are over-robust for common choices of $\mathcal{B}(\mathbf{X}, \mathbf{A})$ for many test nodes $v$ in CSBM graphs. A similar phenomenon for image data has been discussed by Tramèr et al. (2020a).

Now, let us contrast over- to adversarial robustness. In Figure 2a the decision boundary of a classifier $f$ follows the one of a base classifier $g$ except for the dotted line. The dashed area between $f$ and $g$ is a region of over-robustness for the blue class and of adversarial examples for the red class. In practice, the extent of the perturbation sets $\mathcal{B}(\cdot)$ is bounded. As a result, using adversarial examples, it is only possible to measure the right boundary of the dashed area (see Figure 2b). The concept of over-robustness allows us to additionally measure the left boundary and hence, provides a more complete picture of the robustness of $f$. Note that the blue points, using conventional adversarial robustness, are judged robust in the whole of $\mathcal{B}(\cdot)$ even though their true class changed. Semantic-aware adversarial robustness allows to (correctly) cut off $\mathcal{B}(\cdot)$ at the decision boundary of $g$.

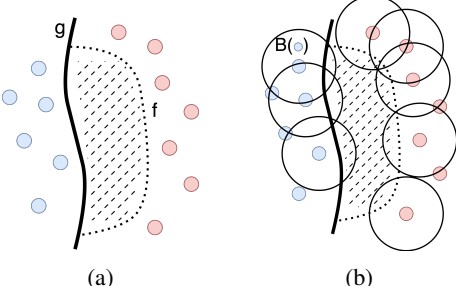

(a)          (b)

Figure 2: a) Decision boundary of classifier $f$ follows a base classifier $g$ except for the dotted line. b) Finite perturbation sets $\mathcal{B}(\cdot)$ intersect only from one side with the dashed area. Over- and adversarial robustness are needed to characterize robustness.

Contrary to Figure 2a, a region of over-robust examples cannot always be identified as a region of adversarial examples for different data points. Denote $\mathcal{G} = (\mathbf{X}, \mathbf{A})$ and collect all over-robust examples in a set $\mathcal{B}_O(\mathcal{G}, v) \subset \mathcal{B}(\mathcal{G})$ and adversarial examples in a set $\mathcal{B}_A(\mathcal{G}, v) \subset \mathcal{B}(\mathcal{G})$. Note that it follows from Definition 1 and 2 that $\mathcal{B}_O(\mathcal{G}, v) \cap \mathcal{B}_A(\mathcal{G}, v) = \emptyset$. Now, we find that in general, for a given over-robust example $\tilde{\mathcal{G}} \in \mathcal{B}(\mathcal{G})$ one can't always find a corresponding clean graph $\mathcal{G}' \neq \mathcal{G}$ not in $\mathcal{B}(\mathcal{G})$ for which $\tilde{\mathcal{G}}$ is an adversarial example. It suffices to look at the case of $f$ being constant:

**Proposition 1.** *Given $f$ is a constant classifier, then $\mathcal{B}_A(\mathcal{G}, v)$ is empty for every possible graph $\mathcal{G}$.*

This follows as $f$ can never fulfill both, item $(i)$ and item $(iii)$ in Definition 1. However, over-robust examples for $f$ may exists. Every example in $\mathcal{B}(\mathcal{G})$ will be over-robust, for which $g$ changes its label for node $v$. This also shows that *over-robustness* can differentiate two classifiers, which have the same adversarial robustness, but one has learned a better decision boundary, while the other has not. We further develop other interesting conceptual cases in Appendix E.

## 4   BAYES CLASSIFIER AS REFERENCE CLASSIFIER

In the following, we derive a Bayes optimal classifier to use as a reference classifier $g$. Using the Bayes decision rule as $g$ is a natural choice, because, as shown below, it provides us with the information, if another class is now more likely based on the true data generating distribution and hence, is the closest we can get to a semantic content returning oracle $\Omega$ (see Section 3).

Assume that we want to learn an inductive node-classifier $f$ on a given, fully labeled graph $(\mathbf{X}, \mathbf{A}, y) \sim \mathcal{D}_n$ with $n$ nodes. We will focus on the most simple case of classifying an inductively sampled node. We denote the conditional distribution over graphs with an inductively added

node as $\mathcal{D}(\mathbf{X}, \mathbf{A}, y)$. Then, the target node $v$ corresponds to the newly sampled, $n+1$-th node. How well our classifier $f$ generalizes to the newly added node is captured by the expected $0/1$-loss of $f$:

$$\mathop{\mathbb{E}}_{(\mathbf{X}', \mathbf{A}', y') \sim \mathcal{D}(\mathbf{X}, \mathbf{A}, y)} [\ell_{0/1}(y_v', f(\mathbf{X}', \mathbf{A}', y)_v] \tag{1}$$

To derive Bayes optimality, we have to find an optimal classifier $f^*$ for $v$, depending on $(\mathbf{X}', \mathbf{A}', y)$, minimizing (1). The following theorem shows that, similar to inductive classification for i.i.d. data, $f^*$ should choose the most likely class based on the seen data (Proof in Appendix F.1):

**Theorem 1.** *The Bayes optimal classifier, minimizing the expected $0/1$-loss (1), is $f^*(\mathbf{X}', \mathbf{A}', y)_v = \arg\max_{\hat{y} \in \{0,1\}} \mathbb{P}[y_v' = \hat{y} | \mathbf{X}', \mathbf{A}', y]$.*

### 4.1 ROBUSTNESS-ACCURACY TRADEOFF

We show that given a Bayes optimal reference classifier $g$, our robustness notions (Definition 1 and 2) imply that optimal robustness, i.e., the non-existence of adversarial and over-robust examples for $f$, is possible while preserving good generalization in the sense of (1). Our argumentation for the non-i.i.d. graph data case takes inspiration from Suggala et al. (2019)'s study for i.i.d. data. In the following, we assume we are given a graph $\mathcal{G}'$ sampled from $\mathcal{D}(\mathcal{G}, y)$, where $\mathcal{G} = (\mathbf{X}, \mathbf{A})$ represents a fully labeled training graph with $n$ nodes and labels $y$. Let $v$ refer to the $n+1$-th (inductively added) node. First, observe from Definition 1 that a graph $\tilde{\mathcal{G}}' \in \mathcal{B}(\mathcal{G}')$ is an adversarial example, iff $g(\tilde{\mathcal{G}}', y)_v = g(\mathcal{G}', y)_v$ and $\ell_{0/1}(f(\tilde{\mathcal{G}}', y)_v, g(\mathcal{G}', y)_v) - \ell_{0/1}(f(\mathcal{G}', y)_v, g(\mathcal{G}', y)_v) = 1$. In analogy to the generalization error (1), we define the adversarial generalization error:

**Definition 3.** *Let $f$ be a node classifier and $g$ a reference node classifier. Assume $\mathcal{G}'$ is itself in $\mathcal{B}(\mathcal{G}')$. Then the expected adversarial $0/1$-loss for an inductively added target node $v$ is defined as the probability that a conditionally sampled graph can be adversarially perturbed:*

$$\mathop{\mathbb{E}}_{(\mathcal{G}', y') \sim \mathcal{D}(\mathcal{G}, y)} \left[ \max_{\substack{\tilde{\mathcal{G}}' \in \mathcal{B}(\mathcal{G}') \\ g(\tilde{\mathcal{G}}', y)_v = g(\mathcal{G}', y)_v}} \left\{ \ell_{0/1}(f(\tilde{\mathcal{G}}', y)_v, g(\mathcal{G}', y)_v) - \ell_{0/1}(f(\mathcal{G}', y)_v, g(\mathcal{G}', y)_v) \right\} \right] \tag{2}$$

From Definition 2 it follows that a graph $\tilde{\mathcal{G}}' \in \mathcal{B}(\mathcal{G})$ is an over-robust example, iff $f(\tilde{\mathcal{G}}', y)_v = f(\mathcal{G}', y)_v$ and $\ell_{0/1}(g(\tilde{\mathcal{G}}', y)_v, f(\mathcal{G}', y)_v) - \ell_{0/1}(g(\mathcal{G}', y)_v, f(\mathcal{G}', y)_v) = 1$. Thus, we define:

**Definition 4.** *Let $f$ be a node classifier and $g$ a reference node classifier. Assume $\mathcal{G}'$ is itself in $\mathcal{B}(\mathcal{G}')$. Then the expected over-robust $0/1$-loss for an inductively added target node $v$ is defined as the probability that a conditionally sampled graph can be perturbed to be an over-robust example:*

$$\mathop{\mathbb{E}}_{(\mathcal{G}', y') \sim \mathcal{D}(\mathcal{G}, y)} \left[ \max_{\substack{\tilde{\mathcal{G}}' \in \mathcal{B}(\mathcal{G}') \\ f(\tilde{\mathcal{G}}', y)_v = f(\mathcal{G}', y)_v}} \left\{ \ell_{0/1}(g(\tilde{\mathcal{G}}', y)_v, f(\mathcal{G}', y)_v) - \ell_{0/1}(g(\mathcal{G}', y)_v, f(\mathcal{G}', y)_v) \right\} \right] \tag{3}$$

We denote the expected adversarial $0/1$-loss as $\mathscr{L}_{\mathcal{G}, y}^{adv}(f, g)_v$ and the expected over-robust $0/1$-loss as $\mathscr{L}_{\mathcal{G}, y}^{over}(f, g)_v$. Minimizing only one of the robust objectives and disregarding the standard loss (1), may not yield a sensible classifier. Exemplary, the adversarial loss (2) achieves its minimal value of $0$ for a constant classifier. Therefore, we collect them in an overall expected robustness loss term:

$$\mathscr{L}_{\mathcal{G}, y}^{rob}(f, g)_v = \lambda_1 \mathscr{L}_{\mathcal{G}, y}^{adv}(f, g)_v + \lambda_2 \mathscr{L}_{\mathcal{G}, y}^{over}(f, g)_v \tag{4}$$

where $\lambda_1 \geq 0$ and $\lambda_2 \geq 0$ define how much weight we give to the adversarial and the over-robust loss. Now, we want to find a node-classifier $f$ with small robust and standard loss. Denoting the expected $0/1$-loss (1) as $\mathscr{L}_{\mathcal{G}, y}(f)_v$, this leads us to optimize the following objective:

$$\arg\min_{f \in \mathcal{H}} \mathscr{L}_{\mathcal{G}, y}(f)_v + \lambda \mathscr{L}_{\mathcal{G}, y}^{rob}(f, g)_v \tag{5}$$

where $\lambda \geq 0$ defines a tradeoff between standard accuracy and robustness and $\mathcal{H}$ represent a set of admissible functions, e.g. defined by a chosen class of GNNs. Now, the following holds:

**Theorem 2.** *Assume a set of admissible functions $\mathcal{H}$, which includes a Bayes optimal classifier $f^*_{Bayes}$ and let the reference classifier $g$ be itself a Bayes optimal classifier. Then, any minimizer $f^* \in \mathcal{H}$ of (5) is a Bayes optimal classifier.*

Proof see Appendix F.2.1. Theorem 2 implies that minimizing both, the standard and robust loss for any $\lambda \geq 0$, always yields a Bayes optimal classifier. Therefore, optimizing for robustness does not tradeoff accuracy of the found classifier by (5) and hence, establishes that classifying an inductively sampled node does not suffer from a robustness-accuracy tradeoff. Theorem 2 raises the important question if common GNNs define function classes $\mathcal{H}_{GNN}$ expressive enough to represent a Bayes classifier for (1) and hence, can achieve optimal robustness. Theorem 3 in Appendix F.2.2 shows that only being a minimizer for the robust loss $\mathcal{L}_{\mathcal{G},y}^{rob}(f,g)_v$ does not imply good generalization.

## 5 RESULTS

Using Contextual Stochastic Block Models (CSBMs) we measure the extent of semantic content violations in common threat models (Section 5.1). Then, we study over-robustness in CSBMs (Section 5.2) and real-world graphs (Section 5.2.1). In CSBMs, we use the Bayes optimal classifier (Theorem 1), denoted $g$, to measure semantic change. The robustness of the Bayes classifier defines the maximal meaningful robustness achievable (see Section 3). For results using the Contextual Barabási–Albert Model with Community Structure (CBA), we refer to Appendix I.

| Accuracy (Bayes) | | |
|---|---|---|
| K | X | (X, A) |
| 0.1 | 50.8% | 89.7% |
| 0.5 | 59.0% | 90.3% |
| 1.0 | 68.4% | 91.7% |
| 1.5 | 76.5% | 93.1% |
| 2.0 | 83.4% | 94.7% |
| 3.0 | 92.6% | 97.4% |
| 4.0 | 97.5% | 99.0% |
| 5.0 | 99.3% | 99.8% |

Table 1: Mean accuracy of the Bayes optimal classifier on test nodes $v$ with $(\mathbf{X}, \mathbf{A})$ and without $(\mathbf{X})$ structure information.

**Experimental Setup.** We sample training graphs with $n = 1000$ nodes from a $\text{CSBM}_{n,p,q}^{\mu,\sigma^2}$ in the hard regime (Section 2). Each element of the class mean vector $\mu \in \mathbb{R}^d$ is set to $K\sigma/2\sqrt{d}$, resulting in a distance between the class means of $K\sigma$. We set $\sigma = 1$ and vary $K$ from close to *no* discriminative features $K = 0.1$ to making structure information *unnecessary* $K = 5$ (see Table 1). We choose $p = 0.63\%$ and $q = 0.15\%$ resulting in the expected number of same-class and different-class edges for a given node to fit CORA (Sen et al., 2008). Following Fountoulakis et al. (2022), we set $d = \lfloor n/\ln^2(n) \rfloor = 21$. We use an 80%/20% train/validation split on the nodes. As usual in the inductive setting, we remove the validation nodes from the graph during training. At test time, we inductively sample 1000 times an additional node conditioned on the training graph. For each $K$, we sample 10 different training graphs.

**Models and Attacks.** We study a wide range of popular GNN architectures: Graph Convolutional Networks (GCN) (Kipf & Welling, 2017), Simplified Graph Convolutions (SGC) (Wu et al., 2019a), Graph Attention Networks (GAT) (Veličković et al., 2018), GATv2 (Brody et al., 2022), APPNP (Gasteiger et al., 2019), and GraphSAGE (Hamilton et al., 2017). Furthermore, we study a simple Multi-Layer Perceptron (MLP) and Label Propagation (LP) (Zhou et al., 2004). The combination of a model with LP (Huang et al., 2021) is denoted by *Model+LP* (see Section 2). To find adversarial examples, we employ the established attacks *Nettack* (Zügner et al., 2018), *DICE* (random addition of different-class edges) (Waniek et al., 2018), *GR-BCD* (Geisler et al., 2021) and *SGA* (Geisler et al., 2021). Furthermore, we employ $\ell_2$-*strong*: Connect to the most distant different-class nodes in feature space using $\ell_2$-norm. To find over-robust examples, we use a "weak" attack, which we call $\ell_2$-*weak*: Connect to the closest different-class nodes in $\ell_2$-norm. Theorem 4 in Appendix F.3 shows that a strategy to change, with least structure changes, the true most likely class on CSBMs, i.e., an "optimal attack" against the Bayes classifier $g$, is given by (arbitrarily) disconnecting same-class edges and adding different-class edges to the target node. Therefore, the attacks *DICE*, $\ell_2$-*strong*, and $\ell_2$-*weak* have the same effect on the semantic content of a graph. We investigate perturbation sets $\mathcal{B}_\Delta(\cdot)$ with local budgets $\Delta$ from 1 up to the degree (deg) of a node $+2$, similarly to Zügner et al. (2018). Further details, including the hyperparameter settings can be found in Appendix H.

**Robustness Metrics.** To analyse the robustness of varying models across different graphs, we need to develop comparable metrics summarizing the robustness properties of a model $f$ on a given graph. First, to correct for the different degrees of nodes, we measure the adversarial robustness of $f$ (w.r.t. $g$) at node $v$ relative to $v$'s degree and average over all test nodes $V'$[3]:

$$R(f,g) = \frac{1}{|V'|}\sum_{v \in V'} \frac{\text{Robustness}(f,g,v)}{\deg(v)} \tag{6}$$

---

[3]Excluding degree 0 nodes. This is one limitation of this metric, however, these are very rare in the generated CSBM graphs and non-existing in common benchmark datasets such as CORA.

where $g$ represents the Bayes classifier and hence, Robustness$(f, g, v)$ refers to the (minimal) number of semantics-aware structure changes $f$ is robust against (Definition 1). Exemplary, $R(f, g) = 0.5$ would mean that on average, node predictions are robust against changing 50% of the neighbourhood structure. Using the true labels $y$ instead of a reference classifier $g$ in (6), yields the conventional (degree-corrected) adversarial robustness, unaware of semantic change, which we denote $R(f) := R(f, y)$. To measure **over-robustness**, we measure the fraction of conventional adversarial robustness $R(f)$, which cannot be explained by semantic-preserving robustness $R(f, g)$: $R^{over} = 1 - R(f, g)/R(f)$. Exemplary, $R^{over} = 0.2$ means that 20% of the measured robustness is robustness beyond semantic change. In Appendix G we present a metric for semantic-aware adversarial robustness and how to calculate an overall robustness measure using the harmonic mean.

## 5.1 EXTENT OF SEMANTIC CONTENT CHANGE IN COMMON PERTURBATION MODELS

We investigate how prevalent perturbed graphs with changed semantic content are for common perturbation model choices. We denote the perturbation set allowing a local budget of $\Delta$ edge perturbations as $\mathcal{B}_\Delta(\cdot)$. Table 2 shows the fraction of test nodes, for which we find perturbed graphs in $\mathcal{B}_\Delta(\cdot)$ with changed ground truth labels. Surprisingly, even for very modest budgets, if structure matters ($K \leq 3$), this fraction is significant. Exemplary, for $K=1.0$ and $\mathcal{B}_{\deg+2}(\cdot)$, we find perturbed graphs with changed semantic content for 99.4% of the target nodes. This establishes for CSBMs, a *negative* answer to a question formulated in the introduction: *If structure matters, does completely reconnecting a node preserve its semantic content?* Similar to CSBMs, nodes in real-world graphs have mainly low-degree (Figure 12 in Appendix J.1). This provides evidence that

| Threat | K | | | | | | | |
|---|---|---|---|---|---|---|---|---|
| Models | 0.1 | 0.5 | 1.0 | 1.5 | 2.0 | 3.0 | 4.0 | 5.0 |
| $\mathcal{B}_1(\cdot)$ | 14.3 | 11.2 | 9.1 | 6.8 | 4.4 | 1.9 | 0.8 | 0.2 |
| $\mathcal{B}_2(\cdot)$ | 35.9 | 31.2 | 25.7 | 19.8 | 14.1 | 6.2 | 2.2 | 0.7 |
| $\mathcal{B}_3(\cdot)$ | 58.5 | 53.8 | 46.8 | 38.2 | 28.8 | 14.3 | 5.1 | 1.7 |
| $\mathcal{B}_4(\cdot)$ | 76.5 | 73.0 | 66.6 | 58.1 | 47.0 | 25.7 | 9.8 | 3.4 |
| $\mathcal{B}_{\deg}(\cdot)$ | 75.7 | 60.0 | 55.4 | 49.1 | 39.6 | 21.9 | 9.0 | 3.2 |
| $\mathcal{B}_{\deg+2}(\cdot)$ | 100 | 100 | 99.4 | 92.9 | 80.5 | 51.7 | 24.8 | 9.1 |

Table 2: Percentage (%) of test nodes for which perturbed graphs in $\mathcal{B}_\Delta(\cdot)$ violate semantic content preservation. Calculated by connecting $\Delta$ different class nodes ($\ell_2$-weak) to every target node. For $K \geq 4.0$ structure is not necessary for good generalization (Table 1). Results using other attacks are similar (see Appendix J.3). Standard deviations are insignificant and hence, omitted.

similar conclusion could be drawn for certain real-world graphs. The examined $\mathcal{B}_\Delta(\cdot)$ subsume all threat models against edge-perturbations employing the $\ell_0$-norm to measure small changes to the graph's structure, as we investigate the lowest choices of local budgets possible[4] and for these already find a large percentage of perturbed graphs, violating the semantics-preservation assumption.

Values in Table 2 are lower bounds on the prevalence of graphs in $\mathcal{B}_\Delta(\cdot)$ with changed semantic content. We calculate these values by connecting $\Delta$ different-class nodes ($\ell_2$-weak) to every target node and hence, the constructed perturbed graphs $\tilde{\mathcal{G}}$ are at the boundary of $\mathcal{B}_\Delta(\cdot)$. Then, we count how many $\tilde{\mathcal{G}}$ have changed their true most likely class using the Bayes classifier $g$. Thus, exemplary, if a classifier $f$ is robust against $\mathcal{B}_3(\cdot)$ for CSBMs with $K=1.0$, $f$ classifies the found graphs at the boundary of $\mathcal{B}_3(\cdot)$ wrong in 47% of cases. Therefore, $f$ shows high **over-robustness** $R^{over}$.

## 5.2 OVER-ROBUSTNESS OF GRAPH NEURAL NETWORKS

Section 5.1 establishes the **existence** of perturbed graphs with changed semantic content in common threat models. Now, we examine how much of the measured robustness of common GNNs can actually be attributed to over-robustness ($R^{over}$), i.e., to robustness beyond semantic change. As a qualitative example, we study a local budget of the degree of the target nodes $\mathcal{B}_{\deg}(\cdot)$, results on other perturbation sets can be found in Appendix J.4. Figure 3a shows the over-robustness of GNNs when attacking their classification of inductively added nodes. For $K \leq 3$ the graph structure is relevant in the prediction (see Table 1). We find that in this regime, a significant amount of the measured robustness of **all** GNNs can be attributed to **over-robustness**. Exemplary, 30.3% of the conventional adversarial robustness measurement of a GCN for $K = 0.5$ turns out to be over-

---

[4]Global budgets again result in local edge-perturbations, however, now commonly allowing for way stronger perturbations than $\mathcal{B}_{\deg+2}(\cdot)$ to individual nodes. The number of allowed perturbation is usually set to a small one- or two-digit percentage of the total number of edges in the graph. For CORA with 5278 edges, a relative budget of 5% leads to a budget of 263 edge-changes, which can be distributed in the graph without restriction.

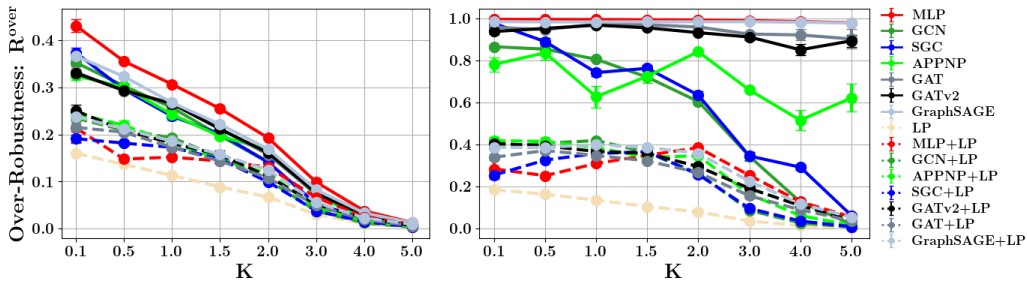

(a) Local Budget $\Delta$: Degree of Node        (b) No Budget Restriction.

Figure 3: Fraction of Robustness beyond Semantic Change (Attack: $\ell_2$-weak; Plot with Standard Error). (a) A large part of the measured robustness against $\mathcal{B}_{\text{deg}}(\cdot)$ can be attributed to over-robustness. (b) Applying $\ell_2$-weak without budget restriction until it changes a classifiers prediction. Note that over-robustness can be significantly reduced by label propagation and for (b) that some over-robust models, especially for high $K$, show high adversarial robustness (Appendix J.6).

robustness. LP achieves the lowest $R^{over}$ and $R^{over}$ **significantly reduces** when it is applied on top of a GNN. Exemplary, GCN+LP for $K = 0.5$ drops to $R^{over} = 20.9\%$. Adding LP does **not** decrease test-accuracy (Appendix J.2) and often **increases** adversarial robustness as long as structure matters (see Figure 15 in Appendix J.6.1).

An MLP achieves maximal robustness. Thus, for attacks choosing perturbations independent of the attacked model, it provides an upper bound on the measurable over-robustness for a particular node, as the complete budget $\Delta$ will be exhausted without flipping an MLP's prediction. Exemplary, we measure $43\%$ over-robustness for $K = 0.1$ using $\ell_2$-weak. This means that for a perfectly robust classifier against $\mathcal{B}_{\text{deg}}(\cdot)$ for CSBM graphs with $K = 0.1$, $43\%$ of measured conventional adversarial robustness against $\ell_2$-weak is undesirable over-robustness. Note that all GNNs are **close** to this upper bound. Stronger attacks such as Nettack or GR-BCD still show that a significant part of the measured robustness is over-robustness (see Appendix J.5). Exemplary, Nettack performs strongest against SGC and GCN. However, for a GCN at $K = 0.5$, still $11.4\%$ of the measured robustness is in fact over-robustness. An MLP for Nettack for $K = 2$ still shows $19.2\%$ $R^{over}$, indicating that we can expect a model robust against Nettack in $\mathcal{B}_{\text{deg}}(\cdot)$ to have high $R^{over}$. For bounded perturbation sets $\mathcal{B}_{\Delta}(\cdot)$, maximal over-robustness necessarily decreases if $K$ increases, as the more informative features are, the more structure changes it takes to change the semantic content (i.e., the Bayes decision). Thus, less graphs in $\mathcal{B}_{\Delta}(\cdot)$ have changed semantics (also see Table 2).

As a **positive take-away** for robustness measurements on real-world graphs, we find that if the attack is strong enough, the robustness rankings derived from conventional adversarial robustness are **consistent** with using semantic-aware adversarial robustness (Appendix J.6.1 and J.6.5). However, if the attack is weak, semantic-awareness can **significantly** change robustness rankings (Appendix J.6.2 and J.6.3). For high $K$, some GNNs show high (semantics-aware) adversarial robustness (Appendix J.6.1) additional to high $R^{over}$. Interestingly, if structure matters, the best harmonic mean robustness is in general achieved by MLP+LP (Appendix J.6), which also has the best non-trivial adversarial robustness, while achieving competitive test-accuracies. For Figure 3b, we apply $\ell_2$-weak without budget restriction until it changes a model's prediction. As a result, we find that all GNNs (but not LP) have extensive areas of over-robustness in input space (compare to Figure 5 in Appendix E).

### 5.2.1 OVER-ROBUSTNESS ON REAL-WORLD GRAPHS

Table 2 shows that only a few perturbations can change the semantic-content a graph encodes about a target node. Additionally, if structure matters ($K \leq 3$), the Bayes decision on CSBMs changes on average after at most changing as many edges as the target node's degree. This is visualized in Figure 1 measured for $K = 1.5$ and for other $K$ in Figure 31 in Appendix J.8. It is challenging to derive a reference classifier for real-world datasets and hence, directly measure over-robustness. However, we can investigate the degree-dependent robustness of GNNs and see if we similarly find high robustness beyond the degree of nodes. Figure 4 shows that a majority of test node predictions of a GCN on

(inductive) Cora-ML (Bojchevski & Günnemann, 2018) are robust beyond their degree, by several multiples. The median robustness for degree 1 nodes, lies at over 10 structure changes. Figure 30 in Appendix J.7 shows a similar plot for CSBMs. Results are obtained by applying a variant of $\ell_2$-weak on a target node $v$. However, as Cora-ML is a multi-class dataset, we ensure all inserted edges connect to the same class $c$, which is different to the class of $v$. The vulnerability of a GCN to adversarial attacks likely stems from the lower quartile of robust node classifications in Figure 4. We conjecture *over-robustness* for the upper quartile of highly robust node classifications. In Appendix K we show that combining the GCN with LP significantly reduces the upper extent of its robustness while achieving similar test accuracy and show similar results on other real-world datasets.

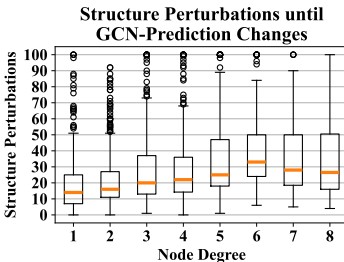

Figure 4: (Inductive) Cora-ML.

## 6 RELATED WORK

We now briefly discuss the most important related work and refer to Appendix L for an extended discussion. Semantics-aware adversarial robustness was theoretically studied by Suggala et al. (2019). Similarly, their work shows no (adversarial) robustness-accuracy tradeoff, postulated among others by Tsipras et al. (2019). However, they only discuss i.i.d. data and focus on the image domain. A notion of overstability was investigated by Jia & Liang (2017) in natural language processing. However, their studied perturbations still preserve the ground-truth answers to predict by models. Tramèr et al. (2020a) discussed over-robustness under the name of invariance-based adversarial examples for the image domain. They did not use the concept of a reference classifier, but used humans as a labeling oracle. To the best of our knowledge, we are the first to study a notion of over-robustness for graphs, and a robustness-accuracy tradeoff for non-i.i.d. data. For graphs, Dai et al. (2018) note the possibility of measuring semantics with a gold standard classifier and generate semantic-preserving perturbations for graph classification on Erdős-Rényi graphs. Although no work has explicitly addressed semantic content preservation for node classification, Zügner et al. (2018) proposed to approximate unnoticeability by preserving the degree distribution; Li et al. (2021) introduced a metric for degree assortativity, but did not restrict perturbations; Chen et al. (2022) proposed a node- and edge-centric homophily metric, but focused on adding malicious nodes instead of edges.

## 7 DISCUSSION AND CONCLUSION

We have shown that common threat models on CSBMs include many graphs with changed semantic content. As a result, (full) conventional robustness leads to sub-optimal generalization and robustness *beyond* the point of semantic change. But we also found that the same threat models include truly adversarial examples. This dichotomy is caused by the low-degrees of nodes and the brittleness of their class-membership to a few edges. Thus, it needs both notions, adversarial and over-robustness for a complete picture of robustness in graph learning. As real-world graphs also contain mainly low-degree nodes, this calls for more caution when applying $\ell_0$-norm restricted threat models. These thread models should not be an end to, but the start of an investigation into realistic perturbation models and works thinking about unnoticeability are positive directions into this endeavour. On CSBMs a significant part of conventional robustness of GNNs is in fact over-robustness (with similar patterns on real-world graphs). This raises the question what kind of robustness do defenses improve on in GNNs?

Applying label propagation on top of GNN predictions has shown to be a simple way to reduce over-robustness while not harming generalization or adversarial robustness. Therefore, LP can be seen as a defense against an attack, where the adversary overtakes a clean node (e.g., social media user) and, with its malicious activity, tries to stay undetected. This shows that not including the known labels in their predictions can be a significant limitation of GNNs. As visually inspecting graphs is difficult, synthetic graph generation models have proved to be an important tool to further a principled understanding of graph attacks and defenses. Concluding, using semantics-aware robustness, we have shown that for inductively classifying a newly sampled node, optimal robustness is achieveable, while maintaining high accuracy.

## ACKNOWLEDGEMENTS

The authors want to thank Bertrand Charpentier, Yan Scholten, Marten Lienen and Nicholas Gao for valuable discussions. Furthermore, Jan Schuchardt, Filippo Guerranti, Johanna Sommer and David Lüdke for feedback to the manuscript. This paper has been supported by the DAAD programme Konrad Zuse Schools of Excellence in Artificial Intelligence, sponsored by the German Federal Ministry of Education and Research, and the German Research Foundation, grant GU 1409/4-1.

## REPRODUCIBILITY STATEMENT

The source code, together will all experiment configuration files of all our experiments can be found on the project page: https://www.cs.cit.tum.de/daml/revisiting-robustness/. Furthermore, we detail our hyperparameter search procedure and all searched through values of all our models and attacks in Appendix H. Details on the parametrization of the used CSBMs can be found in Section 5. We performed all experiments trying to control the randomness as much as possible. We set random seeds and ensure no outlier phenomena by averaging over multiple seeds (including generating multiple CSBM graphs for each CSBM parameterization) as detailed in Section 5 and Appendix H. The experimental setup of our real-world graph experiments is outlined in Appendix H.1.

## ETHICS STATEMENT

Robustness is an important research direction for the reliability of machine learning in real-world applications. A rigorous study counteracts possible exploits by real-world adversaries. We think that the benefits of our work outweigh the risks and see no direct negative implications. However, there remains the possibility of non-benign usage. Specifically, perturbations that are over-robust are likely less noticeable in comparison to prior work. To mitigate this risk and other threats originating from unrobustness, we urge practitioners to assess their model's robustness (at best trying to include domain knowledge to go beyond $\ell_0$-norm perturbations and closer to truly realistic threat models).

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

## A  LIST OF SYMBOLS AND ABBREVIATIONS

| | |
|---|---|
| $[n]$ | Set $\{1, \ldots, n\}$ |
| $\mathcal{A}$ | Adversary |
| $\mathbf{A}$ | Adjacency matrix |
| $\tilde{\mathbf{A}}$ | Perturbed adjacency matrix |
| $\mathbf{A}'$ | Adjacency matrix of a graph with an inductively added node |
| $\tilde{\mathbf{A}}'$ | Perturbed adjacency matrix of a graph with an inductively added node |
| $\mathbf{A}^T$ | Transpose of matrix $\mathbf{A}$ |
| $\mathbf{A}_{i,:}$ | $i$-th row of matrix $\mathbf{A}$ |
| $\mathbf{A}_{i,j}$ | Element of matrix $\mathbf{A}$ in $i$-th row and $j$-th column |
| a.e. | Almost everywhere |
| $\mathcal{B}(\mathbf{X}, \mathbf{A}), \mathcal{B}(\cdot)$ | Perturbation set (possible perturbed graphs for a given clean graph) |
| $\mathcal{B}_\Delta(\cdot)$ | Perturbation set allowing $\Delta \in \mathbb{N}$ edge changes |
| $\mathcal{B}_{deg}(\cdot)$ | Perturbation set $\mathcal{B}_\Delta(\cdot)$ with $\Delta \equiv$ degree of the target node |
| $\mathcal{B}_A(\mathcal{G}, v)$ | Set of adversarial examples for a node $v$ and clean graph $\mathcal{G}$ |

| | |
|---|---|
| $\mathcal{B}_O(\mathcal{G}, v)$ | Set of over-robust examples for a node $v$ and clean graph $\mathcal{G}$ |
| Ber | Bernoulli distribution |
| Bin | Binomial distribution |
| CBA | Contextual Barabási–Albert model with community structure |
| CSBM | Contextual stochastic block model |
| d | Feature dimension |
| $\mathcal{D}_n$ | Data generating distribution |
| $\mathcal{D}(\mathbf{X}, \mathbf{A}, y)$ | Conditional distribution over graphs with an inductively added node |
| $\deg(v)$ | Degree of node $v$ |
| $f$ | Investigated node classifier, e.g. a GNN |
| $f^*$ | Bayes optimal classifier and/or classifier minimizing some loss |
| $f^*_{Bayes}$ | Bayes optimal classifier |
| $F_1^{rob}$ | Harmonic mean between $1 - R^{over}$ and $R^{adv}$ |
| $g$ | Reference node classifier (see Section 3) |
| $\mathcal{G}$ | The graph without labels, i.e. $(\mathbf{X}, \mathbf{A})$ |
| $\tilde{\mathcal{G}}$ | Perturbed graph without labels, i.e. $(\tilde{\mathbf{X}}, \tilde{\mathbf{A}})$ |
| $\mathcal{G}'$ | Sampled graph with an inductively added node without labels |
| $\tilde{\mathcal{G}}'$ | Perturbed graph with an inductively added node without labels |
| $\mathcal{H}$ | Function class, i.e. set of possible considered node classifiers |
| $\mathbb{I}(cond)$ | Indicator function, 1 if $cond$ evaluates to true, otherwise 0 |
| $K$ | Distance between the class means in a CSBM / CBA in $\sigma$ units |
| $\ell_{0/1}$ | 0/1-loss |
| $\ell_p$ | $p$-norm |
| $\mathscr{L}_{\mathcal{G},y}(f)_v$ | Expected 0/1-loss of node $v$ (see Equation 1) |
| $\mathscr{L}_{\mathcal{G},y}^{adv}(f, g)_v$ | Expected adversarial 0/1-loss of node $v$ (see Equation 2) |
| $\mathscr{L}_{\mathcal{G},y}^{over}(f, g)_v$ | Expected over-robust 0/1-loss of node $v$ (see Equation 3) |
| $\mathscr{L}_{\mathcal{G},y}^{rob}(f, g)_v$ | Expected robust loss of node $v$ (see Equation 4) |
| LP | Label propagation |
| $m$ | The degree each newly added node in a CBA should have |
| $\mu$ | Class mean vector in a CSBM / CBA |
| $n$ | Number of nodes |
| $\mathcal{N}(v)$ | Neighbourhood of node $v$ |
| $p$ | Connection probability between same-class nodes in a CSBM |
| $q$ | Connection probability between different-class nodes in a CSBM |
| $R(f, g)$ | Degree-corrected, semantic-aware adversarial robustness of $f$ w.r.t. $g$ |
| $R(f)$ | Degree-corrected classic adversarial robustness of $f$ ($\equiv R(f, y)$). Note: $R(g)$ represents the maximal achievable semantic-aware robustness w.r.t. $g$. |
| $R^{over}$ | Fraction of $R(f)$ not explained by semantic-aware robustness $R(f, g)$ |
| $R^{adv}$ | Fraction of $R(g)$ achieved by semantic-aware robustness $R(f, g)$ |
| $\sigma$ | Variance of the node features in a CSBM / CBA |
| $v$ | A node in the graph. Usually, the target node. |
| $\Omega$ | Semantic-content returning oracle |
| $\omega_{c_1 c_2}$ | Affinity in a CBA between classes $c_1$ and $c_2$ |
| $\mathbf{X}$ | Feature matrix |
| $\tilde{\mathbf{X}}$ | Perturbed feature matrix |
| $\mathbf{X}'$ | Feature matrix of a graph with an inductively added node |
| $\tilde{\mathbf{X}}'$ | Perturbed feature matrix of a graph with an inductively added node |
| $y$ | Complete label vector of a given graph (possibly unknown) |
| $y'$ | Complete label vector of a graph with an inductively added node (possibly unknown) |
| $y_i$ | Label of node $i$ |
| $y_L$ | Known label vector of a given graph |

Table 3: List of most important symbols and abbreviations used in this work.

## B WHY GLOBAL GRAPH PROPERTIES DO NOT PRESERVE SEMANTIC CONTENT

Here we provide formal arguments, why the employed global graph properties and metrics by Zügner et al. (2018), Li et al. (2021) and Chen et al. (2022) do not necessarily preserve or correlate with the semantic content in a graph. In Appendix B.1, we discuss the degree distribution preservation proposed by Zügner et al. (2018). Empirical results showing that preserving degree distribution has no effect on the over-robustness can be found in Appendix I. Appendix B.2 focus on degree assortativity proposed by Li et al. (2021) and Appendix B.3 on other homophily metrics proposed by Chen et al. (2022).

### B.1 DEGREE DISTRIBUTION PRESERVATION

Assume we are given an undirected graph. Choose two arbitrary edges $e_1 = (i, j)$ and $e_2 = (u, v)$ from the set of edges in the graph. Without loss of generality, we assume $i < j$ and $u < v$ and binary classification. Now, we replace these two edges with $(i, v)$ and $(u, j)$. Because $deg(i), deg(j), deg(u)$ and $deg(v)$ are unchanged, the above procedure preserves the degree distribution in the graph.

Assume now that we choose as edges $e_1$ only same-class edges of class $0$ and of $e_2$ only same-class edges of class $1$. By repeating the above procedure for all possible $(e_1, e_2)$-pairs we effectively rewiring a homophilic graph until it becomes heterophilic, while preserving its degree distribution. Given structure is relevant for the semantics, this can't preserve semantics on real-world graphs. For CSBMs and CBAs Theorem 4 and Theorem 5 prove that this rewiring is theoretically most effective to change the semantic content the graph encodes about its nodes. Similar arguments can be made for different edges choices preserving the degree distribution but completely changing the original semantic content, e.g. to change different-class edges to same-class edges or repeating randomly choosing edge-pairs until the original graph structure has been completely destroyed.

Empirical results showing that preserving degree distribution has no effect on the over-robustness can be found in Appendix I. We also want to note that Zügner et al. (2018) requires the graphs to follow a power-law degree distribution. However, not all graphs in practice satisfy this criterion (Clauset et al., 2009).

### B.2 DEGREE ASSORTATIVITY

Degree assortativity, similarly to the degree distribution, can be an orthogonal measure to semantic content preservation. This is as it too only depends on the degree of the nodes and hence, we derive a perturbation scheme preserving the degrees of the incident nodes, provable preserving degree assortativity, but changing the semantic content.

Without loss of generality, we again assume an undirected graph. Then, the degree assortativity coefficient (Li et al., 2021; Newman, 2003) is defined using the so called degree mixing matrix $M$. $M_{i,j}$ represents the fraction of edges, connecting degree $i$ nodes with degree $j$ nodes. For undirected graphs, $M_{i,j}$ is symmetric and satisfies $\sum_{ij} M_{i,j} = 1$ and $\sum_j M_{i,j} = \alpha_i$. Degree assortativtiy is then defined as

$$ r(\alpha) = \frac{\sum_{ij} ij(M_{i,j} - \alpha_i^2)}{\sigma_\alpha^2} \tag{7} $$

where $\sigma_\alpha$ denotes the standard deviation of the degree distribution $\alpha$.

Now, choose two arbitrary edges $e_1 = (v_1, v_2)$ and $e_2 = (u_1, u_2)$ fulfilling $deg(v_1) = deg(u_1)$ and $deg(v_2) = deg(u_2)$. If we replace these two edges with $(v_1, u_2)$ and $(u_1, v_2)$, we preserve the degrees of the original nodes and hence the degree distribution $\alpha$, the degree counts, as well as $M_{i,j}$. Therefore, $r(\alpha)$ stays unchanged. This perturbation scheme can be seen as a special case of the one defined in Appendix B.1. Li et al. (2021) defines as a metric to measure unnoticeability, the expected (normalized) change in degree assortativity by a perturbation scheme, which they call Degree Assortativity Change (DAC). Because the above proposed perturbation scheme does not

change $r(\alpha)$, it has a DAC of 0. However, if $e_1$ represents a same-class edge for class 0 and $e_2$ a same-class edge for class 1, of which there will be many for lower degree connections[5] such as degree 1 nodes with degree 2 nodes or degree 2 nodes with degree 2 nodes, this perturbation scheme significantly rewires the graph by reducing homophily and increasing heterophily. This again is provably most effective perturbations to change the semantic content on CSBMs/CBAs as shown by Theorem 4/Theorem 5. Note that DAC can be non-zero for semantic preserving operations on CSBMs/CBAs, exemplary when $e_1$ and $e_2$ are same-class edges, both of the same class, but connecting nodes of different degrees.

While Li et al. (2021) propose to measure an attacks effect on unnoticeability based on degree assortativity change, it does not propose to restrict the perturbation set through degree assortativity preservation or similar. Furthermore, it is unclear, if one wants to preserve degree assortativity, what sensible values for the preservation would be. All attacks investigated by Li et al. (2021) show small DAC, i.e. on the order of a few percentage points assortativity change at most - while being very effective attacks. For most datasets, exemplary Cora or Pubmed, degree assortativity changes by less than 1%.

### B.3 OTHER HOMOPHILY METRICS

Chen et al. (2022) propose two homophily metrics for graph injection attacks, edge-centric homophily and node-centric homophily. They argue that edge-centric homophily has several shortcomings and focus their attention on node-centric homophily. Therefore, we do the same here. Node-centric homophily of a node $u$ is defined as

$$h_u = sim(r_u, X_u), r_u = \sum_{j \in \mathcal{N}(u)} \frac{1}{\sqrt{d_j d_u}} X_{j,:} \tag{8}$$

where $sim(\cdot)$ is a similarity metric and chosen by (Chen et al., 2022) to be the cosine similarity.

Chen et al. (2022) analyse node-centric homophily changes when adding malicious nodes into the graph (graph injection attacks, GIA) and when adding/removing edges (graph modification attacks, GMA - representing the scenario studied in this work). For adding/removing edges, they similarly to our work, study an inductive evasion setting and use an attack which chooses to add/remove those edges having the largest effect on the attack loss based on the gradient of the relaxed adjacency matrix. Especially, they create perturbed graphs by applying either graph injection or graph modification attacks. They observe that GIA significantly changes the homophily distribution of a graph given by Equation 8, while GMA has only very slight effects on the measured homophily distribution. Especially, GIA produces a significant tail in the homophily distribution not seen in the original graph, which can be avoided by restricting the node-centric homophily change to not surpass certain thresholds. However, GMA does not produce these tails in the homophily distribution and hence, it is not clear how node-centric homophily can be effectively leveraged for preserving semantic content for GMAs. Therefore, node-centric homophily is a highly relevant and interesting metric for GIAs but its applicability to GMAs is limited.

Note that for node-centric homophily, it is not as trivial as for the other graph properties to derive an attack preserving the metric. Especially, node-centric homophily is not a global but a local property and dependent on the continuous node features instead of the discrete node degrees. Therefore, we informally show that our employed $\ell_2$-weak attack, given a limited budget, approximatively preserves the node-centric homophily with high probability for our synthetic graph models, while changing the semantic content of a node. Note that the feature distributions on CSBMs/CBAs, being a Gaußian mixture model, overlap[6]. Without loss of generality, we assume a binary classification setting with node $u$ having class 0. Assuming homophily, Equation 8 will result with a probability greater 0.5 in an average closer to the mean feature vector of class 0 than class 1. If the number of nodes in the graph increases, there will, with high probability, be nodes of class 1 with node features more likely based on class 0, i.e. more closer to the mean feature vector of class 0 than class 1 in feature space. Most nodes have low degree and hence, they change their semantic content

---

[5]This is due to homophily and a majority of nodes being low degree the real-world graph (Figure 8) and synthetic graphs (Figure 6).

[6]Note that otherwise, the graph structure would not be necessary to separate the classes.

by connecting only to a few different class node (see Appendix J.8). If we now employ the $\ell_2$-weak attack, i.e. connect a target node of class 0 to the closest nodes in feature space of class 1, given the graph is large enough, node-centric homophily will not significantly change into the direction of the mean of class 1, as we will - with high probability - find enough nodes closer to the mean of class 0 than 1 to change the semantic content of the target node, before we have to include nodes with features having a large effect on the node-centric homophily. Because semantic content for low-degree nodes often changes after 1 or 2 edge insertions, it can be argued that the node-centric homophily measure for $\ell_2$-weak attack will not change significantly, before we already have measured significant over-robustness, given the graph is large enough.

## C   CONTEXTUAL BARABÁSI–ALBERT MODEL WITH COMMUNITY STRUCTURE

We use the Barabási–Albert model with community structure (Hajek & Sankagiri, 2019; Jordan, 2013) as a (tractable) graph generation model generating a power-law degree distribution and extend it with node features sampled from a Gaußian mixture distribution. In line with the CSBM, we call the resulting model Contextual Barabási–Albert Model with Community Structure (CBA). Compared to a CSBM, it requires the following additional parameters:

- $m$: The degree of each added node (number of edges to insert with each added node).
- $\omega_{y_i y_j}$: The affinity between labels $y_i$ and $y_j$. The affinity must be provided for each possible label pair and we collect the individual terms in the affinity matrix $\omega$. This replaces the parameters $p, q$ in a CSBM.

Furthermore, our initial graph consists of a single node with a self-loop. We implicitly assume that each added node also has one self-loop. Technically, we do not record the self-loops in the adjacency matrix and degree counts but just add them for the sampling probability calculation (see below). Thus, the resulting graph has no self-loops. Sampling from a CBA can now be written as the following iterative process over nodes $i = 2, \ldots, n$:

1. Sample label $y_i \sim \text{Ber}(1/2)$ (Ber denoting the Bernoulli distribution).
2. Sample feature vector $\mathbf{X}_{i,:}|y_i \sim \mathcal{N}((2y_i - 1)\mu, \sigma\mathbf{I})$ with $\mu \in \mathbb{R}^d$, $\sigma \in \mathbb{R}$.
3. Choose $m$ neighbours with probability $p_j^{(i)} = \frac{(1+k_j)\omega_{y_i y_j}}{\sum_{m=1}^{i-1}(1+k_m)\omega_{y_i y_j}}$. In other words, the neighbours are sampled from a multinomial distribution. If a node got sampled more than once, i.e. $\mathbf{A}_{i,j} > 1$, then we set $\mathbf{A}_{i,j} = 1$. Note that this implies $\mathbb{P}[\mathbf{A}_{i,j} = 1|y] = 1 - \mathbb{P}[\mathbf{A}_{i,j} = 0|y] = 1 - \text{Bin}(0|m, p_j^{(i)})$. (Bin denoting the binomial distribution). Furthermore, for all $j \leq i$, we set $\mathbf{A}_{i,j} = \mathbf{A}_{j,i}$.

We denote the above process process $(\mathbf{X}, \mathbf{A}, y) \sim \text{CBA}_{n,m,\omega}^{\mu,\sigma^2}$. Inductively adding $n'$ nodes can be performed by repeating the above process for $i = n + 1, \ldots, n + n'$. Note that a separation into an easy and hard regime for learning made by Fountoulakis et al. (2022) is only based on the fact that the node features are sampled from a Gaußian mixture distribution and does not making use of how the graph structure is generated. Hence, the observation for a CSBM carries to a CBA.

## D   ON THE CHOICES OF GRAPH GENERATION MODELS

There has been a recent trend in using synthetic graph models to generated principled insights into GNNs (see related work in Appendix L). As graph generation model choice, particularly prevalent are the contextual stochastic block model (CSBM) and the degree-corrected stochastic block model (DCSBM). First, we talk about a limitation of CSBMs and we outline why the DCSBM is not applicable to the setting studied in our work. Secondly, we give an argument why we choose a CSBM compared to alternative graph models, in particular preferential attachment models.

The (C)SBM allows to generate graphs with same-class edge probabilities $p$ and different-class edges probabilities $q$, which can be fitted to real-world graphs, such that generated nodes have in

expectation equal (average) same-class degree or different-class degree as the real-world graph and they allow to configure the degree of homophily or heterophily. One shortcoming of stochastic block models is that they do not produce a power-law degree distribution, which is often observed in practice. As a result, Karrer & Newman (2011) introduced a degree-corrected stochastic block model, which allows to fit its expected degree distribution to arbitrary empirical distributions found in real-world graphs. However, to correctly parametrize a DCSBM, one needs knowledge of the expected degrees of all individual nodes in the graph in advance. Then, its theoretic guarantees only hold for a graph of prespecified size and it does not define an iterative growth process. Thereby, its principled applicability is limited to transductive learning[7] not matching our setting of inductive node classification. Classic stochastic block models, including the CSBM, do not suffer from this limitation, as a new node can be added inductively with connection probabilities $p$ and $q$, making CSBMs applicable for our study of inductive node classification.

Stochastic block models have a long history of study in the statistics and machine learning literature (Abbe, 2018) and are the canonical graph generation models to study tasks such as community detection or clustering. This and their extension (CSBM) with node features sampled from a Gaußian mixture model (Binkiewicz et al., 2017; Deshpande et al., 2018), make them an appealing first choice of study for the graph neural network community compared to alternative models outlined below. Hence, (C)SBMs have been used as the model of choice by many previous related work, among others by Fountoulakis et al. (2022); Baranwal et al. (2021) and Palowitch et al. (2022).

A well-known alternative type of graph generation models, which result in a power-law degree distribution, are based on preferential attachment of which the Barabási–Albert (BA) model (Albert & Barabási, 2002) is a widely used instantiation. However, the BA and most of its later variations, do not show community structure making them not applicable to study node classification tasks. Indeed, we are not aware of any works using preferential attachment models to further our understanding of graph neural networks and there are only few works which extent classical preferential attachment models to exhibit community structure. Notably, Hajek & Sankagiri (2019) investigate the properties of a classical BA model implanted with community structure, initially conceived in a more general setting by Jordan (2013). Other works combining preferential attachment with community structure are Li & Maini (2005) and Lee et al. (2015). For our work, we choose the model from Hajek & Sankagiri (2019) and, similarly to the CSBM, extend it with node features sampled from a Gaußian mixture distribution (CBA). We choose this model, as it is based on the classical BA model and compared to the work of Li & Maini (2005), is more concise and principled, with less additionally added parameters following more closely the original BA model, and enjoying a deeper mathematical theory. Furthermore, compared to the work of Lee et al. (2015) it does not require to define a similarity function between individual nodes.

Note that all preferential attachment models with community structure suffer from having to fix the node degree of newly added nodes in advance and that higher degrees are only observed for the earlier nodes in the graph, making newly sampled nodes less representative of the whole graph compared to SBM models. Because of these reasons, we think that using the above preferential attachment models with community structure can provide interesting additional information for the (inductive) study of GNNs, but cannot substitute for using CSBMs.

Furthermore, note that even though a CSBM does not produce a power-law degree distribution, it still fulfills the important property that a majority of nodes have low degree and its degree distribution visually is not so dissimilar to what we find in real-world graph (compare with Figure 12). Furthermore, Clauset et al. (2009) show that not all graphs in the real-world follow a power-law degree distribution.

---

[7]In a DCSBM edges are sampled from a Poison distribution allowing multi-edges in the graphs. This has to be corrected to fit many real-world graphs but in doing so, one violates the exact matching of the expected degree distribution to a real-world graph.

## E  CONCEPTUAL DIFFERENCES BETWEEN OVER- AND ADVERSARIAL ROBUSTNESS

Figure 5a shows the decision boundary of a classifier $f$ following the one of a base classifier $g$ except for the dotted line. The dashed region between $f$'s and $g$'s decision boundary is a region of over-robustness for the blue class and a region of adversarial examples for the red class.

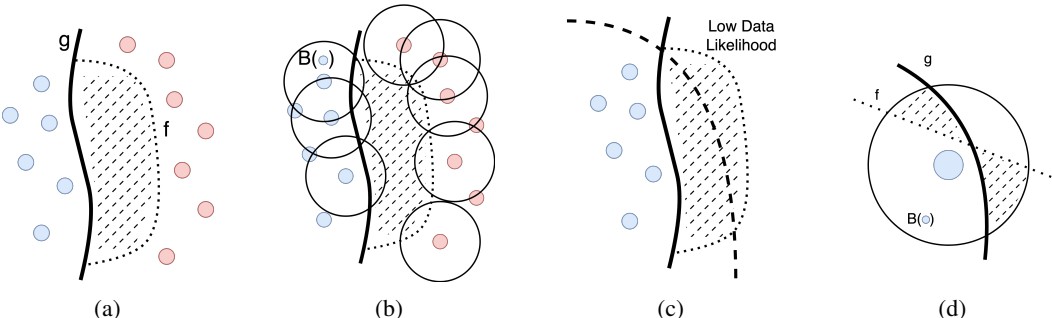

Figure 5: Conceptual differences between over- and adversarial robustness. a) The decision boundary of classifier $f$ follows the one of a base classifier $g$ except for the dotted line. b) Finite perturbation budgets induce bounded perturbation sets $\mathcal{B}(\cdot)$ intersecting only from one side with the dashed area. c) The red class is not seen because it lies in a low data likelihood region. d) Zoomed: A node whose perturbation set includes a region of adversarial and over-robust examples.

Note that in Figure 5b, using the classical concept of adversarial robustness, perturbed examples of blue datapoints crossing the decision boundary of $g$ but still in $\mathcal{B}(\cdot)$ will be judged adversarially robust. Therefore, they may be used to provide a learning signal for $f$ to further solidify its too insensitive decision boundary. Using our refined concept of adversarial robustness, the set of potentially adversarial examples $\mathcal{B}(\cdot)$ is (correctly) cut off at the decision boundary of $g$.

In Figure 5a it is assumed that datapoints on both sides of the decision boundary $g$ have been sampled. This may be likely if every datapoint in the input space has a comparable sampling probability. However, in practice there are regions of high and low data likelihood and hence, it could be that datapoints on one side of the dashed regions have not been sampled as exemplified in Figure 5c). There, the dashed line indicates the transition from a high to low data likelihood area. As a result, only the concept of over-robustness can capture the misbehaviour of the classifier $f$. The reverse scenario is also possible in which only the examples of the right class are sampled and the left class is in a low likelihood region. Indeed, in our results (see Section 5.2), there are some cases where we measure both high adversarial and over-robustness, exactly fitting the scenario visualized in Figure 5c. Note that it makes sense to robustify a classifier even against low-likelihood events as in safety-critical scenarios correct behaviour for unusual or rare events is crucial (Hendrycks et al., 2021).

Figure 5d zooms in to a more intricate case. Disregarding the base classifier $g$ would lead to wrongly interpreting every example above the decision boundary of $f$ as adversarial. With our refined notions, it is possible correctly identify the adversarial region as above $f$ until $g$, the over-robust region as below the decision boundary of $f$ but right of $g$ and the correctly classified area in the top-right.

## F  PROOFS

### F.1  BAYES CLASSIFIER

We proof Theorem 1 by deriving the base classifier for multi-class node classification with $C$ classes. Theorem 1 then follows as special case by setting $C = 2$. We restate Theorem 1 for multiple classes:

**Theorem 1.** *The Bayes optimal classifier, minimizing the expected* $0/1$-*loss (1), is* $f^*(\mathbf{X}', \mathbf{A}', y)_v = \underset{\hat{y} \in \{0, \dots, C-1\}}{argmax} \; \mathbb{P}[y'_v = \hat{y} | \mathbf{X}', \mathbf{A}', y]$.

*Proof.* Lets denote by $x_{-i}$ all elements of a vector $x$ except the $i$-th one. First, note that $\mathcal{D}(\mathbf{X}, \mathbf{A}, y)$ from which we sample $(\mathbf{X}', \mathbf{A}', y')$ defines a conditional joint distribution[8]

$$
\begin{aligned}
\mathbb{P}[\mathbf{X}', \mathbf{A}', y' | \mathbf{X}, \mathbf{A}, y] &= \mathbb{P}[y'_v | \mathbf{X}', \mathbf{A}', y'_{-v}, \mathbf{X}, \mathbf{A}, y] \cdot \mathbb{P}[\mathbf{X}', \mathbf{A}', y'_{-v} | \mathbf{X}, \mathbf{A}, y] \\
&= \mathbb{P}[y'_v | \mathbf{X}', \mathbf{A}', y] \cdot \mathbb{P}[\mathbf{X}', \mathbf{A}', y | \mathbf{X}, \mathbf{A}, y] \\
&= \mathbb{P}[y'_v | \mathbf{X}', \mathbf{A}', y] \cdot \mathbb{P}[\mathbf{X}', \mathbf{A}' | \mathbf{X}, \mathbf{A}, y]
\end{aligned}
\tag{9}
$$

where the first line follows from the basic definition of conditional probability, the second lines from the definition of the inductive sampling scheme (i.e., $y'_{-v} = y$, and $\mathbf{X}'$ and $\mathbf{A}'$ containing $\mathbf{X}$ and $\mathbf{A}$), and the third line from $\mathbb{P}[y | \mathbf{X}, \mathbf{A}, y] = 1$. Thus, we can rewrite the expected loss (1) with respect to these probabilities as

$$
\mathbb{E}_{(\mathbf{X}', \mathbf{A}', y') \sim \mathcal{D}(\mathbf{X}, \mathbf{A}, y)} [\ell_{0/1}(y'_v, f(\mathbf{X}', \mathbf{A}', y)_v]
$$

$$
= \mathbb{E}_{\mathbf{X}', \mathbf{A}' | \mathbf{X}, \mathbf{A}, y} \left[ \mathbb{E}_{y'_v | \mathbf{X}', \mathbf{A}', y} [\ell_{0/1}(y'_v, f(\mathbf{X}', \mathbf{A}', y)_v]] \right]
\tag{10}
$$

$$
= \mathbb{E}_{\mathbf{X}', \mathbf{A}' | \mathbf{X}, \mathbf{A}, y} \left[ \sum_{k=0}^{C-1} \ell_{0/1}(k, f(\mathbf{X}', \mathbf{A}', y)_v) \cdot \mathbb{P}[y'_v = k | \mathbf{X}', \mathbf{A}', y] \right]
\tag{11}
$$

The following argument is adapted from Hastie et al. (2009). Equation (11) is minimal for a classifier $f^*$, if it is (point-wise) minimal for every $(\mathbf{X}', \mathbf{A}', y)$. This means

$$
\begin{aligned}
f^*(\mathbf{X}', \mathbf{A}', y)_v &= \underset{\hat{y} \in \{0, \dots, C-1\}}{argmin} \sum_{k=0}^{C-1} \ell_{0/1}(k, \hat{y}) \cdot \mathbb{P}[y'_v = k | \mathbf{X}', \mathbf{A}', y] \\
&= \underset{\hat{y} \in \{0, \dots, C-1\}}{argmin} \sum_{k=0}^{C-1} (1 - \mathbb{I}[k = \hat{y}]) \cdot \mathbb{P}[y'_v = k | \mathbf{X}', \mathbf{A}', y] \\
&= \underset{\hat{y} \in \{0, \dots, C-1\}}{argmax} \sum_{k=0}^{C-1} \mathbb{I}[k = \hat{y}] \cdot \mathbb{P}[y'_v = k | \mathbf{X}', \mathbf{A}', y] \\
&= \underset{\hat{y} \in \{0, \dots, C-1\}}{argmax} \; \mathbb{P}[y'_v = \hat{y} | \mathbf{X}', \mathbf{A}', y]
\end{aligned}
\tag{12}
\tag{13}
$$

where line (13) follows from fact that in the sum of line (12), there can only be one non-zero term. Equation (13) tells us that the optimal decision is to choose the most likely class. Due to $f^*(\mathbf{X}', \mathbf{A}', y) = \underset{\hat{y} \in \{0,1\}}{argmax} \mathbb{P}[\hat{y} | \mathbf{X}', \mathbf{A}', y]$ minimizing (11) it also minimizes the expected loss (1) and hence, is a Bayes optimal classifier. $\qquad \square$

## F.2   ROBUSTNESS-ACCURACY TRADEOFF

### F.2.1   THEOREM 2

For the proofs of Theorem 2 and Theorem 3, we assume that the set of cases where the two classes are equiprobable has measure zero. This is a mild assumption for instance satisfied in contextual stochastic block models.

For convenience we restate Theorem 2:

**Theorem 2.** *Assume a set of admissible functions $\mathcal{H}$, which includes a Bayes optimal classifier $f^*_{Bayes}$ and let the reference classifier $g$ be itself a Bayes optimal classifier. Then, any minimizer $f^* \in \mathcal{H}$ of (5) is a Bayes optimal classifier.*

---

[8]Note that we actually deal with a discrete-continuous joint distribution.

*Proof.* The proof strategy is mainly adapted from Suggala et al. (2019). Let $f_{Bayes}^* \in \mathcal{H}$ and $g$ be a Bayes optimal classifiers. Assume $f^*$ is a minimizer of (5). Further, assume that $f^*(\mathcal{G}', y)_v$ disagrees with $f_{Bayes}^*(\mathcal{G}', y)_v$ for a set of graphs $\mathcal{G}' \sim \mathcal{D}(\mathcal{G}, y)$ with non-zero measure. Note that a Bayes optimal classifier follows the decision rule $\arg\max_{\hat{y} \in \{0,1\}} \mathbb{P}[y_v' = \hat{y} | \mathcal{G}', y]$ (see Theorem 1). We assume that the set of cases where there are two or more maximal, equiprobable classes has measure zero. Then, the Bayes Decision rule is unique on the support of $\mathcal{D}(\mathcal{G}, y)$. Therefore, $f_{Bayes}^*(\mathcal{G}', y)_v = g(\mathcal{G}', y)_v$ a.e.

We will show that the joint expected loss (5) is strictly larger for $f^*$ than $f_{Bayes}^*$. We start by showing that the standard expected loss $\mathscr{L}_{\mathcal{G},y}(f^*)_v > \mathscr{L}_{\mathcal{G},y}(f_{Bayes}^*)_v$:

$$
\begin{aligned}
&\mathscr{L}_{\mathcal{G},y}(f^*)_v - \mathscr{L}_{\mathcal{G},y}(f_{Bayes}^*)_v \\
&= \mathbb{E}_{(\mathcal{G}',y') \sim \mathcal{D}(\mathcal{G},y)} [\ell_{0/1}(y_v', f^*(\mathcal{G}', y)_v) - \ell_{0/1}(y_v', f_{Bayes}^*(\mathcal{G}', y)_v)] \\
&= \mathbb{E}_{\mathcal{G}'|\mathcal{G},y} \left[ \mathbb{E}_{y_v'|\mathcal{G}',y} [\ell_{0/1}(y_v', f^*(\mathcal{G}', y)_v) - \ell_{0/1}(y_v', f_{Bayes}^*(\mathcal{G}', y)_v)] \right] &(14) \\
&= \mathbb{E}_{\mathcal{G}'|\mathcal{G},y} \left[ \mathbb{E}_{y_v'|\mathcal{G}',y}[\mathbb{I}(y_v' \neq f^*(\mathcal{G}', y)_v)] - \mathbb{E}_{y_v'|\mathcal{G},y}[\mathbb{I}(y_v' \neq f_{Bayes}^*(\mathcal{G}', y)_v)] \right] &(15) \\
&= \mathbb{E}_{\mathcal{G}'|\mathcal{G},y} \left[ \mathbb{P}[y_v' \neq f^*(\mathcal{G}', y)_v|\mathcal{G}', y] - \mathbb{P}[y_v' \neq f_{Bayes}^*(\mathcal{G}', y)_v|\mathcal{G}', y] \right] &(16) \\
&> 0 &(17)
\end{aligned}
$$

Line 14 follows from rewriting the conditional joint distribution defined by $\mathcal{D}(\mathcal{G}, y)$ (see Equation (9)). Line 15 follows from the definition of the $\ell_{0/1}$-loss and the linearity of expectation. The last line 17 follows from the Bayes optimal classifier being defined as a pointwise minimizer of the probability terms in line (16) and the initial assumption of $f^*(\mathcal{G}', y)_v$ disagreeing with $f_{Bayes}^*(\mathcal{G}', y)_v$ for a set of graphs with non-zero measure.

Now, we investigate the expected robust $0/1$-losses for $f^*$ and $f_{Bayes}^*$. Because the decision rule defined by $g$ equals the one of $f_{Bayes}^*$ a.e., it follows that:

$$
\begin{aligned}
&\mathscr{L}_{\mathcal{G},y}^{adv}(f_{Bayes}^*, g)_v \\
&= \mathbb{E}_{(\mathcal{G}',y') \sim \mathcal{D}(\mathcal{G},y)} \left[ \max_{\substack{\tilde{\mathcal{G}}' \in \mathcal{B}(\mathcal{G}') \\ g(\tilde{\mathcal{G}}',y)_v = g(\mathcal{G}',y)_v}} \ell_{0/1}(f(\tilde{\mathcal{G}}', y)_v, g(\mathcal{G}', y)_v) - \ell_{0/1}(f(\mathcal{G}', y)_v, g(\mathcal{G}', y)_v) \right] \\
&= 0
\end{aligned}
$$

and similarly:

$$
\begin{aligned}
&\mathscr{L}_{\mathcal{G},y}^{over}(f_{Bayes}^*, g)_v \\
&= \mathbb{E}_{(\mathcal{G}',y') \sim \mathcal{D}(\mathcal{G},y)} \left[ \max_{\substack{\tilde{\mathcal{G}}' \in \mathcal{B}(\mathcal{G}') \\ f(\tilde{\mathcal{G}},y)_v = f(\mathcal{G}',y)_v}} \ell_{0/1}(g(\tilde{\mathcal{G}}', y)_v, f(\mathcal{G}', y)_v) - \ell_{0/1}(g(\mathcal{G}', y)_v, f(\mathcal{G}', y)_v) \right] \\
&= 0
\end{aligned}
$$

Because the expected adversarial and over-robust $0/1$-losses are non-negative, i.e. $\mathscr{L}_{\mathcal{G},y}^{adv}(f, g)_v \geq 0$ and $\mathscr{L}_{\mathcal{G},y}^{over}(f, g)_v \geq 0$ for any $f \in \mathcal{H}$, it follows that $\mathscr{L}_{\mathcal{G},y}^{rob}(f^*, g)_v \geq \mathscr{L}_{\mathcal{G},y}^{rob}(f_{Bayes}^*, g)_v$. Therefore,

$$
\mathscr{L}_{\mathcal{G},y}(f^*)_v + \lambda \mathscr{L}_{\mathcal{G},y}^{rob}(f^*, g)_v > \mathscr{L}_{\mathcal{G},y}(f_{Bayes}^*, g)_v + \lambda \mathscr{L}_{\mathcal{G},y}^{rob}(f_{Bayes}^*, g)_v
$$

This is a contradiction with $f^*$ being a minimizer of (5). Therefore, $f^*(\mathcal{G}', y)_v$ must equal $f_{Bayes}^*(\mathcal{G}', y)_v$ a.e. and hence, $f^*(\mathcal{G}', y)_v$ is a Bayes optimal classifier.

$\square$

### F.2.2 THEOREM 3

Here we investigate what happens, if we would only optimize for minimal $\mathscr{L}^{rob}_{\mathcal{G},y}(f,g)_v = \lambda_1\mathscr{L}^{adv}_{\mathcal{G},y}(f,g)_v + \lambda_2\mathscr{L}^{over}_{\mathcal{G},y}(f,g)_v$. Will we also find a classifier, which has not only small robust but also small standard loss? Theorem 3 below establishes that this does not hold in general. Thus, being a minimizer for the robust loss $\mathscr{L}^{rob}_{\mathcal{G},y}(f,g)_v$, i.e., achieving optimal robustness (low over- and high adversarial robustness) does not imply achieving minimal generalization error. Theorem 3 showcases that the concepts of over- and adversarial robustness do not interchange with standard accuracy.

**Theorem 3.** *Assume $\mathcal{H}$ is a set of all measurable functions. Let $f^*_{Bayes}$ be a Bayes optimal classifier and let the base classifier $g$ also be a Bayes optimal classifier. Then, there exists a function $f^*_{rob} \in \mathcal{H}$ minimizing the robust loss $\mathscr{L}^{rob}_{\mathcal{G},y}(f^*_{rob}, g)_v$ and satisfying*

$$\mathscr{L}_{\mathcal{G},y}(f^*_{rob})_v > \mathscr{L}_{\mathcal{G},y}(f^*_{Bayes})_v.$$

*Proof.* We assume that the set of cases where the two classes are equiprobable has measure zero. To prove this result, we define a node-classifier $f(\mathcal{G}',y)_v := 1 - f^*_{Bayes}(\mathcal{G}',y)$. We will first show that $f(\mathcal{G}',y)_v$ has minimal expected adversarial $0/1$-loss and later show the same for the expected over-robust $0/1$-loss.

The adversarial loss $\mathscr{L}^{adv}_{\mathcal{G},y}(f,g)_v$ takes the expectation over

$$\max_{\substack{\tilde{\mathcal{G}}'\in\mathcal{B}(\mathcal{G}') \\ g(\tilde{\mathcal{G}}',y)_v=g(\mathcal{G}',y)_v}} \left\{\ell_{0/1}(f(\tilde{\mathcal{G}}',y)_v, g(\mathcal{G}',y)_v) - \ell_{0/1}(f(\mathcal{G}',y)_v, g(\mathcal{G}',y)_v)\right\} \tag{18}$$

Now, note that the second term in equation (18) $\ell_{0/1}(f(\mathcal{G}',y)_v, g(\mathcal{G}',y)_v)$ is 1 by definition of $f$. Furthermore, because we maximize (18) over graphs with $g(\tilde{\mathcal{G}}',y)_v = g(\mathcal{G}',y)_v$, the first term $\ell_{0/1}(f(\tilde{\mathcal{G}}',y)_v, g(\mathcal{G}',y)_v) = \ell_{0/1}(f(\tilde{\mathcal{G}}',y)_v, g(\tilde{\mathcal{G}}',y)_v)$ and hence, is always 1 by definition of $f$. As a result, $\mathscr{L}^{adv}_{\mathcal{G},y}(f,g)_v = 0$. Because $\mathscr{L}^{adv}_{\mathcal{G},y}(f,g)_v$ is non-negative, $f$ achieves minimal adversarial loss.

Now we look at the expected over-robust $0/1$-loss $\mathscr{L}^{over}_{\mathcal{G},y}(f,g)_v$. It takes the expectation over

$$\max_{\substack{\tilde{\mathcal{G}}'\in\mathcal{B}(\mathcal{G}') \\ f(\tilde{\mathcal{G}}',y)_v=f(\mathcal{G}',y)_v}} \left\{\ell_{0/1}(g(\tilde{\mathcal{G}}',y)_v, f(\mathcal{G}',y)_v) - \ell_{0/1}(g(\mathcal{G}',y)_v, f(\mathcal{G}',y)_v)\right\} \tag{19}$$

Here again, the second term $\ell_{0/1}(g(\mathcal{G}',y)_v, f(\mathcal{G}',y)_v) = 1$ as established above. However, because in the set of graphs we are optimizing over $f(\tilde{\mathcal{G}},y)_v = f(\mathcal{G}',y)_v$, it follows that the first term in (19) $\ell_{0/1}(g(\tilde{\mathcal{G}}',y)_v, f(\mathcal{G}',y)_v) = \ell_{0/1}(g(\tilde{\mathcal{G}}',y)_v, f(\tilde{\mathcal{G}}',y)_v)$ and thus, by definition of $f$, is 1. As a result, $\mathscr{L}^{over}_{\mathcal{G},y}(f,g)_v = 0$. Again, because $\mathscr{L}^{over}_{\mathcal{G},y}(f,g)_v$ is non-negative, $f$ achieves minimal over-robust loss.

Therefore, we have established that $f$ achieves optimal robustness, i.e., $\mathscr{L}^{rob}_{\mathcal{G},y}(f,y)_v = 0$.

Now we will show that $\mathscr{L}_{\mathcal{G},y}(f)_v > \mathscr{L}_{\mathcal{G},y}(f^*_{Bayes})_v$:

$$\begin{aligned}
\mathscr{L}_{\mathcal{G},y}&(f)_v - \mathscr{L}_{\mathcal{G},y}(f^*_{Bayes})_v \\
&= \mathop{\mathbb{E}}_{(\mathcal{G}',y')\sim\mathcal{D}(\mathcal{G},y)}[\ell_{0/1}(y'_v, f(\mathcal{G}',y)_v) - \ell_{0/1}(y'_v, f^*_{Bayes}(\mathcal{G}',y)_v)] \\
&= \mathop{\mathbb{E}}_{(\mathcal{G}',y')\sim\mathcal{D}(\mathcal{G},y)}[\ell_{0/1}(y'_v, 1 - f^*_{Bayes}(\mathcal{G}',y)_v) - \ell_{0/1}(y'_v, f^*_{Bayes}(\mathcal{G}',y)_v)] \\
&= \mathbb{E}_{\mathcal{G}'|\mathcal{G},y}\left[\mathbb{E}_{y'_v|\mathcal{G}',y}[\mathbb{I}(y'_v \neq 1 - f^*_{Bayes}(\mathcal{G}',y)_v] - \mathbb{E}_{y'_v|\mathcal{G}',y}[\mathbb{I}(y'_v \neq f^*_{Bayes}(\mathcal{G}',y)_v)]\right] \\
&= \mathbb{E}_{\mathcal{G}'|\mathcal{G},y}\left[\mathbb{E}_{y'_v|\mathcal{G}',y}[\mathbb{I}(y'_v = f^*_{Bayes}(\mathcal{G}',y)_v] - \mathbb{E}_{y'_v|\mathcal{G}',y}[\mathbb{I}(y'_v \neq f^*_{Bayes}(\mathcal{G}',y)_v)]\right] \quad (20) \\
&= \mathbb{E}_{\mathcal{G}'|\mathcal{G},y}\left[\mathbb{P}[y'_v = f^*_{Bayes}(\mathcal{G}',y)_v|\mathcal{G}',y] - \mathbb{P}[y'_v \neq f^*_{Bayes}(\mathcal{G}',y)_v|\mathcal{G}',y]\right] \\
&> 0
\end{aligned}$$

Line (20) follows from the fact of binary classes. The last line follows again from the definition of the Bayes classifier and assuming that the set of cases where the two classes are equiprobable classes has measure zero.

$\square$

### F.3 OPTIMAL ATTACK ON CSBMS

**Theorem 4.** *Given a graph generated by a CSBM. The minimal number of structure changes to change the Bayes classifier (Theorem 1) for a target node $v$ is defined by iteratively: i) connecting $v$ to another node $u$ with $y_v \neq y_u$ or ii) dropping a connection to another node $u$ with $y_v = y_u$.*

*Proof.* Assume $(\mathbf{X}', \mathbf{A}', y') \sim \text{CSBM}_{1,p,q}^{\mu,\sigma^2}(\mathbf{X}, \mathbf{A}, y)$ with $q < p$ (homophily assumption). Recall the Bayes decision $y^* = argmax_{\hat{y} \in \{0,1\}} \; \mathbb{P}[y_v' = \hat{y} | \mathbf{X}', \mathbf{A}', y]$. We want to prove which structure perturbations result in a minimally changed adjacency matrix $\tilde{\mathbf{A}}'$, as measured using the $\ell_0$-norm, but for which

$$y_{new}^* = \underset{\hat{y} \in \{0,1\}}{argmax} \; \mathbb{P}[y_v' = \hat{y} | \mathbf{X}', \tilde{\mathbf{A}}', y] \neq y^* \quad (21)$$

Therefore, we want to change

$$\mathbb{P}[y_v' = y^* | \mathbf{X}', \mathbf{A}', y'] > \mathbb{P}[y_v' = 1 - y^* | \mathbf{X}', \mathbf{A}', y'] \quad (22)$$

to

$$\mathbb{P}[y_v' = y^* | \mathbf{X}', \tilde{\mathbf{A}}', y'] < \mathbb{P}[y_v' = 1 - y^* | \mathbf{X}', \tilde{\mathbf{A}}', y'] \quad (23)$$

To achieve this, first note that we can rewrite Equation (22) using Bayes theorem:

$$\frac{\mathbb{P}[\mathbf{X}_{v,:}', \mathbf{A}_{v,:}' | y_v' = y^*, \mathbf{X}, \mathbf{A}, y] \cdot \mathbb{P}[y_v' = y^* | \mathbf{X}, \mathbf{A}, y]}{\mathbb{P}[\mathbf{X}_{v,:}', \mathbf{A}_{v,:}' | \mathbf{X}, \mathbf{A}, y]}$$
$$> \frac{\mathbb{P}[\mathbf{X}_{v,:}', \mathbf{A}_{v,:}' | y_v' = 1 - y^*, \mathbf{X}, \mathbf{A}, y] \cdot \mathbb{P}[y_v' = 1 - y^* | \mathbf{X}, \mathbf{A}, y]}{\mathbb{P}[\mathbf{X}_{v,:}', \mathbf{A}_{v,:}' | \mathbf{X}, \mathbf{A}, y]} \quad (24)$$
$$\iff \mathbb{P}[\mathbf{X}_{v,:}', \mathbf{A}_{v,:}' | y_v' = y^*, \mathbf{X}, \mathbf{A}, y] > \mathbb{P}[\mathbf{X}_{v,:}', \mathbf{A}_{v,:}' | y_v' = 1 - y^*, \mathbf{X}, \mathbf{A}, y] \quad (25)$$

where in Equation 24 we use $(\mathbf{A}_{v,:}')^T = \mathbf{A}_{:,v}'$ and Equation 25 follows from $\mathbb{P}[y_v' = y^* | \mathbf{X}, \mathbf{A}, y] = \mathbb{P}[y_v' = 1 - y^* | \mathbf{X}, \mathbf{A}, y] = \frac{1}{2}$ by definition of the sampling process. Now, we take the logarithm of both sides in (25) and call the log-difference $\Delta$:

$$\Delta(\mathbf{A}') := \log \mathbb{P}[\mathbf{X}_{v,:}', \mathbf{A}_{v,:}' | y_v' = y^*, \mathbf{X}, \mathbf{A}, y] - \log \mathbb{P}[\mathbf{X}_{v,:}', \mathbf{A}_{v,:}' | y_v' = 1 - y^*, \mathbf{X}, \mathbf{A}, y] \quad (26)$$

Clearly, Equation 25 is equivalent to

$$\Delta(\mathbf{A}') \geq 0 \quad (27)$$

Using the properties of the sampling process of a CSBM (see Section 2), we can rewrite

$$\mathbb{P}[\mathbf{X}_{v,:}', \mathbf{A}_{v,:}' | y_v' = y^*, \mathbf{X}, \mathbf{A}, y] = \mathbb{P}[\mathbf{X}_{v,:}' | y_v' = y^*] \cdot \mathbb{P}[\mathbf{A}_{v,:}' | y_v' = y^*, y] \quad (28)$$
$$= \mathbb{P}[\mathbf{X}_{v,:}' | y_v' = y^*] \cdot \prod_{i \in [n] \setminus \{v\}} \mathbb{P}[\mathbf{A}_{v,i}' | y_v' = y^*, y_i] \quad (29)$$

and therefore

$$\Delta(\mathbf{A}') = \log \frac{\mathbb{P}[\mathbf{X}'_{v,:}|y'_v = y^*]}{\mathbb{P}[\mathbf{X}'_{v,:}|y'_v = 1 - y^*]}$$
$$+ \sum_{i \in [n] \setminus \{v\}} \underbrace{\left(\log \mathbb{P}[\mathbf{A}'_{v,i}|y'_v = y^*, y_i] - \log \mathbb{P}[\mathbf{A}'_{v,i}|y'_v = 1 - y^*, y_i]\right)}_{\Delta_i(\mathbf{A}')} \tag{30}$$

Now, to achieve (23), we want to find those structure perturbations, which lead to $\Delta(\tilde{\mathbf{A}}') < 0$ the fastest (i.e., with least changes). First, note that the first term in Equation 30 does not depend on the adjacency matrix and hence, can be ignored. The second term shows that the change in $\Delta(\mathbf{A}')$ induced by adding or removing an edges $(v, i)$ is additive and independent of adding or removing another edge $(v, j)$. Denote by $\tilde{\mathbf{A}}'(u)$ the adjacency matrix constructed by removing (adding) edge $(v, u)$ from $\mathbf{A}'$ if $(v, u)$ is (not) already in the graph. We define the change potential of node $u$ as $\tilde{\Delta}_u := \Delta_u(\tilde{\mathbf{A}}'(u)) - \Delta_u(\mathbf{A}')$. Then, we only need find those nodes $u$ with maximal change potential $|\tilde{\Delta}_u| = |\Delta_u(\tilde{\mathbf{A}}'(u)) - \Delta_u(\mathbf{A}')|$ and $\tilde{\Delta}_u < 0$ and disconnect (connect) them in decreasing order of $|\tilde{\Delta}_u|$ until $\Delta(\tilde{\mathbf{A}}') < 0$. We will now show that any node $u$ has maximal negative change potential, who either satisfies i) $y_u = y_*$ and $\mathbf{A}'_{v,u} = 1$ or ii) $y_u \neq y_*$ and $\mathbf{A}'_{v,u} = 0$.

To prove this, we make a case distinction on the existence of $(v, u)$ in the unperturbed graph and the class of $y_u$:

**Case $\mathbf{A}'_{v,u} = 0$:**

We distinguish two subcases:

i) $y_u \neq y_*$:

We can write

$$\tilde{\Delta}_u = \log \mathbb{P}[\mathbf{A}'_{v,u} = 1|y'_v = y^*, y_u] - \log \mathbb{P}[\mathbf{A}'_{v,u} = 1|y'_v = 1 - y^*, y_u]$$
$$- \log \mathbb{P}[\mathbf{A}'_{v,u} = 0|y'_v = y^*, y_u] + \log \mathbb{P}[\mathbf{A}'_{v,u} = 0|y'_v = 1 - y^*, y_u]$$
$$= \log q - \log p - \log(1 - q) + \log(1 - p) \tag{31}$$
$$< 0 \tag{32}$$

Equation 31 follows from the sampling process of the CSBM. Equation 32 follows from $q < p$, implying $\log q - \log p < 0$ and $-\log(1 - q) + \log(1 - p) < 0$.

ii) $y_u = y_*$

We can write

$$\tilde{\Delta}_u = \log p - \log q - \log(1 - p) + \log(1 - q) > 0 \tag{33}$$

where the last $>$ follows similarly from $q < p$.

**Case $\mathbf{A}'_{v,u} = 1$:**

i) $y_u \neq y_*$:

We can write

$$\tilde{\Delta}_u = \log \mathbb{P}[\mathbf{A}'_{v,u} = 0|y'_v = y^*, y_u] - \log \mathbb{P}[\mathbf{A}'_{v,u} = 0|y'_v = 1 - y^*, y_u]$$
$$- \log \mathbb{P}[\mathbf{A}'_{v,u} = 1|y'_v = y^*, y_u] + \log \mathbb{P}[\mathbf{A}'_{v,u} = 1|y'_v = 1 - y^*, y_u]$$
$$= \log p - \log q - \log(1 - p) + \log(1 - q) > 0 \tag{34}$$

where Equation 34 follows by the insight, that $\tilde{\Delta}_u$ is the same as for case $\mathbf{A}'_{v,u} = 0$ except multiplied with $-1$.

ii) $y_u = y_*$

We can write

$$\tilde{\Delta}_u = \log q - \log p - \log(1 - q) + \log(1 - p) < 0 \tag{35}$$

where the first equality follows from the insight, that $\tilde{\Delta}_u$ is again the same as for case $\mathbf{A}'_{v,u} = 0$ except multiplied with $-1$. The last $>$ follows again from $q < p$.

The theorem follows from the fact that only the cases where we add an edge to a node of different class, or drop an edge to a node with the same class have negative change potential and the fact, that both cases have the same change potential. □

### F.4 OPTIMAL ATTACK ON CBA

**Theorem 5.** *Given a graph generated by a CBA. The minimal number of structure changes to change the Bayes classifier (Theorem 1) for a target node $v$ is defined by iteratively: i) connecting $v$ to another node $u$ with $y_v \neq y_u$ or ii) dropping a connection to another node $u$ with $y_v = y_u$.*

*Proof.* For this proof, we assume homophily, i.e. the affinity between same class nodes $\omega_{y_u y_v}$, i.e., if $y_u = y_v$ is bigger than between different class nodes, i.e. if $y_u \neq y_v$. We denote the same class affinity as $p$ and the different class affinity as $q$. Now, the proof follows the proof of Theorem 4 until Equation 27. Then, the proof strategy is similar to the one for Theorem 4, but has to be adapted to the specificities of the CBA.

Using the properties of the sampling process of a CBA (see Section C), we can rewrite

$$\mathbb{P}[\mathbf{X}'_{v,:}, \mathbf{A}'_{v,:} | y'_v = y^*, \mathbf{X}, \mathbf{A}, y] = \mathbb{P}[\mathbf{X}'_{v,:} | y'_v = y^*] \cdot \mathbb{P}[\mathbf{A}'_{v,:} | y'_v = y^*, y] \tag{36}$$

$$= \mathbb{P}[\mathbf{X}'_{v,:} | y'_v = y^*] \cdot \prod_{i \in [n] \setminus \{v\}} \mathbb{P}[\mathbf{A}'_{v,i} | y'_v = y^*, y] \tag{37}$$

Note that if $i < v$, $\mathbb{P}[\mathbf{A}'_{v,i} | y'_v = y^*, y] = \mathbb{P}[\mathbf{A}'_{v,i} | y_1, \dots, y'_v = y^*]$ and if $i > v$, $\mathbb{P}[\mathbf{A}'_{v,i} | y'_v = y^*, y] = \mathbb{P}[\mathbf{A}'_{v,i} | y_1, \dots, y'_v = y^*, \dots, y_i]$. In the following, for a more concise presentation, we write $\mathbb{P}[\mathbf{A}'_{v,i} | y'_v = y^*, y]$ in general to include both cases.

Therefore

$$\begin{aligned} \Delta(\mathbf{A}') = & \log \frac{\mathbb{P}[\mathbf{X}'_{v,:} | y'_v = y^*]}{\mathbb{P}[\mathbf{X}'_{v,:} | y'_v = 1 - y^*]} \\ & + \sum_{i \in [n] \setminus \{v\}} \underbrace{\left( \log \mathbb{P}[\mathbf{A}'_{v,i} | y'_v = y^*, y] - \log \mathbb{P}[\mathbf{A}'_{v,i} | y'_v = 1 - y^*, y] \right)}_{\Delta_i(\mathbf{A}')} \end{aligned} \tag{38}$$

Now, to achieve (23), we want to find those structure perturbations, which lead to $\Delta(\tilde{\mathbf{A}}') < 0$ the fastest (i.e., with least changes). First, note that the first term in Equation 30 does not depend on the adjacency matrix and hence, can be ignored. The second term shows that the change in $\Delta(\mathbf{A}')$ induced by adding or removing an edges $(v, i)$ is additive and independent of adding or removing another edge $(v, j)$. Denote by $\tilde{\mathbf{A}}'(u)$ the adjacency matrix constructed by removing (adding) edge $(v, u)$ from $\mathbf{A}'$ if $(v, u)$ is (not) already in the graph. We define the change potential of node $u$ as $\tilde{\Delta}_u := \Delta_u(\tilde{\mathbf{A}}'(u)) - \Delta_u(\mathbf{A}')$. Then, we only need find those nodes $u$ with maximal change potential $|\tilde{\Delta}_u| = |\Delta_u(\tilde{\mathbf{A}}'(u)) - \Delta_u(\mathbf{A}')|$ and $\tilde{\Delta}_u < 0$ and disconnect (connect) them in decreasing order of $|\tilde{\Delta}_u|$ until $\Delta(\tilde{\mathbf{A}}') < 0$. We will now show that any node $u$ has maximal negative change potential, who either satisfies i) $y_u = y_*$ and $\mathbf{A}'_{v,u} = 1$ or ii) $y_u \neq y_*$ and $\mathbf{A}'_{v,u} = 0$.

To prove this, we make a case distinction on the existence of $(v, u)$ in the unperturbed graph and the class of $y_u$. Furthermore, without loss of generality, we assume $u < v$.

**Case $\mathbf{A}'_{v,u} = 0$:**

We distinguish two subcases:

i) $y_u \neq y_*$:

Now, we can write

$$
\begin{aligned}
\tilde{\Delta}_u &= \log \mathbb{P}[\mathbf{A}'_{v,u} = 1 | y'_v = y^*, y] - \log \mathbb{P}[\mathbf{A}'_{v,u} = 1 | y'_v = 1 - y^*, y] \\
&\quad - \log \mathbb{P}[\mathbf{A}'_{v,u} = 0 | y'_v = y^*, y] + \log \mathbb{P}[\mathbf{A}'_{v,u} = 0 | y'_v = 1 - y^*, y] \\
&= \log \mathbb{P}[\mathbf{A}'_{v,u} = 1 | y_1, \ldots, y'_v = y^*] - \log \mathbb{P}[\mathbf{A}'_{v,u} = 1 | y_1, \ldots, y'_v = 1 - y^*] \\
&\quad - \log \mathbb{P}[\mathbf{A}'_{v,u} = 0 | y_1, \ldots, y'_v = y^*] + \log \mathbb{P}[\mathbf{A}'_{v,u} = 0 | y_1, \ldots, y'_v = 1 - y^*] \\
&= \log\left[1 - \mathrm{Bin}\left(0 | m, p_u^{(v)}(y'_v = y^*)\right)\right] - \log\left[1 - \mathrm{Bin}\left(0 | m, p_u^{(v)}(y'_v = 1 - y^*)\right)\right] \\
&\quad - \log\left[\mathrm{Bin}\left(0 | m, p_u^{(v)}(y'_v = y^*)\right)\right] + \log\left[\mathrm{Bin}\left(0 | m, p_u^{(v)}(y'_v = 1 - y^*)\right)\right]
\end{aligned}
\tag{39}
$$

Where the last line follows from the definition of the sampling process of the CBA. $\mathrm{Bin}\left(0 | m, p_u^{(v)}(y'_v = y^*)\right)$ denotes the probability of the event 0 under a binomial distribution with the success probability $p_u^{(v)}(y'_v = y^*)$ depending on the class of $y'_v$; $p_u^{(v)}$ is also dependent on the other classes $y_1, \ldots, y_{v-1}$ - however, these do not change, hence we omit to explicitly mention the dependence.

Now, using the properties of the binomial distribution, we can write

$$
\begin{aligned}
\tilde{\Delta}_u &= \log[p_u^{(v)}(y'_v = y^*)] - \log[p_u^{(v)}(y'_v = 1 - y^*)] \\
&\quad - \log[1 - p_u^{(v)}(y'_v = y^*)] + \log[1 - p_u^{(v)}(y'_v = 1 - y^*)] \\
&= \log\left[\frac{(1 + k_u)q}{\sum_{m=1,m\neq u}^{v-1}(1 + k_m)\omega_{y_v y_m} + (1 + k_u)q}\right] - \log\left[\frac{(1 + k_u)p}{\sum_{m=1,m\neq u}^{v-1}(1 + k_m)\omega_{y_v y_m} + (1 + k_u)p}\right] \\
&\quad + \log\left[\frac{\sum_{m=1,m\neq u}^{v-1}(1 + k_m)\omega_{y_v y_m}}{\sum_{m=1,m\neq u}^{v-1}(1 + k_m)\omega_{y_v y_m} + (1 + k_u)q}\right] - \log\left[\frac{\sum_{m=1,m\neq u}^{v-1}(1 + k_m)\omega_{y_v y_m}}{\sum_{m=1,m\neq u}^{v-1}(1 + k_m)\omega_{y_v y_m} + (1 + k_u)p}\right] \\
&= \log q - \log p \tag{40} \\
&< 0 \tag{41}
\end{aligned}
$$

The second line follows from the definition of the sampling process of the CBS and denoting the same class affinity as $p$ and the different class affinity as $q$. Equation 40 follows from splitting up the division / multiplication in the log-terms and simplifying. The last equation follows from $q < p$.

Note that if $u > v$, the sampling probabilities change to $p_v^{(u)}(y'_v = y^*)$, yielding the same result.

ii) $y_u = y_*$

Now we can write

$$
\begin{aligned}
\tilde{\Delta}_u &= \log \mathbb{P}[\mathbf{A}'_{v,u} = 1 | y'_v = y^*, y] - \log \mathbb{P}[\mathbf{A}'_{v,u} = 1 | y'_v = 1 - y^*, y] \\
&\quad - \log \mathbb{P}[\mathbf{A}'_{v,u} = 0 | y'_v = y^*, y] + \log \mathbb{P}[\mathbf{A}'_{v,u} = 0 | y'_v = 1 - y^*, y] \tag{42} \\
&= \log p - \log q \\
&> 0 \tag{43}
\end{aligned}
$$

the second line follows from recognizing that Equation 42 leads to the exact same equations as case $i)$ except for interchanging $p$ and $q$. The last $>$ follows similarly from $q < p$.

**Case** $\mathbf{A}'_{v,u} = 1$:

i) $y_u \neq y_*$:

We can write

$$
\begin{aligned}
\tilde{\Delta}_u &= \log \mathbb{P}[\mathbf{A}'_{v,u} = 0 | y'_v = y^*, y] - \log \mathbb{P}[\mathbf{A}'_{v,u} = 0 | y'_v = 1 - y^*, y] \\
&\quad - \log \mathbb{P}[\mathbf{A}'_{v,u} = 1 | y'_v = y^*, y] + \log \mathbb{P}[\mathbf{A}'_{v,u} = 1 | y'_v = 1 - y^*, y] \\
&= \log p - \log q > 0
\end{aligned}
\tag{44}
$$

where Equation 44 follows by the insight, that it equals Equation 39 multiplied by $-1$.

ii) $y_u = y_*$

We can write

$$
\begin{aligned}
\tilde{\Delta}_u &= \log \mathbb{P}[\mathbf{A}'_{v,u} = 0 | y'_v = y^*, y] - \log \mathbb{P}[\mathbf{A}'_{v,u} = 0 | y'_v = 1 - y^*, y] \\
&\quad - \log \mathbb{P}[\mathbf{A}'_{v,u} = 1 | y'_v = y^*, y] + \log \mathbb{P}[\mathbf{A}'_{v,u} = 1 | y'_v = 1 - y^*, y] \\
&= \log q - \log p < 0
\end{aligned}
\tag{45}
$$

where the result follows from recognizing that Equation 45 equals Equation 42 except multiplied with $-1$.

The theorem follows from the fact that only the cases where we add an edge to a node of different class, or drop an edge to a node with the same class have negative change potential and the fact, that both cases have the same change potential.

$\square$

## G ROBUSTNESS METRICS

We restate the degree corrected robustness of a classifier $f$ w.r.t. a reference classifier $g$:

$$
R(f, g) = \frac{1}{|V'|} \sum_{v \in V'} \frac{\text{Robustness}(f, g, v)}{\deg(v)}
\tag{46}
$$

Using the true labels $y$ instead of a reference classifier $g$ in (46), one can measure the maximal achievable robustness (before semantic content changes) as $R(g) := R(g, y)$, i.e. the semantic boundary. Note that we can exactly compute $R(g)$ due to knowledge of the data generating process. We measure **adversarial robustness** as the fraction of optimal robustness $R(g)$ achieved: $R^{adv} = R(f, g)/R(g)$. Again, to correctly measure $R^{adv}$, the identical attack is performed to measure $R(f, g)$ and $R(g)$.

A model $f$ can have high adversarial- but also high over-robustness (see Section 3). To have a metric, which truly shows a complete picture of the robustness properties of a model, we take the harmonic mean of $R^{adv}$ and the percentage of how much robustness is legitimate $(1 - R^{over})$ and define an **$F_1$-robustness score**: $F_1^{rob}(\cdot, \cdot) = 2 \frac{(1 - R^{over}) \cdot R^{adv}}{(1 - R^{over}) + R^{adv}}$. Only a model showing perfect adversarial robustness and no over-robustness achieves $F_1^{rob} = 1$.

Note that using $F_\beta^{rob} = (1 + \beta^2) \frac{(1 - R^{over}) \cdot R^{adv}}{\beta^2 (1 - R^{over}) + R^{adv}}$, with $\beta \geq 0$, one can weight the importance of over- versus adversarial examples with $R^{adv}$ being $\beta$-times as important as $(1 - R^{over})$.

## H   EXPERIMENT DETAILS

**Datasets.** We use Contextual Stochastic Block Models (CSBMs) and Contextual Barabási–Albert Model with Community Structure (CBA). The main setup is described in Section 5. The experimental setup for the CBA follows the one for CSBMs outline in Section 5, except for setting $m = 2$, i.e. the number of edges for a newly sampled node is set to two, resulting in similar average node degree to CORA (see Table 4). Furthermore, instead of $p$ and $q$, CBA has affinity terms $\omega_{y_i y_j}$, one for same-class nodes, which we set to the average same-class node degree in CORA (see Table 4) and one for different-class nodes, which we analogously set to the average different-class node degree in CORA.

Table 4 summarizes some dataset statistics and contrasts them with CORA. The reported values are independent of $K$, hence we report the average across 10 sampled CSBM graphs for one $K$.

| **Dataset** | # Nodes | # Edges | # Features | # Classes | Average Node Degree | Average Same-Class Node Degree | Average Different-Class Node Degree |
|---|---|---|---|---|---|---|---|
| CSBM | 1,000 | 1,964±25 | 21 | 2 | 3.93±0.05 | 3.18±0.07 | 0.74±0.04 |
| CBA | 1,000 | 1,968±4 | 21 | 2 | 3.94±0.01 | 3.17±0.03 | 0.77±0.03 |
| CORA | 2,708 | 5,278 | 1,433 | 7 | 3.90 | 3.16 | 0.74 |

Table 4: Dataset statistics

Figure 6a shows that CSBM and CBA graphs mainly contain low-degree nodes.

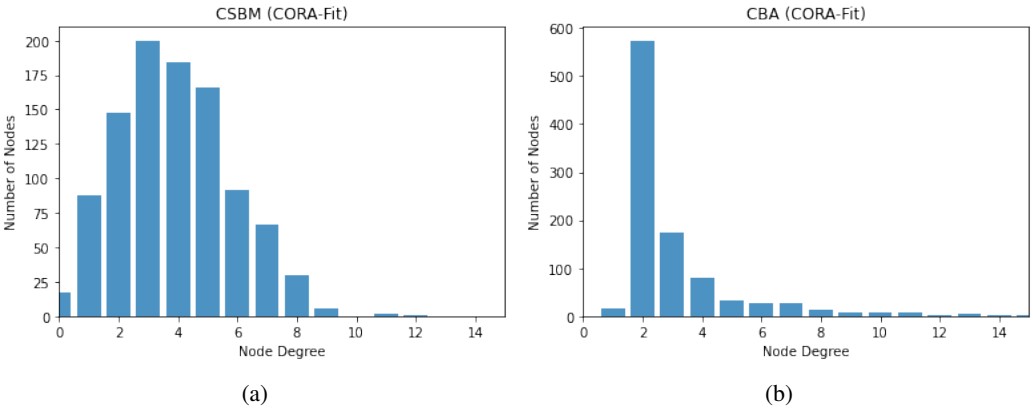

(a)            (b)

Figure 6: (a) Degree distribution of a CSBM graph as parametrized in Section 5 ($n = 1000$). (b) Degree distribution of a CBA parametrized as described in Appendix H.

**Graph Neural Networks and Label Propagation (LP).** We perform extensive hyperparameter search for LP and MLP and each GNN model for each individual $K$ and choose, for each $K$, the on average best performing hyperparamters on 10 graphs sampled from the respective CSBM or CBA. For the *MODEL+LP* variants of our models, we use the individually best performing hyperparameters of the model and LP. Interestingly, we find that very different hyperparameters are optimal for different choices of the feature-information defining parameter $K$. We also find that using the default parameters from the respective model papers successful on the benchmark real-world datasets, don't work well for some choices of $K$, especially low $K$ when structure is very important but features not so.

We train all model for 3000 epochs with a patients of 300 epochs using Adam (Kingma & Ba, 2015) and explore learning rates $[0.1, 0.01, 0.001]$ and weight decay $[0.01, 0.001, 0.001]$ and additionally for

- *MLP:* We use a 1 (Hidden)-Layer MLP and test hidden dimensions $[32, 64, 128, 256]$ and dropout $[0.0, 0.3, 0.5]$. We employ the ReLU activation function.
- *LP:* We use 50 iterations and test $\alpha$ in the range between 0.00 and 1.00 in step sizes of 0.05. LP is the only method having the same hyperparameters on all CSBMs, as it is independent of $K$.

- *SGC:* We explore $[1, 2, 3, 4, 5]$ number of hops and additionally, a learning rate of $0.2$. We investigate dropouts of $[0, 0.3, 0.5]$. SGC was the most challening to train for low $K$.

- *GCN:* We use a two layer (ReLU) GCN with $64$ filters and dropout $[0.0, 0.3, 0.5]$

- *GAT:* We use a two layer GAT with 8 heads and 8 features per head with LeakyReLU having a negative slope of 0.2. We test dropout $[0.0, 0.3, 0.6]$ and neighbourhood dropout $[0.0, 0.3, 0.6]$.

- *GATv2:* We use the best performing hyperparameters of GAT.

- *APPNP:* We use 64 hidden layers, $K = 10$ iterations, dropout $[0.0, 0.3, 0.6]$ and $[0.0, 0.3, 0.5]$ and test $\alpha$ in $[0.05, 0.1, 0.2]$. Interestingly, the higher $K$, we observe higher $\alpha$ performing better.

Further details, such as the best performing hyperparameters, can be found in the released experiment configuration files and source code. Exemplary, the (averaged) validation accuracies on the CSBM graphs can be seen in Figure 7.

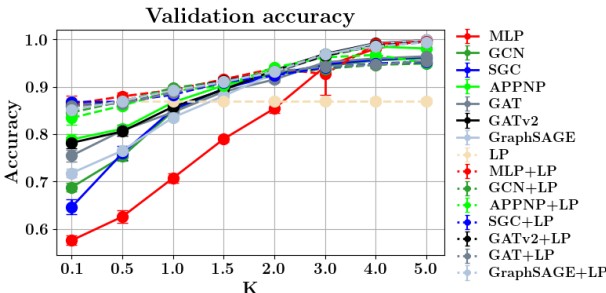

Figure 7: Validation accuracies of the best performing hyperparameters of the different models on CSBMs. Note that GNNs for low $K$ (high-structure relevance) underperform pure LP.

**Attacks.**

- *Nettack* attacks a surrogate SGC model to approximately find maximal adversarial edges. As a surrogate model, instead of using a direct SGC implementation as in the models section, we use a 2-layer GCN with the identity function as non-linearity and $64$ filters and found it trains easier on $K = 0.1$ and hence, provides better adversarial examples for $K = 01$ than direct SGC implementation. For higher $K$, differences are neglectable. We use the same hyperparamter search as outlined for the conventional GCN. For the experiments on CSBMs, we remove the power-law degree distribution test, as a CSBM does not follow a power-law distribution.

- *DICE* randomly disconnects $d$ edges from the test node $v$ to same-class nodes and connects $b$ edges from $v$ to different-class nodes. For a given local budget $\Delta$, we set $d = 0$ and $b = \Delta$.

- *GRBCD* is a $G$reedy gradient-based attack that does not require materializing the dense adjacency matrix due to its use of $R$andomized $B$lock $C$oordinate $D$escent. During the attack, it relaxes the edge weights from $\{0, 1\}$ to $[0, 1]$ and then discretises the result at the end. Thus, we can straight-forwardly attack all models that are differentiable w.r.t. the edge weights (GCN, APPNP). For GAT and GATv2 we softly mask edges based on their weight prior to the attention's softmax activation. Most importantly, since we avoid any surrogate model we can consider GRBCD to be an adaptive attack that crafts model-specific adversarial examples.

- *SGA*, similar to Nettack, attacks a surrogate SGC model. However, it chooses a malicious edge based on the gradient signal obtained from relaxing the adjacency matrix. Furthermore, it restricts the space of possible edges to insert based on the $K$-hop neighbourhood of a target node. As surrogate SGC model, we use the same 2-layer linearized GCN as in Nettack.

- $\ell_2$-*weak* connects a node to its most-similar different class nodes in feature space (using $\ell_2$-norm).

- $\ell_2$-*strong*, in analogy to $\ell_2$-weak, connects a target node to ist most-dissimilar different class nodes in feature space (using $\ell_2$-norm).

Further details, such as the best performing hyperparameters, can be found in the released experiment configuration files and source code.

## H.1   REAL-WORLD GRAPHS

This section covers datasets, models and evaluation procedure for our real-world graph results reported in Section 5.2.1 and Appendix K.

**Datasets.**   To explore potential over-robustness on real-world graphs, the citation networks Cora (Sen et al., 2008), Cora-ML (Bojchevski & Günnemann, 2018), Citeseer (Sen et al., 2008), Pubmed (Sen et al., 2008) and ogbn-arxiv (Hu et al., 2020) are selected. We extract the largest connected component for Cora ML, Citeseer and Pubmed. Table 5 provides an overview over the most important dataset characteristics. Figure 8 visualizes the degree distributions up to degree 25 for ogbn-arxiv and up to degree 15 for all other datasets.

We investigate a *supervised learning setting*, similar to the setting on the CSBM and CBA graphs as well as a *semi-supervised learning setting* more common in practice. In the *supervised learning setting*, for all datasets except ogbn-arxiv, 40 nodes per class are randomly selected as validation and test nodes. The remaining nodes are selected as labeled training set. For ogbn-arxiv, the by Hu et al. (2020) provided temporal split is chosen. Following the inductive approach used in Section 5 for CSBMs, model optimization is performed using the subgraph spanned by all training nodes. Early stopping uses the subgraph spanned by all training and validation nodes based on the validation loss. The *semi-supervised learning setting* follows the commonly employed learning scenario with having access to only a small amount of labeled nodes during training (see e.g. Geisler et al. (2021)), but again splitting inductively. Here, 20 nodes per class are randomly selected as labeled training and validation nodes. Additionally, 40 nodes per class are sampled as test nodes. The remaining nodes are selected as unlabeled training set. Model optimization is again performed inductively using the subgraph spanned by all labeled- and unlabeled training nodes. Early stopping is done w.r.t. all validation nodes and uses the subgraph spanned by the labeled- and unlabeled training and validation nodes.

| Dataset | # Nodes | # Edges | # Features | # Classes | Average Node Degree | Average Same-Class Node Degree | Average Different-Class Node Degree |
|---|---|---|---|---|---|---|---|
| Cora-ML | 2,810 | 15,962 | 2,879 | 7 | 5.68 | 4.46 | 1.22 |
| Cora | 2,708 | 5,278 | 1,433 | 7 | 3.90 | 3.16 | 0.74 |
| Citeseer | 2,110 | 7,336 | 3,703 | 6 | 3.48 | 2.56 | 0.92 |
| Pubmed | 19.717 | 44.324 | 500 | 3 | 4.50 | 3.61 | 0.89 |
| ogbn-arxiv | 169,343 | 1,157,799 | 128 | 40 | 13.67 | 8.95 | 4.73 |

Table 5: Dataset statistics

**Model Architectures.** We evaluate the robustness of Graph Convolutional Networks (GCN), Label Propagation (LP), and GCN followed by LP post-processing (GCN+LP). The GCN architecture and optimization scheme follow Geisler et al. (2021). The GCN has two layers with 64 filters. For ogbn-arxiv three layers with 256 filters are chosen. During training, a dropout of 0.5 is applied. We optimize the model parameters for a maximum of 3000 epochs using Adam (Kingma & Ba, 2015) with learning rate 0.01 and weight decay 0.001. For ogbn-arxiv, no weight decay is applied. LP uses the normalized adjacency as transition matrix and is always performed for ten iterations. This mirrors related architectures like APPNP (Gasteiger et al., 2019). We additionally choose $\alpha = 0.7$ by grid-search over $\{0.1, 0.3, 0.5, 0.7, 0.9\}$ on Cora-ML. For GCN+LP, it should be noted that LP is applied as a post-processing routine at test-time only. It is not included during training.

**Evaluating Degree-Depending Robustness.** We investigate degree-dependent robustness of GNNs by following a similar strategy as the $\ell_2$-weak attack on CSBMs. However, real-world datasets are multi-class. Therefore, we ensure to only connect the target nodes to nodes of one selected class. More concretely, let $v$ be a correctly classified test node with label $c^*$. We investigate for each class $c \neq c^*$, how many edges we can connect to $v$, until the model's prediction changes and denote this number $N_c(v)$. The attack then works as follows: First, we project the high-dimensional feature vectors into lower dimensional space by applying the first weight matrix of the GCN[9]. Then, we iteratively add edges connecting $v$ to the most similar nodes (after projection) in $\ell_2$-norm and evaluate after how many insertions the model's prediction changes. We evaluate for a maximum of 128 edge insertions. We present the results for the class achieving lowest robustness, i.e. smallest

---

[9]For ogbn-arxiv, this projection is omitted as the GCN's filter size is larger than the initial attribute dimension.

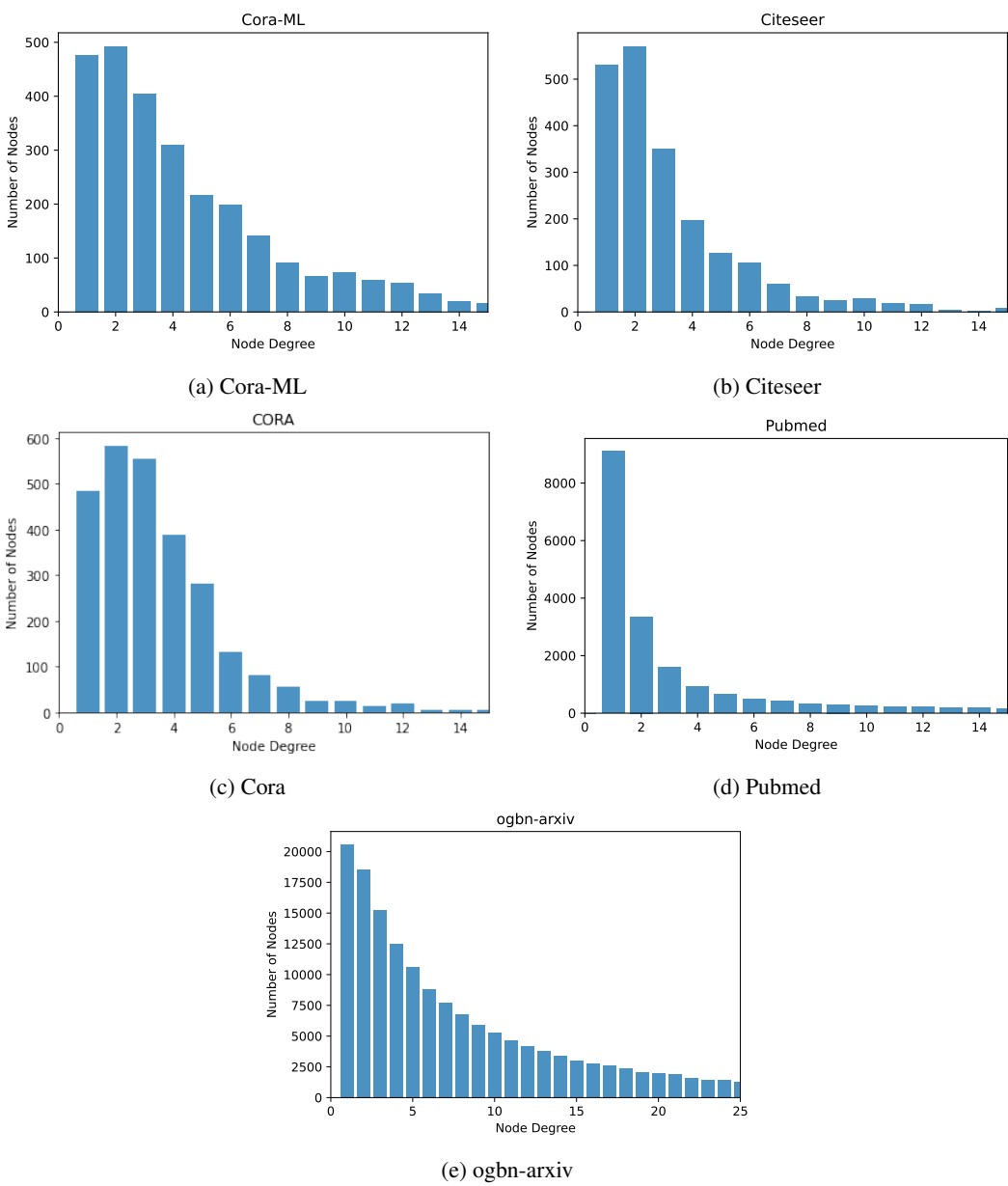

Figure 8: Degree distribution by dataset.

$N_c(v)$, and the results for the class achieving highest robustness, i.e. largest $N_c(v)$. To save on computation, for ogbn-arxiv, not all test nodes are evaluated. Instead, we sample $1500$ nodes from the test set while enforcing the same proportion of classes as in the full test set. Additionally, the above scheme is not applied for every class $c \neq c^*$. We instead compute the average $\ell_2$-distance of each class's closest $3 \cdot \deg(v)$ nodes to the target node and evaluate $N_c(v)$ for the nearest and farthest class.

# I   RESULTS USING CBAS

## I.1   EFFECTS OF DEGREE PRESERVATION ON OVER-ROBUSTNESS

Using CBAs we explore if degree preservation can reduce the measured over-robustness. Note that compared to CSBMs, CBAs degree distribution follows a power-law and hence, Nettacks test for

power-law degree distribution preservation can be employed. For this analysis, we focus on the following models: GCN, APPNP, GAT, LP, MLP as well as their +LP variants. Figure 9 compares the measured over-robustness of Nettack without employing the power-law degree distribution test (left figures), i.e. without degree preservation with employing the power-law degree preserving distribution test (right figures). We find that there is close to **no** difference in the measured over-robustness. Figure 9 explores the perturbation budgets $\mathcal{B}_{deg}(\cdot)$ and $\mathcal{B}_{deg+2}(\cdot)$. The same result, that preserving the degree distribution has close to no effect, can be seen for the other perturbation sets $\mathcal{B}_1(\cdot)$, $\mathcal{B}_2(\cdot)$, $\mathcal{B}_3(\cdot)$ and $\mathcal{B}_4(\cdot)$ in Figure 10. Interestingly, for $K = 0.1$ preserving the degree distribution increases the measured over-robustness for a GCN if $\mathcal{B}_{deg}(\cdot)$ is chosen (Fig. 9a and 9b).

Note that for $K = 0.1$, the measured over-robustness is less than for $K = 0.5$. This can be explained by the fact that for $K = 0.1$ the surrogate SGC model attacked (as well as the all other models) achieve only slightly better performance than random guessing as the learning problem relays heavily on the structure of the graph and only very slightly on the node features. Therefore, Nettack often proposes to add edges which preserve the semantic content but still fool the models, effectively reducing the measured over-robustness. Also note, that the CBA is parametrized using $m = 2$, $\mathcal{B}_{deg}(\cdot)$ equals $\mathcal{B}_2(\cdot)$ and $\mathcal{B}_{deg+2}(\cdot)$ equals $\mathcal{B}_4(\cdot)$ for most target nodes.

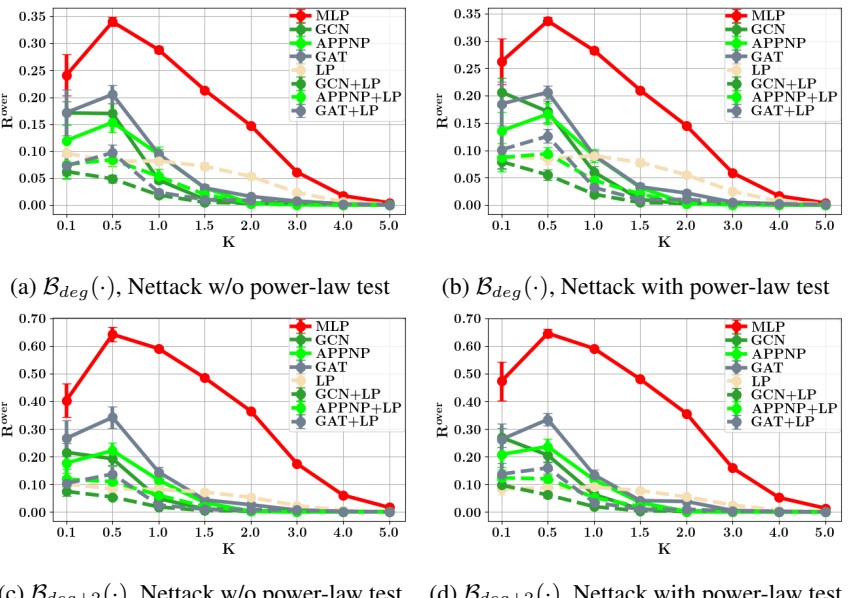

(a) $\mathcal{B}_{deg}(\cdot)$, Nettack w/o power-law test  (b) $\mathcal{B}_{deg}(\cdot)$, Nettack with power-law test

(c) $\mathcal{B}_{deg+2}(\cdot)$, Nettack w/o power-law test  (d) $\mathcal{B}_{deg+2}(\cdot)$, Nettack with power-law test

Figure 9: The measured over-robustness with a perturbation model employing a test for degree preservation compared to the measured over-robustness for a perturbation model not employing a test for degree preservation. Concretely, Nettack with and without power-law degree distribution test for varying local budgets is employed. Degree preservation has close to no effects on the measured over-robustness. Plots with standard error.

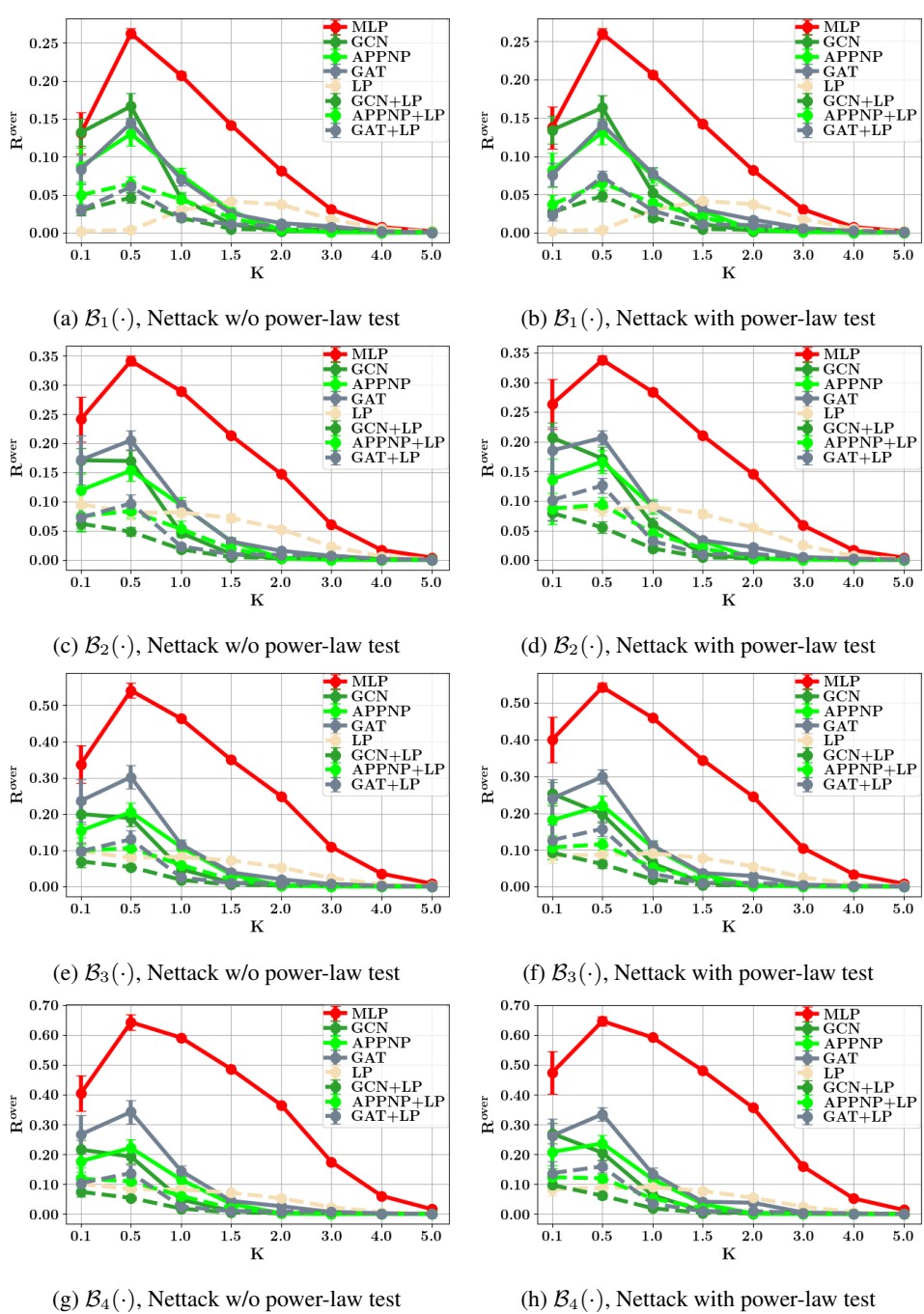

Figure 10: The measured over-robustness with a perturbation model employing a test for degree preservation compared to the measured over-robustness for a perturbation model not employing a test for degree preservation. Concretely, Nettack with and without power-law degree distribution test for varying local budgets is employed. Degree preservation has close to no effects on the measured over-robustness. Plots with standard error.

## I.2 OVER-ROBUSTNESS ON CBAS

Figure 11 shows that the measured over-robustness of the models on CBA graphs is comparable to the measurements on CSBM graphs (see Figure 3a.

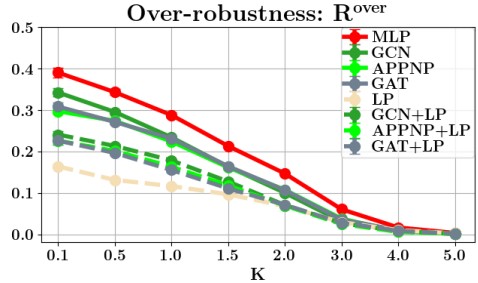

Figure 11: Over-Robustness $R^{over}$ measured using $\ell_2$-weak on CBA graphs with a budget of the degree of the target node. Plot with standard error.

# J    FURTHER RESULTS USING CSBMS

## J.1    DEGREE DISTRIBUTION CSBM VS CORA

Note that degrees in CSBM do **not** follow a power-law distribution. However, they are similar in a different sense to common benchmark citation networks. The goal of this section is to show that the large majority of nodes in both graphs have degrees 2, 3, 4 or 5. Low degree nodes are even more pronounced in CORA than CSBMs.

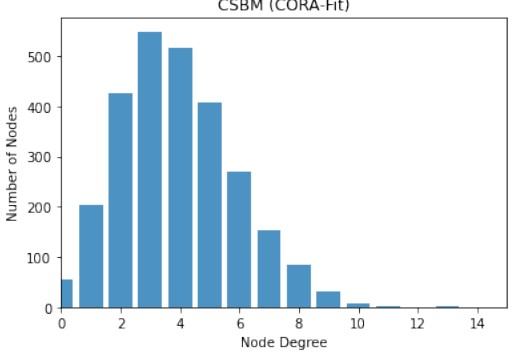 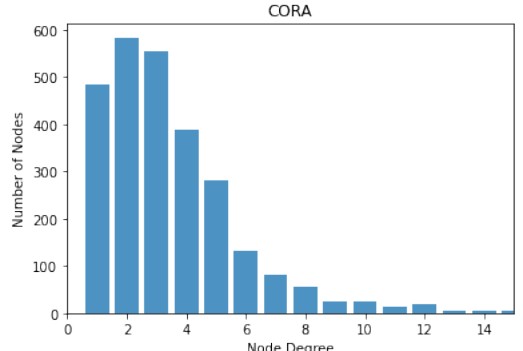

(a) Graph sampled from a CSBM, using $n = 2708$ as CORA. Note that the graph structure is of a CSBM is independent of the $K$ but only dependent on $p, q$ which have been set to fit CORA. Plot cut at node degree 15.

(b) CORA contains mainly low degree nodes. Plot cut at node degree 15.

Figure 12: Degree distribution of the used CSBMs vs CORA, both distribution show that the graphs mainly contain low-degree nodes. This is even more pronounced in CORA than CSBMs.

## J.2    TEST-ACCURACY ON CSBM

This section summaries the detailed performance of the different models on the CSBMs.

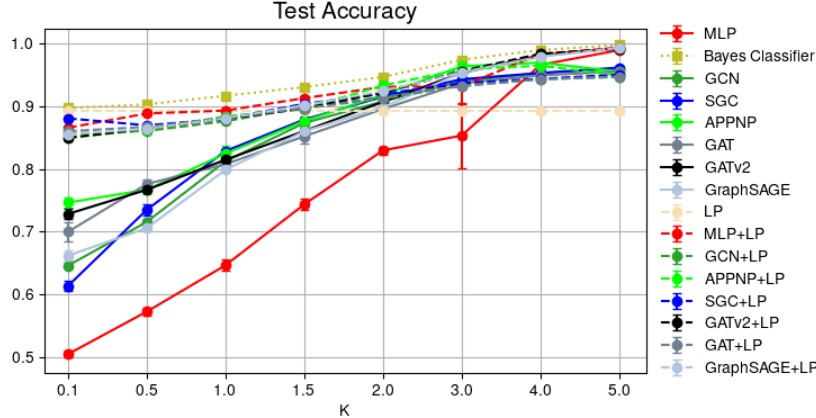

Figure 13: Test accuracy of models on test nodes on the CSBMs.

|  | 0.1 | 0.5 | 1.0 | 1.5 | 2.0 | 3.0 | 4.0 | 5.0 |
|---|---|---|---|---|---|---|---|---|
| Bayes Classifier (BC) | **89.7%** | **90.3%** | **91.7%** | **93.1%** | **94.7%** | **97.4%** | **99.0%** | **99.8%** |
| BC (Features Only) | 50.8% | 59.0% | 68.4% | 76.5% | 83.4% | 92.6% | 97.5% | 99.3% |
| BC (Structure Only) | 89.8% | 89.8% | 89.8% | 89.8% | 89.8% | 89.8% | 89.8% | 89.8% |
| MLP | 50.4% | 57.2% | 64.6% | 74.3% | 83.0% | 85.3% | 96.6% | 99.0% |
| GCN | 64.6% | 71.5% | 81.2% | 87.3% | 90.8% | 94.0% | 95.3% | 96.0% |
| SGC | 61.3% | 73.5% | 82.9% | 87.9% | 91.5% | 94.3% | 95.3% | 96.2% |
| APPNP | 74.6% | 76.7% | 82.4% | 87.7% | 91.8% | **96.4%** | 97.0% | 95.5% |
| GAT | 70.0% | 77.6% | 80.8% | 85.2% | 89.6% | 93.7% | 95.0% | 95.5% |
| GATv2 | 72.8% | 76.7% | 81.5% | 86.1% | 90.7% | 95.3% | 98.1% | 99.3% |
| GraphSAGE | 66.2% | 70.6% | 79.9% | 86.0% | 89.8% | 95.4% | 97.9% | **99.4%** |
| LP | **89.2%** | **89.2%** | 89.2% | 89.2% | 89.2% | 89.2% | 89.2% | 89.2% |
| MLP+LP | 86.6% | 88.8% | **89.3%** | **91.3%** | 93.1% | 93.2% | **98.4%** | 99.3% |
| GCN+LP | 85.6% | 86.1% | 87.6% | 89.8% | 91.5% | 93.2% | 94.3% | 94.7% |
| APPNP+LP | 85.0% | 86.3% | 88.4% | 89.6% | **93.3%** | 95.9% | 96.5% | 95.0% |
| SGC+LP | 88.1% | 87.0% | 88.0% | 90.4% | 92.0% | 93.7% | 94.4% | 95.0% |
| GATv2+LP | 85.0% | 86.4% | 88.0% | 89.6% | 92.2% | 95.6% | **98.4%** | 99.2% |
| GAT+LP | 86.0% | 86.7% | 88.2% | 89.7% | 91.2% | 93.3% | 94.3% | 94.7% |
| GraphSAGE+LP | 85.5% | 86.4% | 88.1% | 90.2% | 92.4% | 95.4% | 97.9% | 99.3% |

Table 6: Average test accuracies of the models on the sampled test nodes on the CSBMs. Standard deviations rarely exceeds 1% and never 2% and hence, is omitted for brevity.

## J.3 EXTENT OF SEMANTIC CONTENT CHANGE IN COMMON PERTURBATION MODELS

The extent of semantic content change looks similar for the other attacks Nettack (Table 7), DICE (Table 8), SGA (Table 9) and $\ell_2$-strong (10) to Table 2 ($\ell_2$-weak).

Values are calculated by attacking an MLP with these respective attacks. As an MLP has perfect robustness, the attacks will exhaust their complete budget in trying to change the MLP's prediction without achieving success. In other words, given any perturbation budget $\Delta$, we construct a maximally perturbed graph with $\Delta$ edge changes and measure, in what fraction of cases, this graph has changed its semantic content. We do not include GR-BCD, as it is not model agnostic and attacking an MLP with GR-BCD, because all gradients w.r.t. the adjacency matrix are zero, is equivalent to random selection of edges, making it similar to but weaker compared to DICE. However, the results in Appendix J.5 indicate that GR-BCD would result in a similar high fraction of graphs with changed semantic content if it would be able to use up its complete budget $\Delta$, as it shows similar over-robustness to the other attacks, when attacking other models than MLP.

**Nettack:**

Furthermore, for $K = 0.1$ the SGC uses by Nettack has only mediocre test-accuracy due to features not being very informative. Therefore, Nettack sometimes proposes to add same-class edges or remove different-class edges.

| Threat Models | $K{=}0.1$ | $K{=}0.5$ | $K{=}1.0$ | $K{=}1.5$ | $K{=}2.0$ | $K{=}3.0$ | $K{=}4.0$ | $K{=}5.0$ |
|---|---|---|---|---|---|---|---|---|
| $\mathcal{B}_1(\cdot)$ | 10.6 | 9.7 | 9.1 | 6.8 | 4.4 | 1.9 | 0.7 | 0.2 |
| $\mathcal{B}_2(\cdot)$ | 23.5 | 24.7 | 24.8 | 19.8 | 14.1 | 6.2 | 2.2 | 0.7 |
| $\mathcal{B}_3(\cdot)$ | 35.7 | 41.0 | 43.6 | 38.1 | 28.8 | 14.2 | 4.9 | 1.6 |
| $\mathcal{B}_4(\cdot)$ | 47.3 | 55.0 | 61.2 | 57.5 | 46.8 | 25.1 | 9.2 | 3.2 |
| $\mathcal{B}_{\text{deg}}(\cdot)$ | 45.4 | 42.9 | 50.0 | 47.6 | 39.4 | 21.9 | 8.9 | 3.1 |
| $\mathcal{B}_{\text{deg+2}}(\cdot)$ | 63.9 | 73.7 | 90.1 | 89.8 | 79.6 | 50.2 | 23.6 | 8.6 |

Table 7: Percentage (%) of nodes for which we find perturbed graphs in $\mathcal{B}_\Delta(\cdot)$ violating semantic content preservation, i.e. with changed ground truth labels. Calculated by adding or dropping $\Delta$ edges suggested by Nettack to every target node. Note that for $K{=}4.0$ and $K{=}5.0$ structure is not necessary for good generalization (Table 1). Standard deviations are insignificant and hence, omitted.

**DICE:**

| Threat Models | $K{=}0.1$ | $K{=}0.5$ | $K{=}1.0$ | $K{=}1.5$ | $K{=}2.0$ | $K{=}3.0$ | $K{=}4.0$ | $K{=}5.0$ |
|---|---|---|---|---|---|---|---|---|
| $\mathcal{B}_1(\cdot)$ | 14.9 | 10.9 | 9.0 | 6.2 | 4.3 | 2.1 | 0.6 | 0.2 |
| $\mathcal{B}_2(\cdot)$ | 36.8 | 31.4 | 26.2 | 19.7 | 13.8 | 6.4 | 2.0 | 0.6 |
| $\mathcal{B}_3(\cdot)$ | 58.6 | 52.9 | 46.4 | 37.7 | 28.8 | 13.9 | 5.0 | 1.3 |
| $\mathcal{B}_4(\cdot)$ | 77.1 | 72.6 | 66.6 | 57.0 | 45.8 | 25.8 | 9.6 | 2.9 |
| $\mathcal{B}_{\text{deg}}(\cdot)$ | 76.6 | 58.9 | 55.6 | 48.9 | 38.5 | 22.1 | 8.7 | 2.8 |
| $\mathcal{B}_{\text{deg+2}}(\cdot)$ | 100.0 | 100.0 | 99.2 | 92.2 | 79.9 | 50.8 | 24.0 | 8.2 |

Table 8: Percentage (%) of nodes for which we find perturbed graphs in $\mathcal{B}_\Delta(\cdot)$ violating semantic content preservation, i.e. with changed ground truth labels. Calculated by randomly connecting $\Delta$ different-class nodes (DICE) to every target node. Note that for $K{=}4.0$ and $K{=}5.0$ structure is not necessary for good generalization (Table 1). Standard deviations are insignificant and hence, omitted.

**SGA:**

| Threat Models | $K{=}0.1$ | $K{=}0.5$ | $K{=}1.0$ | $K{=}1.5$ | $K{=}2.0$ | $K{=}3.0$ | $K{=}4.0$ | $K{=}5.0$ |
|---|---|---|---|---|---|---|---|---|
| $\mathcal{B}_1(\cdot)$ | 14.2 | 11.3 | 9.3 | 6.6 | 4.6 | 1.7 | 0.8 | 0.2 |
| $\mathcal{B}_2(\cdot)$ | 36.1 | 32.5 | 26.1 | 19.8 | 14.2 | 5.6 | 2.2 | 0.6 |
| $\mathcal{B}_3(\cdot)$ | 59.3 | 54.2 | 47.2 | 38.2 | 28.8 | 13.5 | 5.0 | 1.7 |
| $\mathcal{B}_4(\cdot)$ | 76.4 | 73.5 | 66.8 | 58.1 | 46.8 | 24.3 | 9.6 | 3.4 |
| $\mathcal{B}_{\text{deg}}(\cdot)$ | 73.6 | 61.7 | 56.2 | 49.0 | 40.0 | 20.9 | 8.9 | 3.2 |
| $\mathcal{B}_{\text{deg+2}}(\cdot)$ | 100.0 | 99.9 | 99.3 | 92.8 | 80.0 | 50.2 | 24.6 | 9.1 |

Table 9: Percentage (%) of nodes for which we find perturbed graphs in $\mathcal{B}_\Delta(\cdot)$ violating semantic content preservation, i.e. with changed ground truth labels. Calculated by randomly connecting $\Delta$ different-class nodes (SGA) to every target node. Note that for $K{=}4.0$ and $K{=}5.0$ structure is not necessary for good generalization (Table 1). Standard deviations are insignificant and hence, omitted.

$\ell_2$-**strong:**

| Threat Models | $K$=0.1 | $K$=0.5 | $K$=1.0 | $K$=1.5 | $K$=2.0 | $K$=3.0 | $K$=4.0 | $K$=5.0 |
|---|---|---|---|---|---|---|---|---|
| $\mathcal{B}_1(\cdot)$ | 14.5 | 11.1 | 9.4 | 6.7 | 4.4 | 1.9 | 0.8 | 0.2 |
| $\mathcal{B}_2(\cdot)$ | 35.9 | 30.9 | 25.8 | 19.8 | 14.1 | 6.2 | 2.2 | 0.7 |
| $\mathcal{B}_3(\cdot)$ | 58.6 | 53.7 | 46.9 | 38.1 | 28.8 | 14.5 | 5.0 | 1.7 |
| $\mathcal{B}_4(\cdot)$ | 76.5 | 73.0 | 66.7 | 58.0 | 47.0 | 25.9 | 9.7 | 3.4 |
| $\mathcal{B}_{\text{deg}}(\cdot)$ | 77.3 | 59.6 | 55.9 | 49.0 | 39.6 | 22.0 | 8.9 | 3.2 |
| $\mathcal{B}_{\text{deg}+2}(\cdot)$ | 100.0 | 100.0 | 99.4 | 92.9 | 80.5 | 51.8 | 24.7 | 9.1 |

Table 10: Percentage (%) of nodes for which we find perturbed graphs in $\mathcal{B}_\Delta(\cdot)$ violating semantic content preservation, i.e. with changed ground truth labels. Calculated by randomly connecting $\Delta$ different-class nodes ($\ell_2$-strong) to every target node. Note that for $K$=4.0 and $K$=5.0 structure is not necessary for good generalization (Table 1). Standard deviations are insignificant and hence, omitted.

## J.4 OVER-ROBUSTNESS RESULTS FOR OTHER PERTURBATION SETS

Figure 14 shows the measured over-robustness using $\ell_2$-weak for the perturbation sets $\mathcal{B}_1(\cdot)$, $\mathcal{B}_2(\cdot)$, $\mathcal{B}_3(\cdot)$, $\mathcal{B}_4(\cdot)$ and $\mathcal{B}_{deg+2}(\cdot)$. For completeness, $\mathcal{B}_{deg}(\cdot)$ is include in Figure 14. All perturbation sets show significant over-robustness.

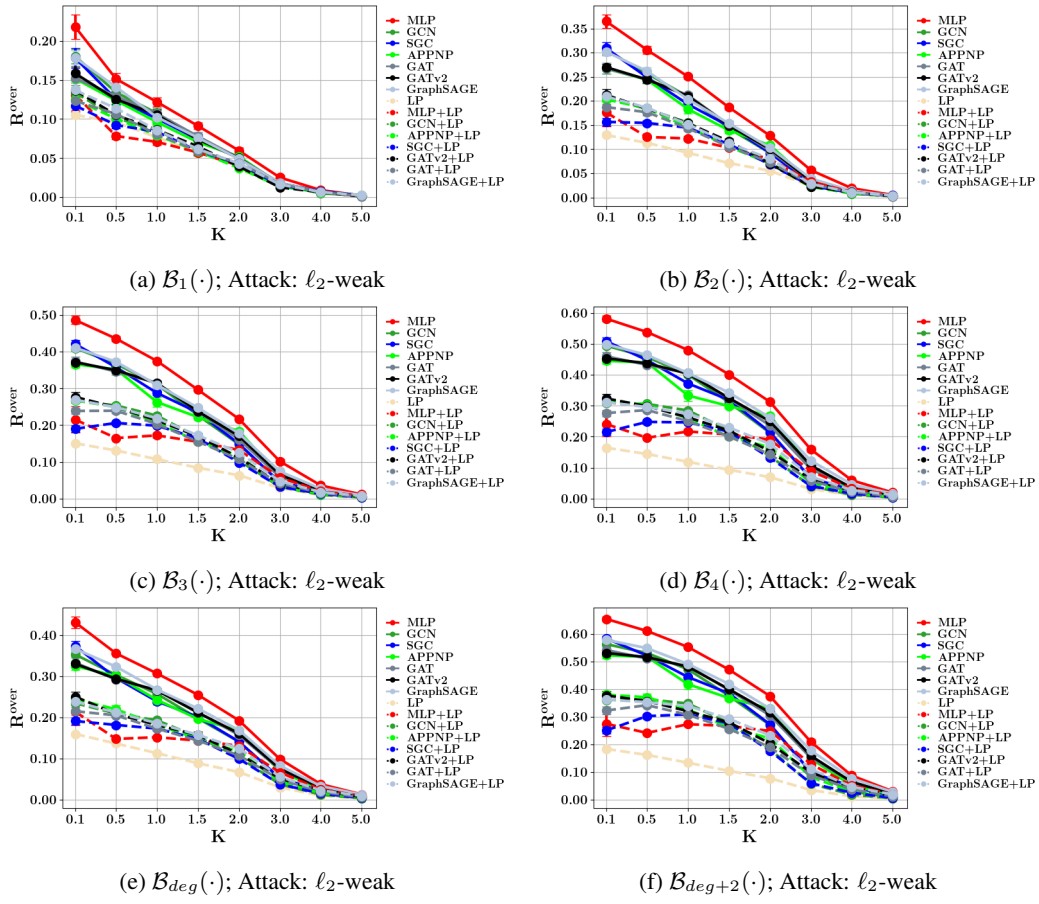

(a) $\mathcal{B}_1(\cdot)$; Attack: $\ell_2$-weak

(b) $\mathcal{B}_2(\cdot)$; Attack: $\ell_2$-weak

(c) $\mathcal{B}_3(\cdot)$; Attack: $\ell_2$-weak

(d) $\mathcal{B}_4(\cdot)$; Attack: $\ell_2$-weak

(e) $\mathcal{B}_{deg}(\cdot)$; Attack: $\ell_2$-weak

(f) $\mathcal{B}_{deg+2}(\cdot)$; Attack: $\ell_2$-weak

Figure 14: Over-robustness $R^{over}$, i.e. the fraction of robustness beyond semantic change for different perturbation sets $\mathcal{B}_\Delta(\cdot)$. (Attack: $\ell_2$-weak; Plots with Standard Error). A significant part of the measured robustness for every perturbation set can be attributed to over-robustness. The larger $\Delta$, the larger the share of over-robustness.

## J.5 Detailed Over-Robustness Results for $\mathcal{B}_{deg}(\cdot)$

$\ell_2$-**weak**:

| K | 0.1 | 0.5 | 1.0 | 1.5 | 2.0 | 3.0 | 4.0 | 5.0 |
|---|-----|-----|-----|-----|-----|-----|-----|-----|
| MLP | 43.1% | 35.6% | 30.7% | 25.5% | 19.3% | 9.9% | 3.8% | 1.3% |
| GCN | 35.3% | 30.3% | 25.7% | 19.8% | 14.1% | 5.2% | 1.6% | 0.4% |
| SGC | 37.4% | 29.6% | 23.9% | 20.1% | 14.1% | 5.2% | 2.3% | 0.6% |
| APPNP | 32.5% | 30.1% | 24.3% | 19.6% | 16.8% | 7.0% | 2.4% | 0.9% |
| GAT | 33.3% | 29.2% | 26.6% | 21.2% | 15.7% | 7.3% | 2.9% | 0.9% |
| GATv2 | 33.0% | 29.4% | 26.6% | 21.2% | 16.1% | 7.5% | 2.9% | 0.9% |
| GraphSAGE | 36.7% | 32.3% | 26.8% | 22.2% | 16.9% | 8.3% | 3.3% | 1.2% |
| LP | 16.0% | 13.7% | 11.3% | 8.9% | 6.7% | 3.0% | 1.0% | 0.3% |
| MLP+LP | 21.4% | 14.8% | 15.2% | 14.4% | 13.1% | 6.6% | 2.5% | 0.9% |
| GCN+LP | 23.5% | 20.9% | 19.3% | 15.2% | 10.3% | 3.7% | 1.2% | 0.3% |
| APPNP+LP | 24.5% | 22.0% | 18.2% | 14.8% | 11.7% | 4.4% | 1.6% | 0.5% |
| SGC+LP | 19.1% | 18.2% | 17.3% | 14.8% | 9.9% | 3.8% | 1.5% | 0.4% |
| GATv2+LP | 25.1% | 20.8% | 18.2% | 15.7% | 11.1% | 5.1% | 2.2% | 0.6% |
| GAT+LP | 21.5% | 20.5% | 17.3% | 14.4% | 10.8% | 4.9% | 1.9% | 0.6% |
| GraphSAGE+LP | 23.9% | 21.2% | 18.6% | 15.8% | 12.5% | 5.6% | 2.3% | 0.9% |

Table 11: Over-Robustness $R^{over}$ measured using $\ell_2$-weak (see also Figure 3a with a budget of the degree of the target node. Standard deviations never exceed 1% except for MLP+LP at $K = 0.1$ which has a standard deviation of 3%.

**Nettack**:

Table 12 shows that over-robustness is not only occurring for weak attacks. Especially, the MLP results show that if we would have a classifier perfectly robust against Nettack in the bounded perturbation set, for all $K \leq 3$ (where structure matters), this would result in high over-robustness.

| | 0.1 | 0.5 | 1.0 | 1.5 | 2.0 | 3.0 | 4.0 | 5.0 |
|---|-----|-----|-----|-----|-----|-----|-----|-----|
| MLP | 26.5% | 26.4% | 28.4% | 25.0% | 19.2% | 9.9% | 3.8% | 1.3% |
| GCN | 13.5% | 11.4% | 6.0% | 2.4% | 0.8% | 0.1% | 0.0% | 0.0% |
| SGC | 7.7% | 4.6% | 2.9% | 1.4% | 0.4% | 0.1% | 0.1% | 0.0% |
| APPNP | 11.9% | 10.1% | 9.0% | 3.8% | 1.4% | 0.1% | 0.6% | 0.6% |
| GAT | 9.7% | 15.1% | 12.7% | 11.0% | 5.4% | 3.3% | 1.6% | 0.5% |
| GATv2 | 14.0% | 12.4% | 13.3% | 13.8% | 8.8% | 5.1% | 2.7% | 0.8% |
| GraphSAGE | 14.6% | 17.7% | 15.7% | 12.3% | 9.9% | 4.8% | 2.1% | 0.7% |
| LP | 5.4% | 5.4% | 5.6% | 5.4% | 4.2% | 1.9% | 0.6% | 0.2% |
| MLP+LP | 9.4% | 6.9% | 10.2% | 9.6% | 10.1% | 5.0% | 1.7% | 0.6% |
| GCN+LP | 4.5% | 3.6% | 2.0% | 1.1% | 0.5% | 0.2% | 0.1% | 0.1% |
| APPNP+LP | 8.5% | 5.6% | 3.1% | 2.1% | 0.3% | 0.1% | 0.4% | 0.3% |
| SGC+LP | 1.0% | 1.1% | 1.1% | 0.5% | 0.2% | 0.1% | 0.1% | 0.0% |
| GATv2+LP | 8.9% | 6.3% | 6.3% | 5.2% | 5.3% | 3.1% | 1.7% | 0.5% |
| GAT+LP | 6.5% | 6.8% | 4.4% | 3.7% | 2.3% | 1.3% | 0.7% | 0.3% |
| GraphSAGE+LP | 7.7% | 7.4% | 7.1% | 4.8% | 4.1% | 1.7% | 0.8% | 0.4% |

Table 12: Over-Robustness $R^{over}$ measured using Nettack with a budget of the degree of the target node. Standard deviations rarely exceed 1% notably for MLP at $K = 0.1$ with 4.8% and MLP+LP at $K = 0.1$ at 3%.

**DICE**:

Table 13 shows, as for Nettack, that over-robustness is not only occurring for weak attacks. Especially, the MLP results show that if we would have a classifier perfectly robust against DICE in the bounded perturbation set, for all $K \leq 3$ (where structure matters), this would result in high over-robustness.

| | 0.1 | 0.5 | 1.0 | 1.5 | 2.0 | 3.0 | 4.0 | 5.0 |
|---|---|---|---|---|---|---|---|---|
| MLP | 43.6% | 35.0% | 30.8% | 25.0% | 18.8% | 10.0% | 3.5% | 1.1% |
| GCN | 34.0% | 29.2% | 23.2% | 16.8% | 11.1% | 3.4% | 0.9% | 0.3% |
| SGC | 37.2% | 28.9% | 21.3% | 16.5% | 10.7% | 3.3% | 0.9% | 0.2% |
| APPNP | 31.1% | 27.8% | 22.7% | 17.2% | 13.6% | 4.3% | 1.7% | 0.8% |
| GAT | 32.9% | 27.9% | 25.7% | 19.2% | 13.1% | 5.7% | 2.2% | 0.7% |
| GATv2 | 31.2% | 27.8% | 23.6% | 18.7% | 13.5% | 6.6% | 2.4% | 0.8% |
| GraphSAGE | 34.7% | 29.9% | 24.7% | 19.4% | 14.6% | 7.0% | 2.6% | 0.9% |
| LP | 16.0% | 13.4% | 10.8% | 8.7% | 6.4% | 2.7% | 0.9% | 0.3% |
| MLP+LP | 20.8% | 14.7% | 15.3% | 14.5% | 12.9% | 6.9% | 2.4% | 0.8% |
| GCN+LP | 21.9% | 20.0% | 17.0% | 12.5% | 8.2% | 2.5% | 0.8% | 0.3% |
| APPNP+LP | 23.4% | 20.1% | 15.9% | 13.2% | 8.7% | 2.7% | 1.1% | 0.5% |
| SGC+LP | 18.5% | 17.6% | 15.3% | 11.9% | 7.6% | 2.6% | 0.8% | 0.2% |
| GATv2+LP | 22.7% | 19.5% | 15.7% | 13.3% | 8.7% | 4.6% | 1.8% | 0.6% |
| GAT+LP | 21.9% | 18.9% | 15.6% | 12.3% | 8.9% | 3.5% | 1.5% | 0.5% |
| GraphSAGE+LP | 22.8% | 19.2% | 16.0% | 12.4% | 9.5% | 4.2% | 1.7% | 0.7% |

Table 13: Over-Robustness $R^{over}$ measured using DICE with a budget of the degree of the target node. Standard deviations are insignificant and removed for brevity.

**SGA**:

Table 14 shows, as for Nettack, that over-robustness is not only occurring for weak attacks. Especially, the MLP results show that if we would have a classifier perfectly robust against SGA in the bounded perturbation set, for all $K \leq 3$ (where structure matters), this would result in high over-robustness.

| | 0.1 | 0.5 | 1.0 | 1.5 | 2.0 | 3.0 | 4.0 | 5.0 |
|---|---|---|---|---|---|---|---|---|
| MLP | 42.3% | 36.4% | 31.2% | 25.4% | 19.5% | 9.2% | 3.8% | 1.3% |
| GCN | 23.4% | 16.9% | 7.3% | 3.2% | 1.1% | 0.1% | 0.1% | 0.0% |
| SGC | 20.0% | 8.4% | 4.3% | 1.8% | 0.4% | 0.0% | 0.1% | 0.0% |
| APPNP | 25.1% | 18.5% | 10.9% | 5.9% | 3.5% | 0.6% | 0.1% | 0.0% |
| GAT | 31.2% | 26.6% | 18.4% | 16.7% | 9.4% | 4.5% | 2.1% | 0.9% |
| GATv2 | 33.8% | 25.0% | 18.8% | 16.9% | 11.4% | 6.8% | 2.8% | 0.8% |
| GraphSAGE | 34.1% | 28.5% | 22.5% | 17.8% | 12.8% | 6.3% | 2.4% | 0.9% |
| LP | 8.3% | 7.7% | 6.3% | 5.6% | 4.0% | 1.9% | 0.6% | 0.2% |
| MLP+LP | 11.2% | 9.9% | 10.1% | 9.3% | 12.2% | 4.6% | 1.9% | 0.6% |
| GCN+LP | 8.7% | 6.2% | 2.9% | 1.3% | 0.6% | 0.1% | 0.1% | 0.0% |
| APPNP+LP | 16.5% | 12.4% | 7.0% | 3.5% | 1.7% | 0.4% | 0.1% | 0.1% |
| SGC+LP | 3.8% | 2.6% | 1.1% | 0.4% | 0.2% | 0.2% | 0.1% | 0.0% |
| GATv2+LP | 21.8% | 17.4% | 8.6% | 10.8% | 5.5% | 4.3% | 1.6% | 0.4% |
| GAT+LP | 19.4% | 16.6% | 6.9% | 7.3% | 4.3% | 2.0% | 1.1% | 0.4% |
| GraphSAGE+LP | 17.6% | 15.9% | 12.9% | 9.8% | 6.2% | 3.2% | 1.1% | 0.1% |

Table 14: Over-Robustness $R^{over}$ measured using SGA with a budget of the degree of the target node. Standard deviations are insignificant and removed for brevity.

**GR-BCD**:

Table 14 shows how much robustness of the GNNs, when using GR-BCD as an attack, is actually over-robustness. GR-BCD is not applicable to all investigated GNNs in the study as explained in Appendix H. However, for the models GR-BCD shows significant over-robustness. However, GR-BCD shows the least over-robustness compared to the other attacks, indicated that its model-dependent attack strategy makes it better adapted to measure true adversarial robustness compared to model-agnostic attacks.

| | 0.1 | 0.5 | 1.0 | 1.5 | 2.0 | 3.0 | 4.0 | 5.0 |
|---|---|---|---|---|---|---|---|---|
| GCN | 11.9% | 7.6% | 3.9% | 1.4% | 0.6% | 0.1% | 0.1% | 0.0% |
| APPNP | 17.3% | 13.1% | 7.6% | 5.0% | 1.6% | 0.1% | 0.6% | 0.6% |
| GAT | 10.0% | 9.2% | 6.6% | 3.7% | 2.1% | 0.7% | 0.1% | 0.1% |
| GATv2 | 10.0% | 8.8% | 6.5% | 4.5% | 2.7% | 1.0% | 0.4% | 0.1% |
| GCN+LP | 3.0% | 1.9% | 1.0% | 0.6% | 0.4% | 0.2% | 0.1% | 0.1% |
| APPNP+LP | 10.5% | 6.6% | 3.9% | 2.3% | 0.3% | 0.1% | 0.4% | 0.3% |
| GATv2+LP | 6.0% | 4.1% | 2.7% | 2.3% | 1.6% | 0.8% | 0.3% | 0.1% |
| GAT+LP | 5.1% | 4.0% | 2.7% | 1.9% | 1.4% | 0.7% | 0.2% | 0.1% |

Table 15: Over-Robustness $R^{over}$ measured using GR-BCD with a budget of the degree of the target node. Standard deviations are insignificant and removed for brevity.

### J.6 ADVERSARIAL-ROBUSTNESS RESULTS FOR $\mathcal{B}_{deg}(\cdot)$

We investigate Nettack (Appendix J.6.1), DICE (Appendix J.6.2), SGA (Appendix J.6.3), $\ell_2$-strong (Appendix J.6.4) and GR-BCD (Appendix J.6.5). If not mentioned otherwise, error bars refer to the standard error.

#### J.6.1 NETTACK

Nettack, next to GR-BCD and SGA, is the strongest attack we employ, hence, it is a good heuristic to measure (semantic-aware) adversarial robustness $R^{adv}$ (see Appendix G) and conventional, non-semantics aware, adversarial robustness $R(f)$ (see Section 5). Figure 15 shows that surprisingly, MLP+LP has highest adversarial robustness, if structure matters $K \leq 3$.

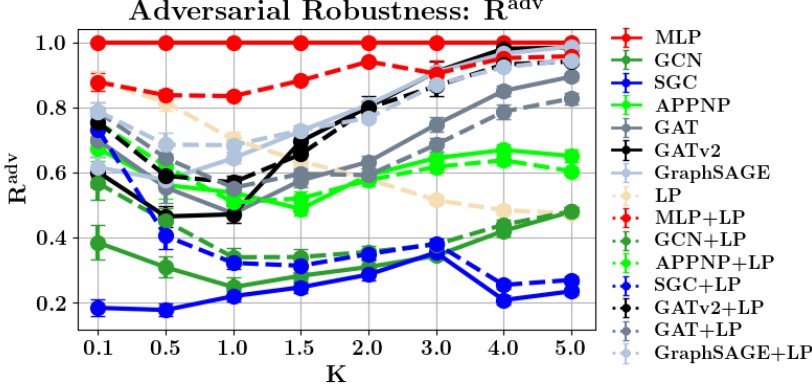

Figure 15: Semantic Aware Robustness $R^{adv}$ measured using Nettack.

Figure 16 shows that similar, albeit slightly less accurate adversarial robustness measurements are obtained by not including the semantic awareness. Rankings can indeed change, but only if these models are already very close regarding $R^{adv}$.

The harmonic mean of $R^{adv}$ and $R^{over}$ (see Metric-Section G) shows a complete picture of the robustness of the analysed models. To measure $R^{over}$, we use $\ell_2$-weak. Note that no GNN achieves top placements using this ranking, but the best models, depending on the amount of feature information, are LP, MLP+LP and MLP.

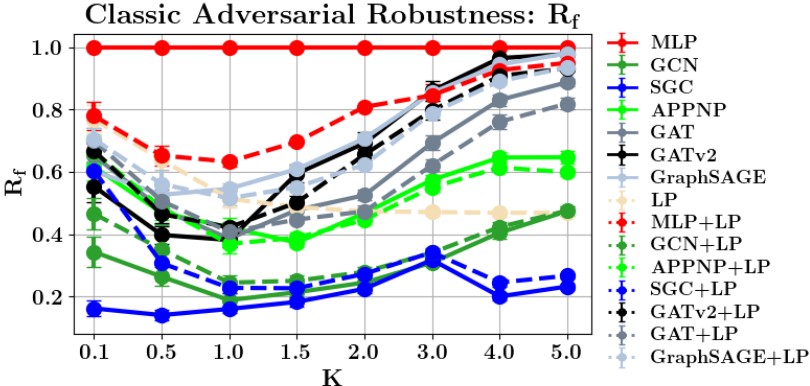

Figure 16: Conventional (Degree-Corrected) Robustness $R(f)$ measured using Nettack.

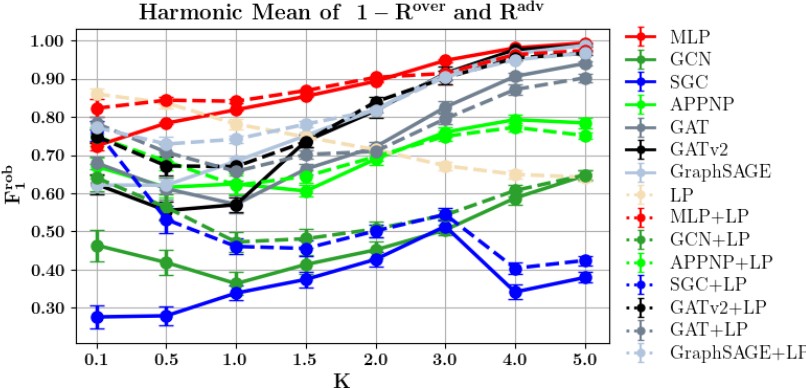

Figure 17: The harmonic mean of $R^{adv}$ and $1 - R^{over}$, with $R^{adv}$ measured using Nettack and $R^{over}$ using $\ell_2$-weak.

### J.6.2    DICE

DICE, as just randomly connecting to different class nodes, turns out to be a very weak attack similar to $\ell_2$-weak. Hence, its adversarial robustness counts differ significantly from the stronger Nettack. Here, we measure a significant difference between true (semantic-aware) adversarial robustness as presented in Figure 18 and conventional adversarial robustness as shown in Figure 19. Indeed, a conventional robustness measurement claims that LP is always the least robust method by significant margins, however, correcting for semantic change uncovers that LP actually is the most robust method for $K \leq 0.5$ and has competitive robustness for $K = 1$ (even until $K = 1.5$ looking at overall robustness 20). We find that with DICE we measure significantly higher over-robustness (Section J.5). Therefore, having a weak attack can result in a significantly different picture of the true compared to conventional adversarial robustness. This gives us insights for applying attacks on real-world graphs, calling for the importance of always choosing a strong, best adaptive attack (Tramèr et al., 2020b; Mujkanovic et al., 2022), if we want to gain insights into the true robustness of a defense and not be "fooled" by over-robust behaviour.

The harmonic mean of $R^{adv}$ and $R^{over}$ (Figure 20 shows a complete picture of the robustness of the analysed models. Note that again no GNN achieves top placements using this ranking, but the best models, depending on the amount of feature information, are LP, MLP+LP and MLP.

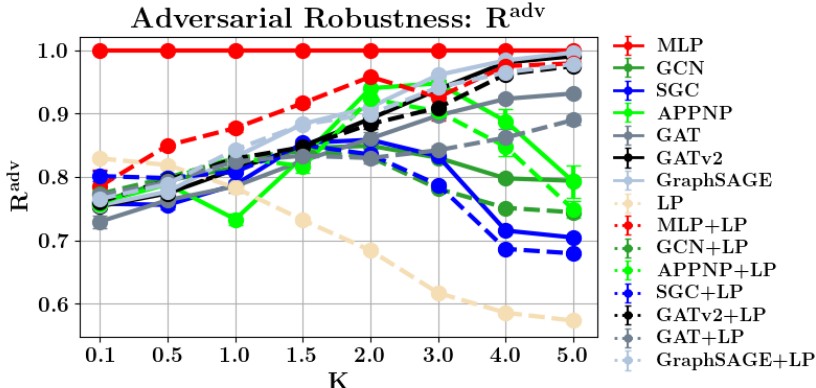

Figure 18: Semantic Aware Robustness $R^{adv}$ measured using DICE.

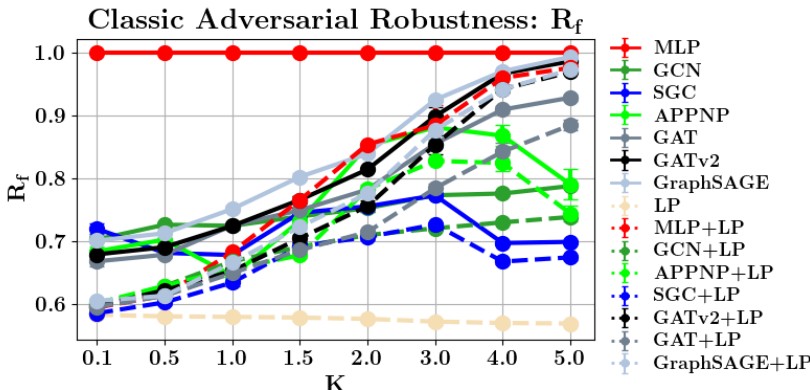

Figure 19: Conventional (Degree-Corrected) Robustness $R_f$ measured using DICE.

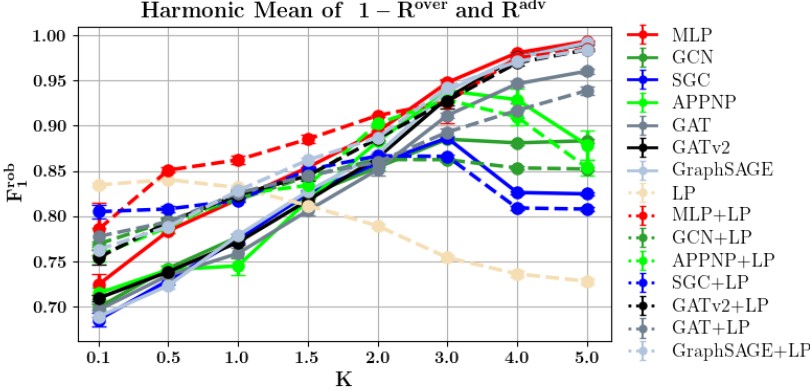

Figure 20: The harmonic mean of $R^{adv}$ and $1 - R^{over}$, with $R^{adv}$ measured using DICE and $R^{over}$ using $\ell_2$-weak.

### J.6.3 SGA

Albeit slightly weaker, we measure similar behaviour for SGA compared to Nettack, which can be explained as both use a SGC surrogate model. Figure 21 again shows that MLP+LP has highest adversarial robustness, if structure matters $K \leq 3$.

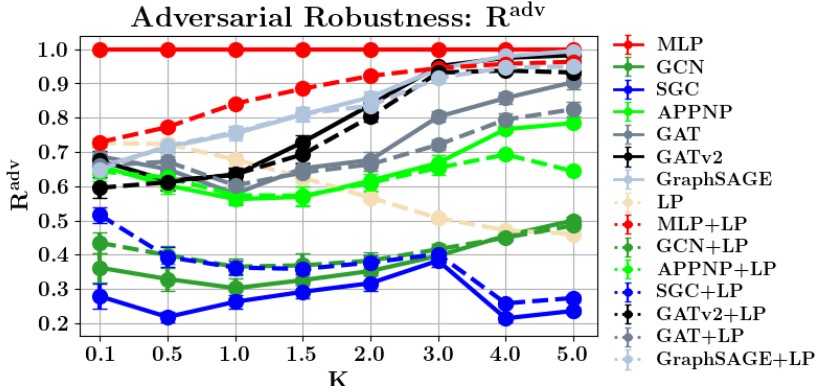

Figure 21: Semantic Aware Robustness $R^{adv}$ measured using SGA.

Figure 22 shows that because SGA is weaker than Nettack that similar to DICE robustness rankings are different using conventional adversarial robustness.

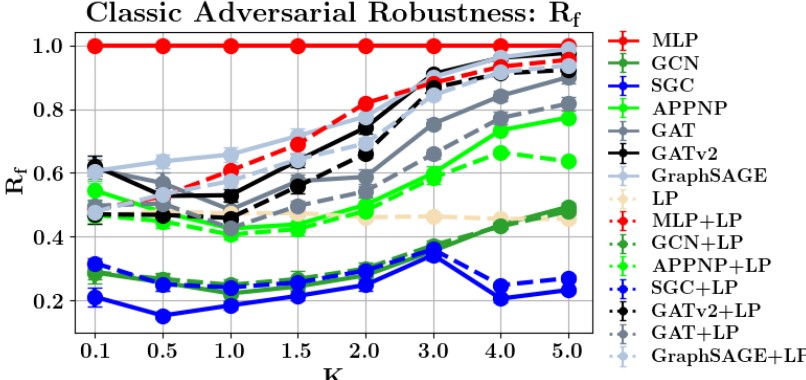

Figure 22: Conventional (Degree-Corrected) Robustness $R_f$ measured using SGA.

The harmonic mean of $R^{adv}$ and $R^{over}$ (Figure 23) shows a complete picture of the robustness of the analysed models. Note that as with Nettack and DICE, no GNN achieves top placements using this ranking, but the best models, depending on the amount of feature information, are LP, MLP+LP and MLP.

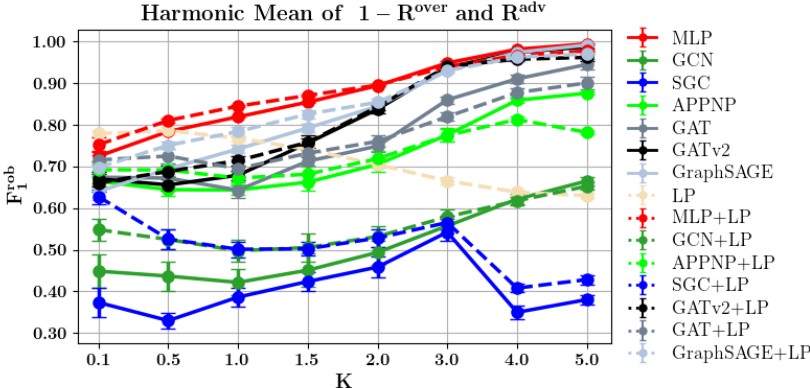

Figure 23: The harmonic mean of $R^{adv}$ and $1 - R^{over}$, with $R^{adv}$ measured using SGA and $R^{over}$ using $\ell_2$-weak.

### J.6.4 $\ell_2$-STRONG

We measure similar behaviour for $\ell_2$-strong compared to DICE, except that it performs way stronger against SGC and GCN, as these models are probably most effected by having neighbours with significantly different node features. Figure 21 shows semantic-aware adversarial robustness.

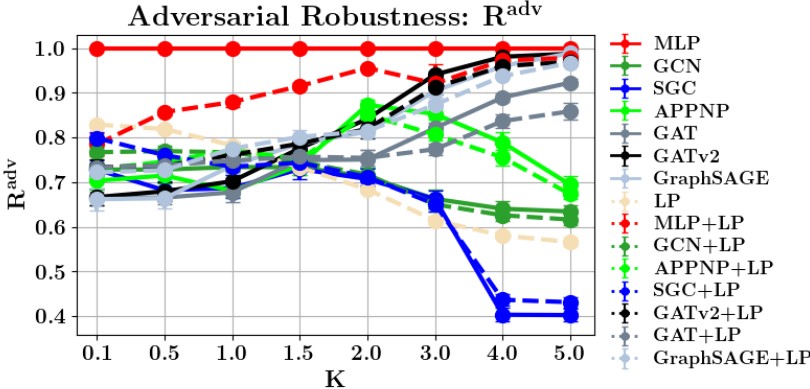

Figure 24: Semantic Aware Robustness $R^{adv}$ measured using $\ell_2$-strong.

Figure 25 shows adversarial robustness measurements not including the semantic awareness. $\ell_2$-strong seems suboptimal to distinguish different models for low $K$. Compared to the similar DICE, it ranks MLP+LP high for $K \geq 1.0$.

The harmonic mean of $R^{adv}$ and $R^{over}$ (Figure 26 shows a complete picture of the robustness of the analysed models. Note that as with Nettack, DICE and SGA, no GNN achieves top placements using this ranking, but the best models, depending on the amount of feature information, are LP, MLP+LP and MLP.

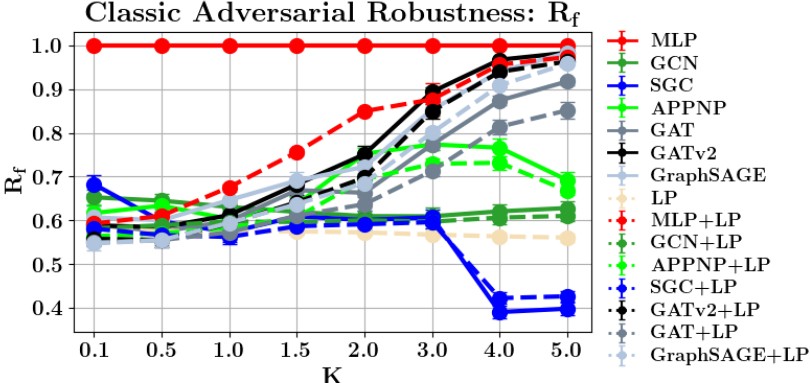

Figure 25: Conventional (Degree-Corrected) Robustness $R(f)$ measured using $\ell_2$-strong.

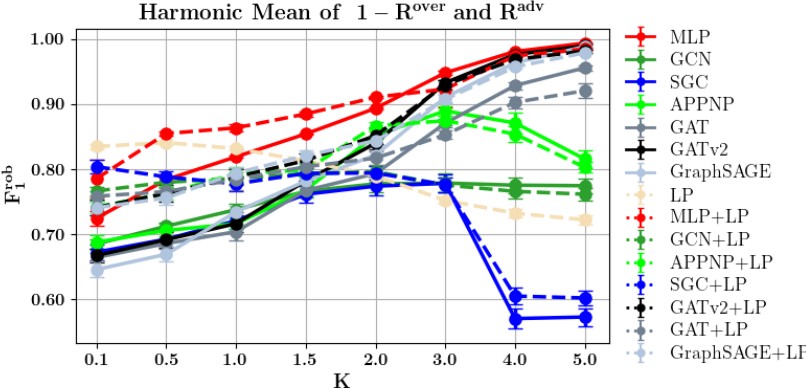

Figure 26: The harmonic mean of $R^{adv}$ and $1 - R^{over}$, with $R^{adv}$ measured using $\ell_2$-strong and $R^{over}$ using $\ell_2$-weak.

### J.6.5 GR-BCD

Figure 21 shows semantic-aware adversarial robustness of GR-BCD. GR-BCD is one of the strongest attacks we test, especially by taking the gradient w.r.t. the actually attacked model instead of using a surrogate model, it adapts to each individually. Because it is such a strong attacks, the semantic-aware adversarial robustness seems similar to the conventional adversarial robustness in Figure 28 and the harmonic mean in in Figure 29. However, semantic-aware adversarial robustness and the harmonic mean show that adding LP as postprocessing has a slightly higher effect than conventionally measurable.

GR-BCD provides strong evidence, that a strong, adaptive attack results in low over-robustness and hence, classic adversarial robustness is a good proxy for semantic-aware adversarial robustness and the over-all robustness as measured using the harmonic mean.

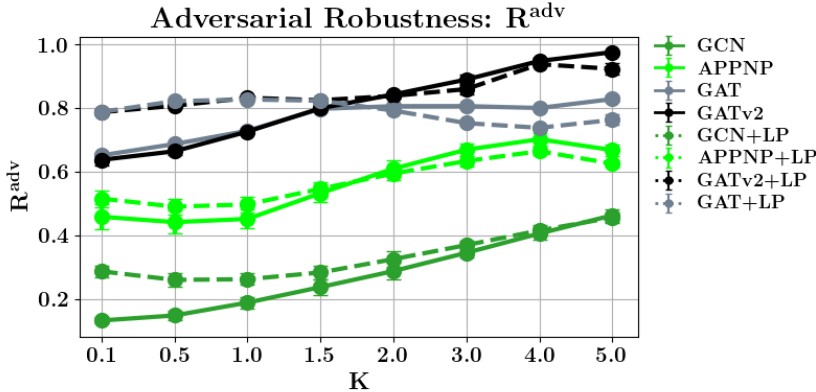

Figure 27: Semantic Aware Robustness $R^{adv}$ measured using GR-BCD.

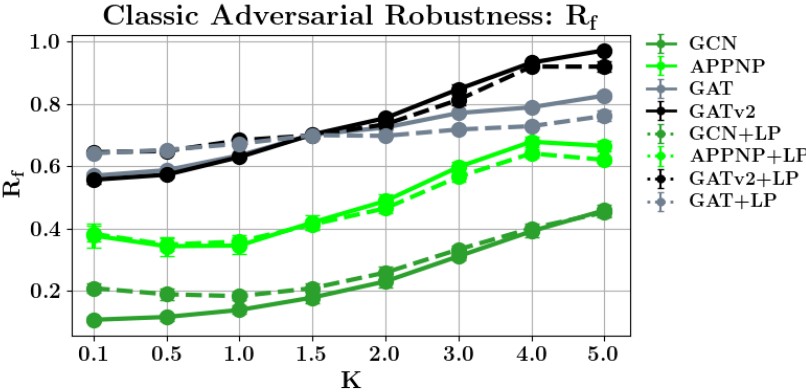

Figure 28: Conventional (Degree-Corrected) Robustness $R(f)$ measured using GR-BCD.

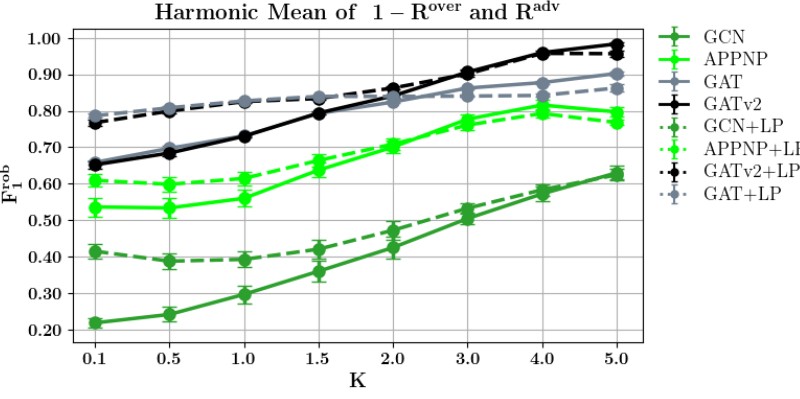

Figure 29: The harmonic mean of $R^{adv}$ and $1 - R^{over}$, with $R^{adv}$ measured using $\ell_2$-strong and $R^{over}$ using GR-BCD.

## J.7 STRUCTURE PERTURBATIONS UNTIL GCN-PREDICTION CHANGES ON CSBMS

Figure 30 shows that on a CSBM with $K = 1.5$, where a GCN already shows significant over-robustness (see Figure 3b), the GCN is less strongly robustness as on Cora-ML (compare to Figure 4).

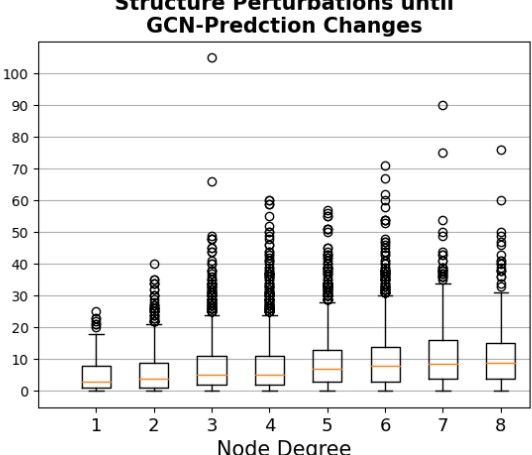

Figure 30: Robustness of GCN predictions on CSBMs with $K = 1.5$ as shown in Figure 1.

## J.8 AVERAGE ROBUSTNESS BAYES CLASSIFIER

Figure J.8 shows that the average robustness of the Bayes classifier increases with $K$ and is linear in the degree of the node. Furthermore, if structure matters ($K \leq 3$), average robustness rarely exceeds the degree of a node.

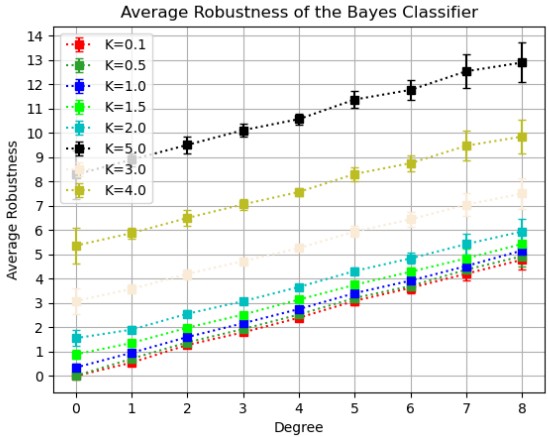

Figure 31: Average Robustness of the Bayes classifier on CSBMs.

## K FURTHER RESULTS ON REAL-WORLD GRAPHS

In this section, we show all results obtained on real-world graphs following the experimental setup described in Section H.1. The results for the fully supervised learning setting, which is more closer to our theoretic investigation, are outlined in Appendix K.1. The results for the semi-supervised learning setting, which is more common in practice, are outline in Appendix K.2

### K.1 SUPERVISED LEARNING SETTING

Table 16 visualizes model performance on the test set. GCN and GCN+LP achieve comparable accuracy while LP accuracy is slightly lower.

| Model
Dataset | GCN | GCN+LP | LP |
|---|---|---|---|
| Citeseer | $69.4 \pm 1.75$ | $69.5 \pm 1.62$ | $66.5 \pm 2.12$ |
| Cora-ML | $87.1 \pm 2.12$ | $86.7 \pm 1.80$ | $84.0 \pm 2.45$ |
| Cora | $87.6 \pm 2.02$ | $87.8 \pm 1.77$ | $85.4 \pm 1.87$ |
| Pubmed | $85.3 \pm 2.26$ | $84.3 \pm 2.26$ | $81.3 \pm 2.88$ |
| ogbn-arxiv | $70.2 \pm 0.65$ | $71.3 \pm 0.61$ | $70.0 \pm 1.21$ |

Table 16: Test accuracy mean and standard deviation over different data splits.

Figure 33 visualizes the robustness per node degree when applying the $\ell_2$-weak attack as described in H.1 on Cora-ML, Citeseer, Cora and Pubmed. The results for ogb-arxiv are presented in Figure 32. For each dataset, the mean minimal and maximal per-class robustness is depicted. The results are similar for all datasets. In general, robustness increases with increasing node degree. For all datasets and node degrees, GCN shows significant robustness to the structural changes, most pronounced on Pubmed (Figure 33h). The mean (maximal per-class) robustness is always a multiple of the considered node degree. LP is considerably less robust, while achieving similar test accuracy (Table 16). In general, less than half of the perturbations needed for GCNs suffice to change the LP's prediction. Combining a GCN with label propagation significantly and consistently reduces the robustness compared to GCN while again, preserving test accuracy (Table 16).

Figure 34 shows the distribution of $\max_{c \neq c^*} N_c(v)$ by degree of $v$. In contrast to Figure 33, the robustness of each node is visualized individually. The median robustness of GCN is significantly higher than that of GCN+LP which in turn is higher than that of LP. The difference between lower and upper quartile (IQR) is comparatively large for GCN, smaller for GCN+LP, and smallest for LP. Moreover, we observe that GCN has the largest amount of highly robust predictions. Applying LP to the GCN output drastically reduces the number and robustness of such outliers.

In conclusion, smoothing the GCN output using label propagation reduces the mean and median robustness, variance of robustness and decreases the number of highly robust predictions. Therefore, we conjecture that pairing GNNs with LP successfully combats over-robustness on real-world graphs.

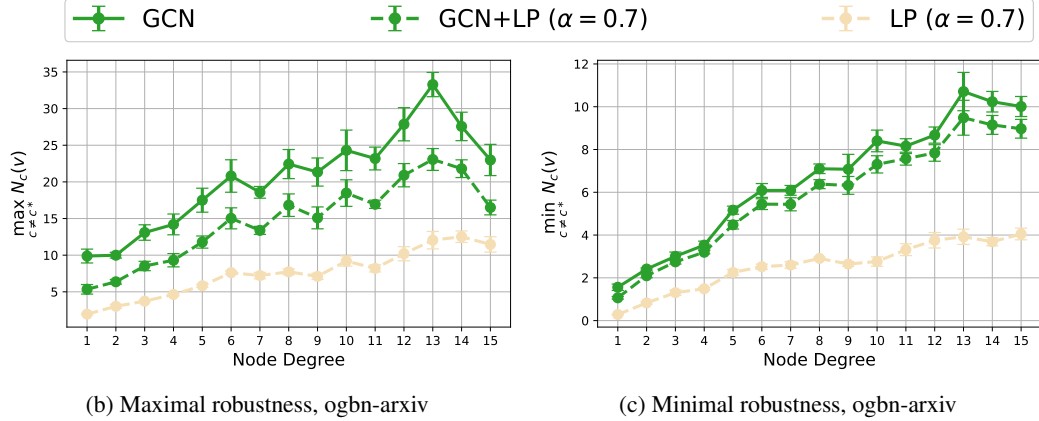

(b) Maximal robustness, ogbn-arxiv   (c) Minimal robustness, ogbn-arxiv

Figure 32: Mean robustness per node degree. Error bars indicate the standard error of the mean over five model initializations.

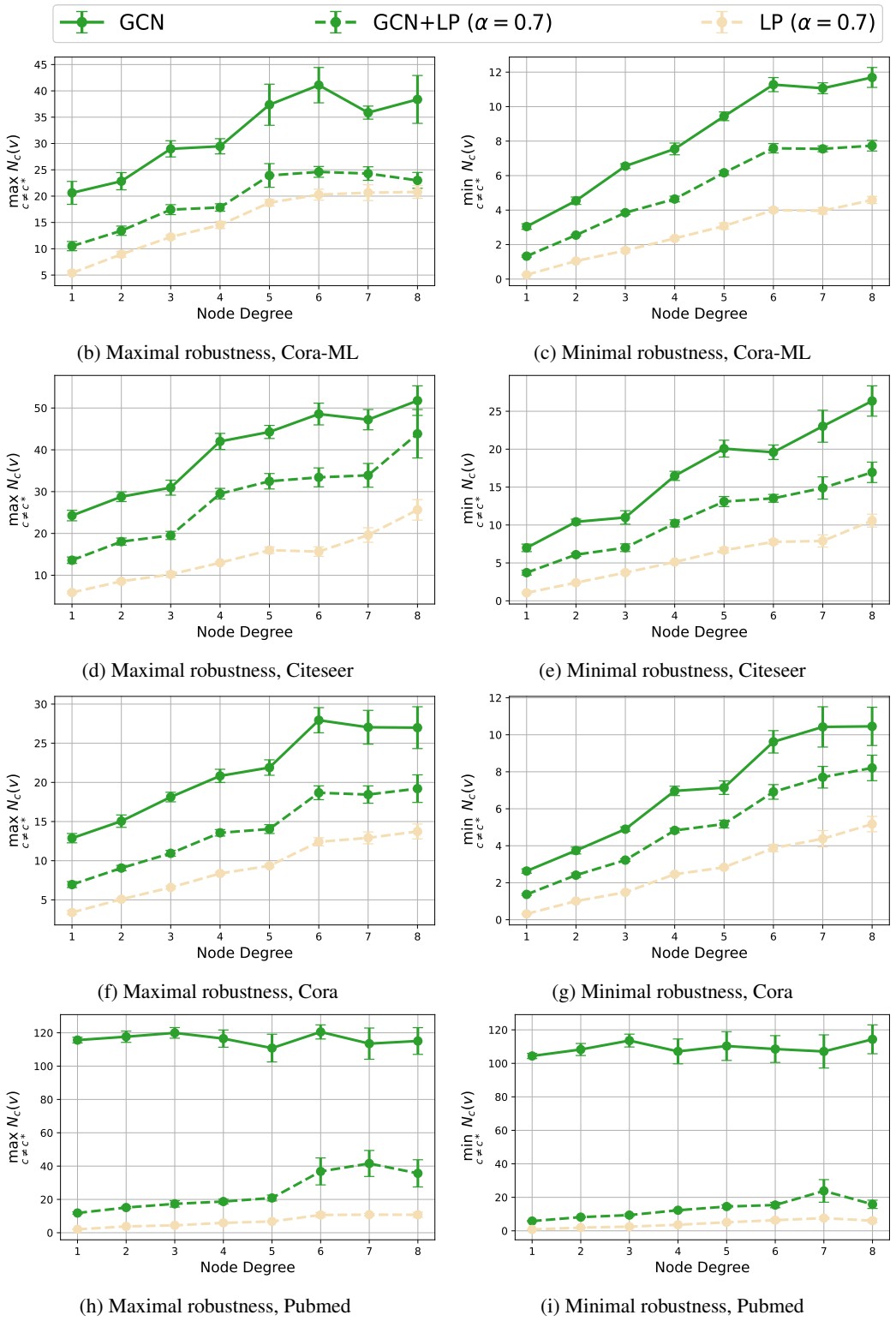

Figure 33: Mean robustness per node degree. Error bars indicate the standard error of the mean over eight data splits.

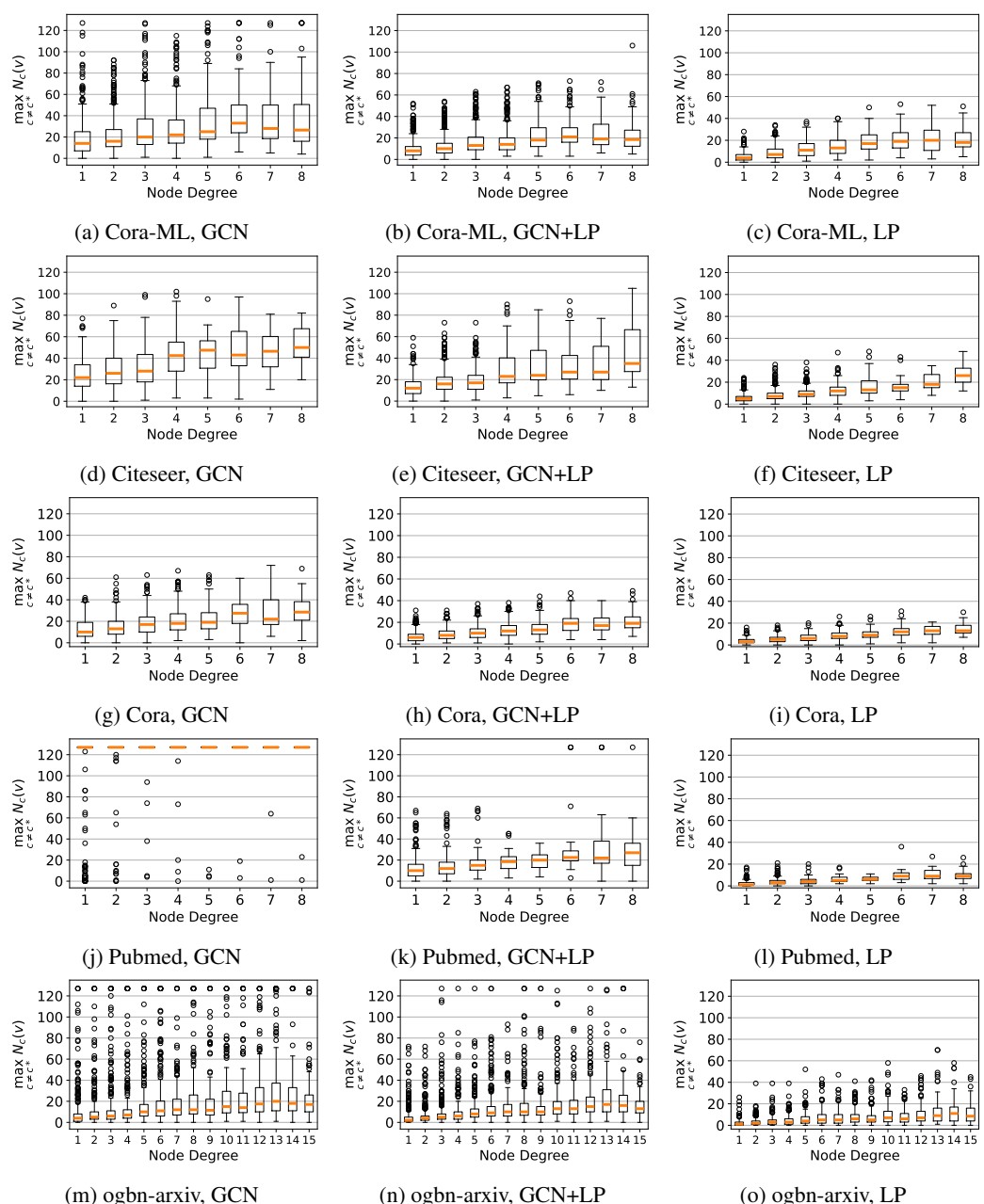

Figure 34: Distribution of maximal (per-class) node robustness by node degree. Results are aggregated over different data splits. The box captures the lower quartile (Q1) and upper quartile (Q3) of the data. The whiskers are placed at $Q1 - 1.5 \cdot IQR$ and $Q3 + 1.5 \cdot IQR$, where $IQR = Q3 - Q1$ is the interquartile range.

## K.2 SEMI-SUPERVISED LEARNING SETTING

Table 17 visualizes model performance on the test set. GCN and GCN+LP achieve comparable accuracy. Because of the low labeling rate (see Table 18), label propagation does not work as a stand-alone classification algorithm, achieving accuracies similar to chance. For completeness, we have still included LP robustness results in the following figures.

Figure 35 visualizes the robustness per node degree when applying the $\ell_2$-weak attack. Compared to the supervised learning setting, the mean robustness is considerably higher for all models. For GCN,

| Model
Dataset | GCN | GCN+LP | LP |
|---|---|---|---|
| Citeseer | $66.5 \pm 2.41$ | $66.5 \pm 2.15$ | $15.8 \pm 2.97$ |
| Cora-ML | $83.5 \pm 1.79$ | $84.0 \pm 2.78$ | $15.1 \pm 4.99$ |
| Cora | $82.3 \pm 1.42$ | $83.0 \pm 1.71$ | $13.0 \pm 2.27$ |
| Pubmed | $76.7 \pm 2.29$ | $76.9 \pm 2.91$ | $37.5 \pm 8.85$ |

Table 17: Model test accuracy and standard deviation over different data splits

| Nodes
Dataset | Training
(labeled) | Training
(unlabeled) | Validation | Test |
|---|---|---|---|---|
| Citeseer | 5.69 | 77.25 | 5.69 | 11.37 |
| Cora-ML | 4.98 | 80.07 | 4.98 | 9.96 |
| Cora | 5.17 | 79.32 | 5.17 | 10.34 |
| Pubmed | 0.30 | 98.78 | 0.30 | 0.61 |

Table 18: Percentage of training, validation and test nodes per dataset for the semi-supervised setting.

nodes of degree one have an average maximal (per-class) robustness of above $40$ on all evaluated datasets. The robustness of GCN and GCN+LP shows a relationship similar to the supervised learning setting. In general, GCN is more robust and additional smoothing with LP reduces the robustness while leaving the test accuracy unchanged (Table 17). Note that the LP results do not yield interpretable information, due to LP not working as a standalone classifier and have only been included for completeness. Figure 35 provides strong evidence, that pairing GNNs with LP significantly reduces over-robustness on real-world graphs in the realistic semi-supervised learning scenario with small labeling rates.

Figure 36 shows the distribution of $\max_{c \neq c^*} N_c(v)$ by degree of $v$. The median robustness of both GCN and GCN+LP is higher when compared to the supervised learning setting. However, again GCN+LP significantly reduces the very high robustness of a GCN, while achieving similar test accuracy (Table 17). Note that we only investigated node-level robustness until 128 edge insertions. On Pubmed, most predictions seem to be more robust than 128 edge insertions. However, adding LP as post-processing, still lets the whiskers and lower quartiles of the box-plot start earlier as without adding LP for most node degrees. In general, the interquartile range (IQR) of both GCN and GCN+LP is noticeable larger when comparing to the supervised learning setting. At the same time, the IQR of GCN+LP is again lower than that of GCN.

In conclusion, smoothing the GCN output using label propagation reduces the mean and median robustness as well as the variance of the robustness for the semi-supervised learning setting. Therefore, the results are consistent with the supervised learning setting. Thus, as pointed out above, they provide strong evidence, that pairing GNNs with LP significantly reduces over-robustness on real-world graphs in the semi-supervised learning scenario often found in practice.

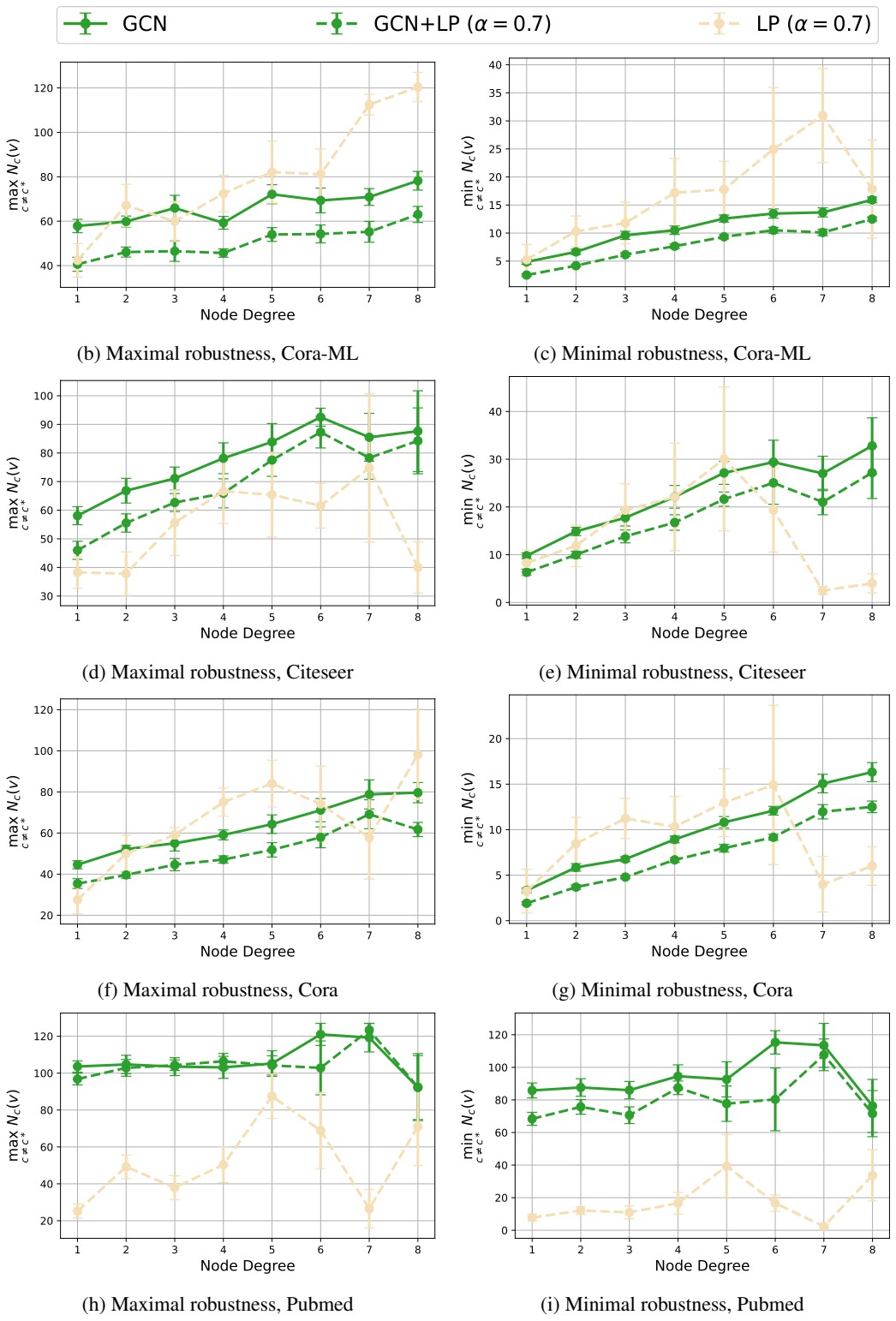

(b) Maximal robustness, Cora-ML

(c) Minimal robustness, Cora-ML

(d) Maximal robustness, Citeseer

(e) Minimal robustness, Citeseer

(f) Maximal robustness, Cora

(g) Minimal robustness, Cora

(h) Maximal robustness, Pubmed

(i) Minimal robustness, Pubmed

Figure 35: Mean robustness per node degree. Error bars indicate the standard error of the mean over eight data splits.

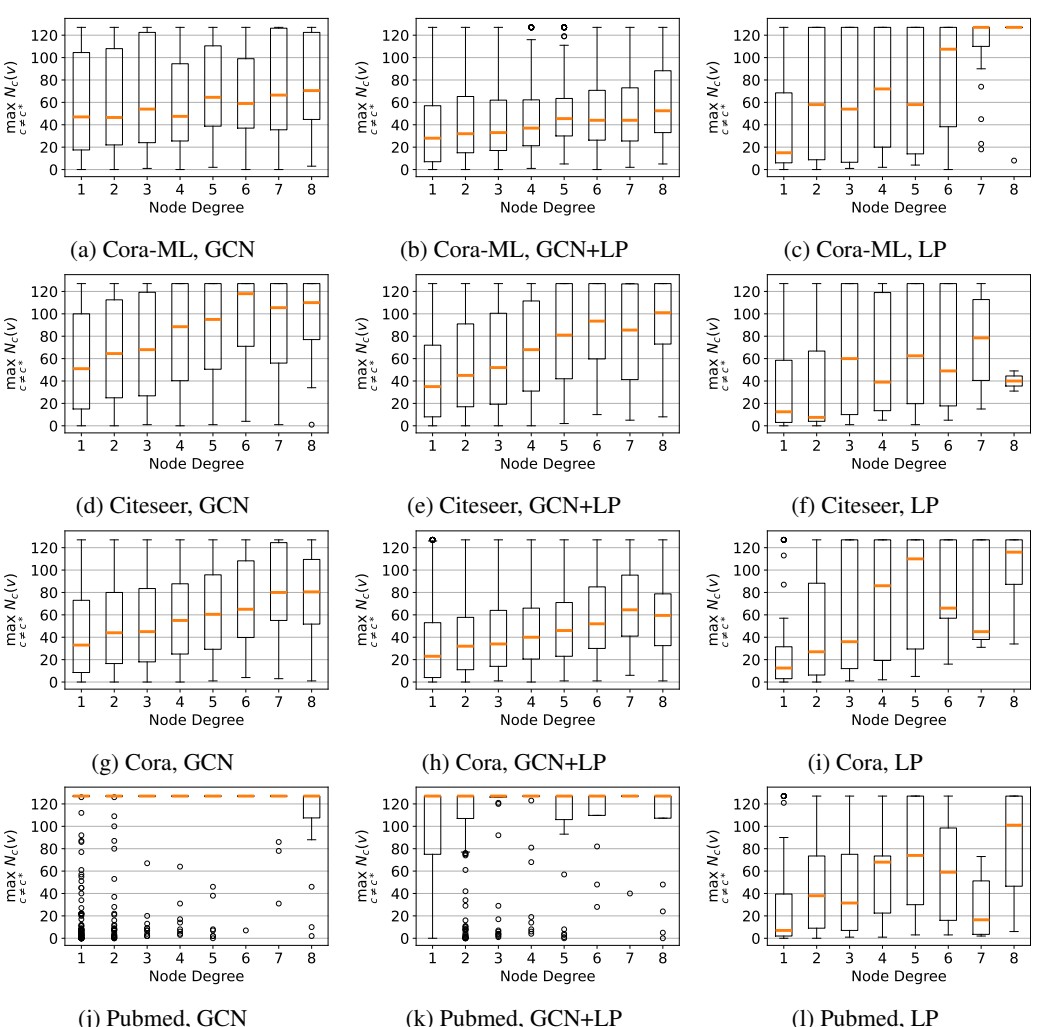

Figure 36: Distribution of maximal (per-class) node robustness by node degree. Results are aggregated over eight data splits. The box captures the lower quartile (Q1) and upper quartile (Q3) of the data. The whiskers are placed at $Q1 - 1.5 \cdot IQR$ and $Q3 + 1.5 \cdot IQR$, where $IQR = Q3 - Q1$ is the interquartile range.

## L    ADDITIONAL RELATED WORK

We see us related to works using synthetic graph models to generate principled insights into GNNs. Notably, Fountoulakis et al. (2022) show that in non-trivially CSBMs settings (hard regime), GATs, with high probability, can't distinguish same-class edges from different-class edges and degenerate to GCNs. Baranwal et al. (2021) study GCNs on CSBMs and find graph convolutions extent the linear separability of the data. Palowitch et al. (2022) generate millions of synthetic graphs to explore the performance of common GNNs on graph datasets with different characteristics to the common benchmark real-world graphs. However, their studied degree-corrected SBM is fundamentally limited to transductive learning.

Regarding the bigger picture in robust graph learning, all works measuring small changes to the graph's structure using the $\ell_0$-norm can be seen as related. This is a large body of work and includes but is not limited to i) the attack literature such as (Zügner et al., 2018; Dai et al., 2018; Waniek et al., 2018; Chen et al., 2018; Zügner & Günnemann, 2019a; Geisler et al., 2021); ii) various defenses ranging from detecting attacks (Wu et al., 2019b; Entezari et al., 2020), proposing new robust layers and architectures (Zhu et al., 2019; Geisler et al., 2020) to robust training schemes (Zügner & Günnemann, 2019b; Xu et al., 2019; 2020); iii) robust certification (Bojchevski et al., 2020; Schuchardt et al., 2021). An overview of the adversarial robustness literature on GNNs is given by Günnemann (2022). Zheng et al. (2021) provide a graph robustness benchmark.

Regarding sound perturbations models, distantly related is also the work of Geisler et al. (2022), which apply GNNs to combinatorial optimization tasks and therefore, can describe how the perturbations change or preserve the label and thereby, semantics.

