# OpenReview forum: "Revisiting Robustness in Graph Machine Learning"
_ICLR.cc/2023/Conference — ICLR 2023 poster_

### Official Review · Reviewer_b2Ux · 2022-10-25

**Confidence:** 4
**Correctness:** 3
**Technical Novelty And Significance:** 3
**Empirical Novelty And Significance:** 3
**Recommendation:** 6

**Clarity, Quality, Novelty And Reproducibility:**

The formulation and theoretical analysis of the semantic-preserving perturbations in graph adversarial attack is carefully defined and the proofs of the robustness-accuracy trade-off is clear. The notion of semantic-preserving for common graph adversarial attacks is novel. Moreover, the experimental setup is provided in Appendix for better reproducibility.

**Strength And Weaknesses:**

S1: Revisiting classic graph adversarial attacks from a semantic-preserving angle is novel and interesting.
S2: This paper leverages the synthetic graphs based on CSBMs and Bayes classifiers as reference classifiers to formulate the concept of over-robustness if the perturbation unnecessarily changes the semantics.
S3: The robustness-accuracy tradeoff is also discussed in inductive settings based on the constructed graphs and reference classifiers.

W1: Some works which propose the notion of unnoticeable perturbations mentioned in Introduction lack further comparison in the following part of this paper. whether these unnoticeable perturbations are semantic-preserving or not is unknown, at least from an experimental perspective.
W2: It would be better if more realistic datasets are discussed in this paper. Besides, when the CSBMs violates the real graph construction also needed more discussion.
W3: The attack algorithms used in the Results sections are targeted attacks. I wonder whether the untargeted attack like MetaAttack will also cause the same over-robustness.
W4: There are some typos in this paper, e.g., structure using is miswritten as structureusing in page 1, APPNP has no reference in page 6 section 5.


**Summary Of The Paper:**

This paper discusses the semantics-preserving adversarial attack in graph. Based on CSBMs, the constructed graphs show over-robustness of GNNs. The authors also prove that GNN+LP could be one way to reduce the over-robustness.

**Summary Of The Review:**

The concept of semantic-preserving perturbation and over-robustness of GNNs are interesting and attractive. The theoretical analysis based on these concepts is also sufficient. However, the authors lack more discussion with the unnoticeable perturbations mentioned in Introduction and the attack algorithm set in the experiment is relatively small.

---

> ### Author Response · Authors · 2022-11-17
> **Response to Reviewer b2Ux, 3/3**
>
> ### W3: Regarding Increasing the Set of Attacks and Untargeted Attacks
>
> Following your review, we extended our set of investigated attacks from 2 to 5 by adding the untargeted attack GR-BCD [13], the targeted attack SGA [14], and in analogy to $\ell_2$-weak by $\ell_2$-strong: Connect to the most dissimilar node in feature space. For all attacks, we find significant over-robustness.
>
> Note that MetaAttack is designed for a poisoning scenario, i.e., to insert malicious changes into the dataset before training. However, we study evasion attacks, i.e., we attack already trained GNNs to change their prediction. Thus, MetaAttack is not applicable to our study.
>
> Also note that in CSBMs evaluating a classifier associating the maximum likelihood class to multiple unlabeled nodes at once quickly becomes intractable [15]. This is as all possible labelings of the unlabeled nodes have to be taken into account, resulting in $\mathcal{O}(|\mathcal{Y}|^m)$ terms, where $|\mathcal{Y}|$ is the number of classes and $m$ the number of inductively added nodes. Therefore, in our evaluation framework, we follow our derivation of the Bayes optimal classifier (Theorem 1) and repeat sampling a new node and attacking it. Thus, there is always one node in the set of attacked nodes, resulting in an untargeted attack being equivalent to a targeted attack.
>
> [13] Geisler et al. "Robustness of Graph Neural Networks at Scale", NeurIPS 2021
> [14] Li et al. "Adversarial Attack on Large Scale Graph", TKDE 2021
> [15] Deshpande et al. "Contextual Stochastic Block Models", NeurIPS 2018
>
> ### W4: Typos
>
> We have fixed the typos mentioned in your review, together with other typos we have since found and fixed.

---

> ### Author Response · Authors · 2022-11-17
> **Response to Reviewer b2Ux, 2/3**
>
> ### W2 (a): More realistic datasets
>
> Following your suggestion, we have extended our work to now include experiments on the datasets Cora, Pubmed and ogbn-arxiv (next to Cora-ML and Citeseer already present). Furthermore, we dedicate a whole new chapter, Appendix J, on experiments on realistic datasets, previously found in Appendix E.5.5.
>
> In Appendix J.1, we investigate a supervised setting matching e.g., the official temporal split of ogbn-arxiv. Additionally, in Appendix J.2, we now adopt a realistic semi-supervised learning setting [1,2] under small labeling rates (20 labeled training nodes per class). Here, Label Propagation (LP) alone does not produce correct node classifications. However, as in the supervised case, we find that pairing GCN's predictions with LP significantly reduces excess node-level robustness, while preserving test accuracy.
>
> [6] Thomas N. Kipf and Max Welling. "Semi-supervised classification with graph convolutional networks.", ICLR 2017
> [7] Geisler et al. "Robustness of graph neural networks at scale.", NeurIPS 2021
>
>
> ### W2 (b): CSBMs as a model for real-world graphs.
>
> Following your suggestion, we now discuss the limitations of CSBMs in the newly added Appendix D. In particular, they do not follow a power-law degree distribution. Although CSBMs are the standard model in the literature studying GNNs on synthetic graphs [9, 10, 11, 12], we additionally added experiments using preferential attachment (PA) models (Appendix I) that generate a power-law degree distribution and for which we similarly find significant over-robustness of GNNs. Thus, the phenomenon of over-robustness also appears in graphs with power-law degree distribution.
>
> In Appendix D, we now also contrast CSBMs with alternative graph model such as the degree-corrected stochastic block model (DCSBM) or preferential attachment (PA) models. In the following, we give more details on why CSBMs are the most relevant model for our work, and refer to Appendix D for an extended discussion:
> * DCSBMs can model arbitrary degree distributions but require up-front knowledge of the size of the graph and the expected degree of each individual node. Then, its theoretic guarantees only hold for the graph of prespecified size and it does not define an iterative growth process. Thereby, its principled applicability is limited to transductive learning not matching our setting of inductive node classification.
> * SBMs have a long history of use in machine learning [8] and are the canonical models to study tasks such as community detection, making their extension with node features (CSBMs) an appealing choice of study for the GNN community and hence, as mentioned above, have been the model of choice in the related literature [9, 10, 11, 12].
> * PA models, compared to CSBMs, follow a power-law degree distributions, but do not exhibit community structure making them not applicable to study node classification. Therefore, we use an extension called the Barabási-Albert model with community structure [13] and thereby, are the first work to our knowledge, using PA models to study GNNs. However, all PA models with community structure define an iterative growth process with newly added nodes having a fixed low-degree. Therefore, inductive classification of newly added nodes biases the analysis towards (the more interesting) low-degree nodes.
>
> [8] Emmanuel Abbe. "Community detection and stochastic block models: Recent developments.", JMLR, 18(177):1–86, 2018
> [9] Baranwal et al. "Graph convolution for semi-supervised classification: Improved linear separability and out-of-distribution generalization.", ICML 2021
> [10] Fountoulakis et al. "Graph attention retrospective.", arXiv:2202.13060, 2022
> [11] Palowitch et al. "Graphworld: Fake graphs bring real insights for GNNs",  KDD ’22
> [12] Esser et al. "Learning Theory can (Sometimes) Explain Generalisation in Graph Neural Networks", NeurIPS 2022
> [13] Bruce Hajek and Suryanarayana Sankagiri. "Community recovery in a preferential attachment graph.", IEEE Transactions on Information Theory, 65(11):6853–6874, 2019

---

> ### Author Response · Authors · 2022-11-17
> **Response to Reviewer b2Ux, 1/3**
>
> Thank you very much for your positive and constructive review! We incorporated your raised points into our paper and thereby improved upon the submitted version and solidified our results (e.g., through additional attacks).
>
> ### W1: Further comparisons with proposed notions of unnoticeability and if they are semantic preserving or not
>
> Thank you for pointing this out! As a result, we added a new chapter B in the Appendix, where we formally and in detail discuss the limitations of the previously proposed notions of unnoticeability mentioned in the introduction [1, 2, 3], and why they do not preserve the semantic content in a graph.
>
> Additionally, in Appendix I, we added novel experiments highlighting empirically that these unnoticeability notions have close to no effect on semantic content preservation:
>
> **Regarding preserving the degree distribution [1].** The statistical test proposed by Zügner et al. (2018) [1] requires that the graph's degree distribution follows a power-law. However, this is not satisfied in CSBM graphs and hence, this test is not applicable to our studied graph generation model. As a result, we additionally use a Barabási–Albert graph generation model with community structure (CBA), explained in Appendix C, which follows a power-law degree distribution. Using CBAs, we find that even though we measure significant over-robustness, including Zügner et al.'s test for power-law degree distribution preservation has close to no effect on the measured over-robustness (Fig. 9/10 in Appendix I.1). Therefore, degree distribution preservation does not result in preserving the semantic content, as indicated from our theoretic discussion in Appendix B. Note that CBAs have other shortcomings, making CSBMs a better general model for our study. We added a discussion on graph generation model choices in Appendix D, see also our address for W2 (b) below.
>
> **Regarding degree assortativity [2].** Degree assortativity (DA) measures the correlation of nodes of certain degrees being connected. Li et al. (2021) [2] propose to measure unnoticeability through measuring the change in degree assortativity. However, they do *not* propose to restrict the perturbation set based on this measure. Particularly, in Table 4 in their work, they show that for most real-world graphs (e.g., Cora, Pubmed), the degree assortativity change for successful attacks, for which we measure substantial over-robustness, is significantly smaller than 1%. Furthermore, in Appendix B.2, similarly to the degree distribution, we formally show that preserving DA does not imply semantic content preservation and extensively discuss its limitations. Due to these empirical and theoretic results, degree assortativity change seems not to be the right measure for restricting perturbation sets and measuring semantic content preservation. Therefore, we have not included degree assortativity in our empirical study.
>
> **Other homophily metrics [3].** A similar argument to degree assortativity holds for the node- and edge-centric homophily metrics introduced by Chen et al. (2022) [3]. They use these measures to increase the unnoticeability for attacks inserting malicious nodes. However, they note that their introduced metrics are not able to capture the change in the graph when adding/removing malicious edges (see discussion to Figure 2c in Chen et al.). Particularly, they observe that when inserting malicious nodes, this produces a significant tail in the node-centric homophily distribution not seen in the original graph and propose to restrict the perturbation set by enforcing that the measured homophily does not surpass a fixed threshold. However, this tail in the homophily distributions is not seen when adding/removing malicious edges and hence, their perturbation set restriction is not applicable to the attack setting studied in our work. Due to these reasons, we have not included the homophily metrics by Chen et al. in our empirical study. A more detailed argument, including why our studied $\ell_2$-weak attack can preserve their homophily metrics, can be found in Appendix B.3.
>
> [1] Zügner et al. "Adversarial attacks on neural networks for graph data.", SIGKDD 2018
> [2] Li et al. "Adversarial attack on large scale graph.", IEEE Transactions on Knowledge and Data Engineering 2021
> [3] Chen et al. "Understanding and improving graph injection attack by promoting unnoticeability.", ICLR 2022

---

### Official Review · Reviewer_2ZaK · 2022-10-25

**Confidence:** 5
**Clarity, Quality, Novelty And Reproducibility:** The idea of this paper is novel. It i…
**Correctness:** 4
**Technical Novelty And Significance:** 3
**Empirical Novelty And Significance:** 3
**Recommendation:** 6

**Strength And Weaknesses:**

Strength:
The authors revisted the robustness of graph neural network, and proposed a method to detect the semantic content change of an adversarial graph, which is interesting and meaningful. The presentation is good, and easy to follow. The experimental results demonstrated the effectiveness of the proposed method.

Weakness:
1) It is expected to give more details about Figure 1.

2) There are two many symbols in the paper. I suggest the authors give a table to summary these symbols for reader understanding.

**Summary Of The Paper:**

The authors introduced a principled notion to be aware of semantic content change of an adversarial graph by  Contextual Stochastic Block Models (CSBMs). The experimental results showed for a majority of nodes the prevalent perturbation models include a large fraction of perturbed graphs violating the unchanged semantics assumption and all assessed GNNs show over-robustness - that is robustness beyond the point of semantic change.

**Summary Of The Review:**

I incline to accept this paper, because of its novelty.

---

> ### Author Response · Authors · 2022-11-17
> **Response to Reviewer 2ZaK**
>
> Thank you very much for your positive review! Below, we have addressed your mentioned weaknesses and by incorporating your suggestions, we improved the readability of our paper.
>
> ### 1. Regarding Figure 1
>
> We have reworked Figure 1 to improve its readability and made it more intuitive to understand by inverting the y-axis to result in a more natural presentation. We have further extended the description of the figure in the caption and added a footnote, which details the experimental setup used to produce its data.
>
> ### 2. Regarding a Symbol Table
>
> We have added a comprehensive symbol and abbreviations table in Appendix A and reference it in the Preliminaries.

---

### Official Review · Reviewer_bUiK · 2022-10-25

**Confidence:** 2
**Correctness:** 3
**Technical Novelty And Significance:** 3
**Empirical Novelty And Significance:** 3
**Recommendation:** 6

**Clarity, Quality, Novelty And Reproducibility:**

Generally, this paper is well-polished and easy to read.

- In the Introduction, Paragraph 2, the authors should elaborate more on the reasons other notions of perturbation sets (degree assortativity, other homophily metrics) are not semantically preserved.
- Section 5.2: the authors examine the over-robustness for \mathcal(B)_deg only. It would be nice to see the results with other perturbation radii.
- Page 8: The reason MLP provides an upper bound on the over-robustness for a particular K is unclear to me.
- The experiments should include other perturbation sets such as using degree distribution (Zugner et al. 2018) or degree assortativity (Li et al. 2021).


Minor comments:
- Legends of Figure 3 is not indicating dash lines.
- Citation error in paragraph 4, page 6.


**Strength And Weaknesses:**

Overall, the results of this paper could bring more insight into the current understanding of adversarial robustness for GNNs. The main result that indicates no robustness-accuracy tradeoff for inductively classifying a newly added node is interesting, even though a similar result is introduced by Suggala et al. (2019). This work focuses on non-i.i.d data instead of i.i.d. data in Suggala et al.

The study of over-robustness seems to be appealing. However, there is a concern about its practicality, as it requires the optimal (reference) classifier. Most of the analyses in this paper are on synthetic graphs. The results in Section 5.2.1 are not very significant to me. The authors should discuss how over-robustness could be avoided in real-world applications.



**Summary Of The Paper:**

The paper studies the robustness of machine learning models for node prediction problems. It is well-known that GNN models are sensitive to small perturbations in the graph structure. However, it is difficult to justify which perturbations are semantically-preserved for graph-like input (which is easier for images as humans are often considered robust judgers). The authors used CSBMs to generate synthetic graphs with which the optimal Bayes classifier is derivable and then used it as the reference classifier for semantic changes. Using the Bayes classifier, the authors introduce a notion of over-robustness: the prediction of the studied classifier is not changed for a perturbed graph (which is generally known as adversarial robustness), but the prediction of the reference classifier does. They then showed that there is no inherent trade-off between semantic-aware robustness and test accuracy, similar to Suggala et al. (2019) work. Empirically, they show that most GNN models suffer from over-robustness, and using Label Propagation on top of GNNs does alleviate them.

**Summary Of The Review:**

My initial impression of this paper is positive; thus, I lean towards acceptance of this paper.

---

> ### Author Response · Authors · 2022-11-17
> **Response to Reviewer bUiK, 1/2**
>
> Thank you very much for your positive review and your constructive feedback. In our revised paper, we incorporated your raised points (e.g., by adding more discussion/experiments on avoiding over-robustness in real-world applications) and thereby, improved our work.
>
> ### 1. How to avoid over-robustness in real-world applications
>
> We extended our study of avoiding (over-)robustness against unreasonably severe perturbations through pairing GNNs with Label Propagation (LP) on real-world graphs, by now also including experiments on Cora, Pubmed and ogbn-arxiv. Additionally, we dedicate a whole new chapter, Appendix K, on how to overcome over-robustness in real-world graphs, complementing Section 5.2.1. Similar experiments on Cora-ML & Citeseer have already been included in Appendix E.5.5 in the initially submitted paper.
>
> GNNs with LP consistently achieve comparable test accuracy, while bringing down the excessively high robustness of node predictions (Fig. 4 in Section 5.2.1) to more natural levels (Fig. 25 in Appendix K.1). For example, on Cora-ML we find nodes of degree 1 robust against connecting more than 120 nodes of a different class, strongly violating the homophily of the graph. Pairing the GNN with LP removes these robustness outliers resulting in a behaviour more consistent with the homophily found in the original graph. Therefore, this method is a practical approach to reduce over-robustness in real-world applications.
>
> ###  2. Regarding more elaboration why other notions of perturbation sets (degree assortativity, other homophily metrics) are not semantically preserved
>
> Thank you for pointing this out! As a result, in the Introduction, Paragraph 2, we now refer to the newly added Appendix B, which includes an extensive elaboration on this topic. There, we now formally and in detail discuss the limitations of the global graph properties proposed by [1,2] (degree distribution, degree assortativity) and the other homophily metrics by [3] and why they do not preserve the semantic content in a graph.
>
> [1] Zügner et al. "Adversarial attacks on neural networks for graph data.". SIGKDD 2018
> [2] Li et al. "Adversarial attack on large scale graph.", IEEE Transactions on Knowledge and Data Engineering 2021
> [3] Chen et al. "Understanding and improving graph injection attack by promoting unnoticeability.", ICLR 2022
>
> ### 3. Regarding over-robustness experiments for other perturbation radii
>
> Following your request, we added Appendix J.4, now complementing Section 5.2 with the over-robustness results for all other mentioned perturbation sets, namely $\mathcal{B}_1,\mathcal{B}_2,\mathcal{B}_3,\mathcal{B}_4$ and $\mathcal{B}_\text{deg+2}$. All perturbations sets show significant over-robustness, increasing with the allowed attack budget.
>
> ### 4. Regarding the reasons an MLP provides an upper bound on over-robustness
>
> An MLP bases its decision solely on the node's features ignoring the structure of the graph. Therefore, an MLP shows perfect robustness against any attack. Now, some of the studied attacks are model independent, as their decisions either do not depend on any model ($\ell_2$-weak, $\ell_2$-strong) or use a surrogate model instead (Nettack, SGA). Now, given a graph sampled from a CSBM with a particular $K$. These attacks will choose the same malicious edges to insert for every investigated model. As an MLP has perfect robustness, the attack will exhaust its complete budget in trying to change the MLP's prediction without achieving success. In other words, given any perturbation budget $\Delta$, we measure the maximal achievable robustness $\Delta$. Only, $\tilde{\Delta} \le \Delta$ perturbations preserve the semantics and not perfectly robust models may change their prediction earlier than at $\Delta$ perturbations, resulting in either less or similar over-robustness to an MLP. Therefore, using an MLP, for any given node, we always measure the maximal (over-)robustness for a given budget $\Delta$ using a given attack.
>
> To make this clearer in the paper, we added more explanation on why an MLP provides an upper bound on over-robustness on page 8, Section 5.

---

> ### Author Response · Authors · 2022-11-17
> **Response to Reviewer bUiK, 2/2**
>
> ### 5. Regarding experiments using other perturbations sets such as using degree distribution or degree assortativity
>
> We added new experiments in Appendix I, highlighting empirically that other perturbation sets using e.g., degree distribution preservation, have close to no effect on semantic content preservation:
>
> **Regarding preserving the degree distribution.** The statistical test proposed by Zügner et al. (2018) requires that the graph's degree distribution follows a power-law. However, this is not satisfied in CSBM graphs and hence, this test is not applicable to our studied graph generation model. As a result, we additionally implemented a Barabási–Albert graph generation model with community structure (CBA), explained in Appendix C, which follows a power-law degree distribution. Using CBAs, we find that even though we measure significant over-robustness, including Zügner et al.'s test for power-law degree distribution preservation has close to no effect on the measured over-robustness. Therefore, degree distribution preservation does not result in preserving the semantic content (as indicated from our theoretic discussion in Appendix B). Note that CBAs have other shortcomings, making CSBMs a better general model for our study. We added a discussion on graph generation model choices in Appendix D.
>
> **Regarding degree assortativity.** Degree assortativity (DA) measures the correlation of nodes of certain degrees being connected. Li et al. (2021) proposed to measure unnoticeability through measuring the change in degree assortativity. However, they did *not* propose to restrict the perturbation set based on this measure. Particularly, in Table 4 in their work, they show that for most real-world graphs (e.g., Cora, Pubmed), the degree assortativity change for successful attacks, for which we measure substantial over-robustness, is significantly smaller than 1%. Furthermore, in Appendix B.2, similarly to the degree distribution, we formally show that preserving DA does not imply semantic content preservation and extensively discuss its limitations. Due to these empirical and theoretic results, degree assortativity change seems not to be the right measure for restricting perturbation sets and measuring semantic content preservation. Therefore, we have not included degree assortativity in our empirical study.
>
> **Other homophily metrics by Chen et al. (2022).** We want to note that a similar argument to degree assortativity holds for the node- and edge-centric homophily metrics introduced by Chen et al. (2022). They use these measures to increase the unnoticeability for attacks inserting malicious nodes. However, they note that their introduced metrics are not able to capture the change in the graph when adding/removing malicious edges (see discussion to Figure 2c in Chen et al.). Particularly, they observe that when inserting malicious nodes, this produces a significant tail in the node-centric homophily distribution not seen in the original graph and propose to restrict the perturbation set by enforcing that the measured homophily does not surpass a fixed threshold. However, this tail in the homophily distributions is not seen when adding/removing malicious edges and hence, their perturbation set restriction is not applicable to the attack setting studied in our work. Due to these reasons, we have not included the homophily metrics by Chen et al. (2022) in our empirical study. A more detailed argument, including why our studied $\ell_2$-weak attack can preserve their homophily metrics, can be found in Appendix B.3.
>
> ### 6. Regarding the minor comments
>
> * We fixed the legend of Figure 3 to indicate the dashed lines. We have also changed the font used in the image to match the main text. (Same for Figure 15 and 16 in Appendix J.6.)
> * We fixed the citation error in paragraph 4, page 6.

---

### Author Response · Authors · 2022-11-17
**Meta Response**

We want to thank the reviewers for their helpful feedback. Based on it, we have added several interesting and complementary results in the revised paper.

In particular, we added the following **new experiments**:

* J.3, J.5, J.6: **Added attacks** GR-BCD, SGA and $\ell_2$-strong
* J.4: **Included** over-robustness results for all **other perturbation sets** $\mathcal{B}_1$, $\mathcal{B}_2$, $\mathcal{B}_3$, $\mathcal{B}_4$ and $\mathcal{B}_\text{deg+2}$
* Appendix K:
    * **Added real-world datasets** Cora, Pubmed and ogbn-arxiv (next to Cora-ML and Citeseer already in the initial version of the paper).
    * **Added** experiments in the **semi-supervised learning setting** (K.2), next to the supervised learning setting (K.1)
* Appendix I: **Added** experiments using a Contextual **Barabási–Albert Model** with Community Structure (CBA), showing:
    * The phenomenon of over-robustness also appears in graphs with power-law degree distribution.
    * Preserving degree-distribution does not combat over-robustness

**Further paper updates.** Additionally, we added the following sections and discussions:

* Appendix A: List of symbols and abbreviations.
* Appendix B: Discussion on why previous notions of unnoticeability, in particular the global graph properties proposed by Zügner et al. 2018 (degree distribution) and Li et al. 2021 (degree assortativity) as well as the other homophily metrics by Chen et al. 2022 do not preserve the semantic content in a graph.
* Appendix C: Detailed description of an alternative graph generation model to CSBMs called Contextual Barabási–Albert Model with Community Structure (CBA), which exhibits a power-law degree distribution and hence, allows to study over-robustness for graphs with power-law degree distribution as well as the effects of Zügner et al.'s degree-distribution-preserving test.
* Appendix D: Discussion on limitations of CSBMs and why they are chosen as the graph generation models of choice. Contrast CSBMs with alternative graph generation models and present their respective strengths and weaknesses for studying inductive node classification.
* Appendix F.4: Prove of a new theorem (Theorem 4) deriving an optimal attack against the Bayes classifier on CBAs (similar to Theorem 3 for CSBMs in App. F.3)
* Appendix K: Added extensive discussion how to avoid (over-)robustness in real-world graphs

**Minor Changes.**
- Improved Figure 1 and gave more details
- Legends of Figure 3 correctly indicate dashed lines
- Section 5.2: Improved presentation why an MLP can provide an upper bound on over-robustness.
- Fixed small typos in the manuscript, among others:
    - Robustness definitions 1 and 2 correctly use $y_L$ instead of $y_{trn}$
    - Citation error in Section 5
- Added references in the main text to the newly added discussions and results in the Appendix

**Update 18.11.**
- Corrected wording in Definition 4 (expected over-robust loss)
- Fixed formatting of Section 4 heading

---

### Author Response · Authors · 2022-12-06
**Further Questions**

We hope our responses and new experiments could answer all questions raised by the reviewers. If anything remains unclear or new questions arose, we are happy to engage in a further discussion.

---

### Decision · Program_Chairs · 2023-01-20

**Decision:**

Accept: poster

**Justification For Why Not Higher Score:**

There are weaknesses identified by the reviewers and AC, as pointed out in the meta review.

**Justification For Why Not Lower Score:**

This is the first paper studying over robustness in GNN, and this is the first paper connecting it with optimal Bayes classifier and formally investigate this phenomenon on synthetic data.

**Metareview: Summary, Strengths And Weaknesses:**

The paper studies adversarial robustness of graph neural networks. The authors formally define the "semantically preserved perturbation set" based on the optimal Bayes classifier and found that many GNNs are "over-robust" with respect to this perturbation set. This new finding contradicts with previous work (which often defined "semantic preservation" in an ad-hoc way) and leads to several interesting future directions. Although all reviewers agreed that this is an interesting paper, during the zoom discussion we identified several weaknesses of the paper:

- (Please address in the camera ready version.) The notion "over-robust" is not new to the adversarial robustness community. In fact, it has been implicitly discussed in (Jia and Liang, 2017) where they define this as "overstability". More recently, the same concept has also been discussed in (Tramer et al., 2020) in the computer vision domain, where they found many (MNIST/CIFAR) networks are over robust to semantically preserved perturbations. Please cite and discuss those previous papers in the camera ready version.
Despite that over-robustness is not a new idea, this paper is the first formally defining and measuring it in the context of GNN. Further, the paper showed that it's possible to formally measure over-robustness in the synthetic case, where the data distribution and optimal Bayes classier is known.

- Real world evaluation is the main weakness of this work, recognized by most of the reviewers and the AC. Since it's hard to get the optimal Bayes classifier, the conclusion on real world data is somehow questionable. However, reviewers and AC think the theoretical analysis on synthetic case is interesting enough. It will be great if the authors can discuss their limitations in real world study in the camera ready version, which may motivate future study (potentially involve human evaluation) on real world problems.

(Jia and Liang, 2017) Adversarial Examples for Evaluating Reading Comprehension Systems.

(Tramer et al., 2020) Fundamental Tradeoffs between Invariance and Sensitivity to Adversarial Perturbations.

**Note From Pc:**

if the above contains the word "oral" or "spotlight" please see: "oral" presentation means -> notable-top-5% and "spotlight" means -> notable-top-25%. As stated in our emails, we are disassociating presentation type from AC recommendations

**Summary Of Ac-Reviewer Meeting:**

All the reviewers are happy with the submission but think real world evaluation is the main weakness of the paper. However, we think the theoretical analysis and synthetic examples in this paper is interesting enough for the publication.